

# 1 Global Carbon Budget 2024

Pierre Friedlingstein [1,2], Michael O'Sullivan [1], Matthew W. Jones [3], Robbie M. Andrew [4], Judith Hauck
[5,6], Peter Landschützer [7], Corinne Le Quéré [3], Hongmei Li [8,9], Ingrid T. Luijkx [10], Are Olsen [11,12],
Glen P. Peters [4], Wouter Peters [10,13], Julia Pongratz [14,9], Clemens Schwingshackl [14], Stephen Sitch
[1], Josep G. Canadell [15], Philippe Ciais [16], Robert B. Jackson [17], Simone R. Alin [18], Almut Arneth
[19], Vivek Arora [20], Nicholas R. Bates [21], Meike Becker [11,12], Nicolas Bellouin [22], Carla F. Berghoff
[23], Henry C. Bittig [24], Laurent Bopp [2], Patricia Cadule [2], Katie Campbell [25], Matthew A.
Chamberlain [26], Naveen Chandra [27], Frédéric Chevallier [16], Louise P. Chini [28], Thomas Colligan [29],
Jeanne Decayeux [30], Laique M. Djeutchouang [31,32], Xinyu Dou [33], Carolina Duran Rojas [1], Kazutaka
Enyo [34], Wiley Evans [25], Amanda R. Fay [35], Richard A. Feely [18], Daniel. J. Ford [1], Adrianna Foster
[36], Thomas Gasser [37], Marion Gehlen [16], Thanos Gkritzalis [7], Giacomo Grassi [38], Luke Gregor [39],
Nicolas Gruber [39], Özgür Gürses [5], Ian Harris [40], Matthew Hefner [41,42], Jens Heinke [43], George C.
Hurtt [28], Yosuke Iida [34], Tatiana Ilyina [44,8,9], Andrew R. Jacobson [45], Atul K. Jain [46], Tereza
Jarníková [47], Annika Jersild [29], Fei Jiang [48], Zhe Jin [49,50], Etsushi Kato [51], Ralph F. Keeling [52],
Kees Klein Goldewijk [53], Jürgen Knauer [54,15], Jan Ivar Korsbakken [4], Siv K. Lauvset [55,12], Nathalie
Lefèvre [56], Zhu Liu [33], Junjie Liu [57,58], Lei Ma [28], Shamil Maksyutov [59], Gregg Marland [41,42],
Nicolas Mayot [60], Patrick C. McGuire [61], Nicolas Metzl [56], Natalie M. Monacci [62], Eric J. Morgan
[52], Shin-Ichiro Nakaoka [59], Craig Neill [26], Yosuke Niwa [59], Tobias Nützel [14], Lea Olivier [5],
Tsuneo Ono [63], Paul I. Palmer [64,65], Denis Pierrot [66], Zhangcai Qin [67], Laure Resplandy [68], Alizée
Roobaert [7], Thais M. Rosan [1], Christian Rödenbeck [69], Jörg Schwinger [55,12], T. Luke Smallman
[64,65], Stephen M. Smith [70], Reinel Sospedra-Alfonso [71], Tobias Steinhoff [72,55], Qing Sun [73],
Adrienne J. Sutton [18], Roland Séférian [30], Shintaro Takao [59], Hiroaki Tatebe [74,75], Hanqin Tian [76],
Bronte Tilbrook [26,77], Olivier Torres [2], Etienne Tourigny [78], Hiroyuki Tsujino [79], Francesco Tubiello
[80], Guido van der Werf [10], Rik Wanninkhof [66], Xuhui Wang [50], Dongxu Yang [81], Xiaojuan Yang
[82], Zhen Yu [83], Wenping Yuan [84], Xu Yue [85], Sönke Zaehle [69], Ning Zeng [86, 29], Jiye Zeng [59].
1. Faculty of Environment, Science and Economy, University of Exeter, Exeter EX4 4QF, UK
2. Laboratoire de Météorologie Dynamique, Institut Pierre-Simon Laplace, CNRS, Ecole Normale Supérieure,
Université PSL, Sorbonne Université, Ecole Polytechnique, Paris, France
3. Tyndall Centre for Climate Change Research, School of Environmental Sciences, University of East Anglia,
Norwich Research Park, Norwich NR4 7TJ, UK
4. CICERO Center for International Climate Research, Oslo 0349, Norway
5. Alfred-Wegener-Institut, Helmholtz-Zentrum für Polar- und Meeresforschung, Am Handelshafen 12, 27570
Bremerhaven, Germany
6. Universität Bremen, Bremen, Germany
7. Flanders Marine Institute (VLIZ), Jacobsenstraat 1, 8400, Ostend, Belgium
8. Helmholtz-Zentrum Hereon, Max-Planck-Straße 1, 21502 Geesthacht, Germany



9. Max Planck Institute for Meteorology, Bundesstraße 53, 20146 Hamburg, Germany
10. Wageningen University, Environmental Sciences Group, P.O. Box 47, 6700AA, Wageningen, The
Netherlands
11. Geophysical Institute, University of Bergen, Allégaten 70, 5007 Bergen, Norway
12. Bjerknes Centre for Climate Research, Bergen, Norway
13. University of Groningen, Centre for Isotope Research, Groningen, The Netherlands
14. Ludwig-Maximilians-Universität München, Luisenstr. 37, 80333 München, Germany
15. CSIRO Environment, Canberra, ACT 2101, Australia
16. Laboratoire des Sciences du Climat et de l'Environnement, LSCE/IPSL, CEA-CNRS-UVSQ, Université
Paris-Saclay, F-91198 Gif-sur-Yvette, France
17. Department of Earth System Science, Woods Institute for the Environment, and Precourt Institute for
Energy, Stanford University, Stanford, CA 94305–2210, United States of America
18. National Oceanic and Atmospheric Administration, Pacific Marine Environmental Laboratory
(NOAA/PMEL), 7600 Sand Point Way NE, Seattle, WA 98115, USA
19. Karlsruhe Institute of Technology, Institute of Meteorology and Climate Research/Atmospheric
Environmental Research, 82467 Garmisch-Partenkirchen, Germany
20. Canadian Centre for Climate Modelling and Analysis, Environment and Climate Change Canada, Victoria,
BC, Canada
21. ASU-BIOS, Bermuda Institute of Ocean Sciences, 31 Biological Lane, Ferry Reach, St. Georges,, GE01,
Bermuda
22. Department of Meteorology, University of Reading, Reading, RG6 6BB, UK
23. Instituto Nacional de Investigación y Desarrollo Pesquero, Paseo Victoria Ocampo Nº1, Escollera Norte,
B7602HSA, Mar del Plata, Argentina
24. Leibniz Institute for Baltic Sea Research Warnemuende (IOW), Seestrasse 15, 18119 Rostock, Germany
25. Hakai Institute, British Columbia, V0P 1H0, Canada
26. CSIRO Environment, Castray Esplanade, Hobart, Tasmania 7004, Australia
27. Research Institute for Global Change, JAMSTEC, 3173-25 Showa-machi, Kanazawa, Yokohama, 236-0001,
Japan
28. Department of Geographical Sciences, University of Maryland, College Park, Maryland 20742, USA
29. Earth System Science Interdisciplinary Center, University of Maryland, College Park, MD 20740, USA
30. Centre National de Recherches Météorologiques, Université de Toulouse, Météo-France, CNRS UMR 3589,
Toulouse, France
31. School for Climate Studies, Stellenbosch University, Private Bag X1, Matieland, Stellenbosch, 7602, South
Africa
32. Southern Ocean Carbon – Climate Observatory, CSIR, Rosebank, Cape Town, 7700, South Africa
33. Department of Earth System Science, Tsinghua University, Beijing, China
34. Japan Meteorological Agency, 3-6-9 Toranomon, Minato City, Tokyo 105-8431, Japan
35. Columbia University and Lamont-Doherty Earth Observatory, New York, NY, USA





36. Climate and Global Dynamics Laboratory, National Center for Atmospheric Research, Boulder, CO 80305,
USA
37. International Institute for Applied Systems Analysis (IIASA), Schlossplatz 1, A-2361 Laxenburg, Austria
38. European Commission, Joint Research Centre, 21027 Ispra (VA), Italy
39. Environmental Physics Group, ETH Zürich, Institute of Biogeochemistry and Pollutant Dynamics and
Center for Climate Systems Modeling (C2SM), Zürich, Switzerland
40. NCAS-Climate, Climatic Research Unit, School of Environmental Sciences, University of East Anglia,
Norwich Research Park, Norwich, NR4 7TJ, UK
41. Research Institute for Environment, Energy, and Economics, Appalachian State University, Boone, North
Carolina, USA
42. Department of Geological and Environmental Sciences, Appalachian State University, Boone, North
Carolina, USA
43. Potsdam Institute for Climate Impact Research (PIK), member of the Leibniz Association, P.O. Box 60 12
03, 14412 Potsdam, Germany
44. Universität Hamburg, Bundesstraße 55, 20146 Hamburg, Germany
45. Cooperative Institute for Research in Environmental Sciences, CU Boulder and NOAA Global Monitoring
Laboratory, Boulder, USA
46. Department of Climate, Meteorology and Atmospheric Sciences, University of Illinois, Urbana, IL 61821,
USA
47. University of East Anglia, Norwich, UK
48. Jiangsu Provincial Key Laboratory of Geographic Information Science and Technology, International
Institute for Earth System Science, Nanjing University, Nanjing, 210023, China
49. State Key Laboratory of Tibetan Plateau Earth System and Resource Environment, Institute of Tibetan
Plateau Research, Chinese Academy of Sciences, Beijing 100101, China
50. Institute of Carbon Neutrality, Sino-French Institute for Earth System Science, College of Urban and
Environmental Sciences, Peking University, Beijing 100871, China
51. Institute of Applied Energy (IAE), Minato-ku, Tokyo 105-0003, Japan
52. University of California, San Diego, Scripps Institution of Oceanography, La Jolla, CA 92093-0244, USA
53. Utrecht University, Faculty of Geosciences, Department IMEW, Copernicus Institute of Sustainable
Development, Heidelberglaan 2, P.O. Box 80115, 3508 TC, Utrecht, the Netherlands
54. Hawkesbury Institute for the Environment, Western Sydney University, Penrith, New South Wales,
Australia
55. NORCE Norwegian Research Centre, Jahnebakken 5, 5007 Bergen, Norway
56. LOCEAN/IPSL laboratory, Sorbonne Université, CNRS/IRD/MNHN, Paris, 75252, France
57. Jet Propulsion Laboratory, California Institute of Technology, Pasadena, CA, USA
58. California Institute of Technology, Pasadena, CA, USA
59. Earth System Division, National Institute for Environmental Studies, 16-2 Onogawa, Tsukuba, Ibaraki, 305-
8506 Japan
60. Sorbonne Université, Laboratoire d'Océanographie de Villefranche, Villefranche-sur-Mer, France



61. Department of Meteorology & National Centre for Atmospheric Science (NCAS), University of Reading,
Reading, UK
62. University of Alaska Fairbanks, College of Fisheries and Ocean Sciences, Fairbanks, AK, 99709, USA
63. Fisheries Research and Education Agency, 2-12-4 Fukuura, Kanazawa-Ku, Yokohama 236-8648, Japan
64. National Centre for Earth Observation, University of Edinburgh, EH9 3FF, UK
65. School of GeoSciences, University of Edinburgh, EH9 3FF, UK
66. NOAA Atlantic Oceanographic and Meteorological Laboratory (NOAA/AOML), 4301 Rickenbacker
Causeway, Miami, Florida 33149, USA
67. School of Atmospheric Sciences, Sun Yat-sen University, Zhuhai 519000, China
68. Princeton University, Department of Geosciences and Princeton Environmental Institute, Princeton, NJ,
USA
69. Max Planck Institute for Biogeochemistry, P.O. Box 600164, Hans-Knöll-Str. 10, 07745 Jena, Germany
70. Smith School of Enterprise and the Environment, University of Oxford, Oxford, UK
71. Canadian Centre for Climate Modelling and Analysis, Environment and Climate Change Canada, Victoria,
British Columbia, Canada
72. GEOMAR Helmholtz Centre for OCean Research Kiel, Wischhofstr. 1-3, 24148 Kiel, Germany
73. Institute for Climate and Environmental Physics, Bern, Switzerland
74. Research Center for Environmental Modeling and Application, Japan Agency for Marine-Earth Science and
Technology, Yokohama, Japan
75. Advanced Institute for Marine Ecosystem Change, Japan Agency for Marine-Earth Science and Technology,
Yokohama, Japan
76. Schiller Institute of Integrated Science and Society, Department of Earth and Environmental Sciences,
Boston College, Chestnut Hill, MA 02467, USA
77. Australian Antarctic Partnership Program, University of Tasmania, Hobart, Australia
78. Barcelona Supercomputing Center, Barcelona, Spain
79. JMA Meteorological Research Institute, Tsukuba, Ibaraki, Japan
80. Statistics Division, Food and Agriculture Organization of the United Nations, Via Terme di Caracalla, Rome
00153, Italy
81. Institute of Atmospheric Physics, Chinese Academy of Sciences, Beijing, China
82. Climate Change Science Institute and Environmental Sciences Division, Oak Ridge National Lab, Oak
Ridge, TN 37831, USA.
83. School of Ecology and Applied Meteorology, Nanjing University of Information Science and Technology,
Nanjing 210044, PR. China
84. Institute of Carbon Neutrality, College of Urban and Environmental Sciences, Peking University, Beijing
100091, China
85. School of Environmental Science and Engineering, Nanjing University of Information Science and
Technology (NUIST), Nanjing, 210044, China
86. Department of Atmospheric and Oceanic Science, University of Maryland, Maryland, USA





*Correspondence to*: Pierre Friedlingstein (p.friedlingstein@exeter.ac.uk)
**1.  Abstract**
Accurate assessment of anthropogenic carbon dioxide ($CO_2$) emissions and their redistribution among the
atmosphere, ocean, and terrestrial biosphere in a changing climate is critical to better understand the global
carbon cycle, support the development of climate policies, and project future climate change. Here we describe
and synthesise datasets and methodologies to quantify the five major components of the global carbon budget
and their uncertainties. Fossil $CO_2$ emissions ($E_{FOS}$) are based on energy statistics and cement production data,
while emissions from land-use change ($E_{LUC}$) are based on land-use and land-use change data and bookkeeping
models. Atmospheric $CO_2$ concentration is measured directly, and its growth rate ($G_{ATM}$) is computed from the
annual changes in concentration. The ocean $CO_2$ sink ($S_{OCEAN}$) is estimated with global ocean biogeochemistry
models and observation-based $f$$CO_2$-products. The terrestrial $CO_2$ sink ($S_{LAND}$) is estimated with dynamic
global vegetation models. Additional lines of evidence on land and ocean sinks are provided by atmospheric
inversions, atmospheric oxygen measurements and Earth System Models. The sum of all sources and sinks
results in the carbon budget imbalance ($B_{IM}$), a measure of imperfect data and incomplete understanding of the
contemporary carbon cycle. All uncertainties are reported as $\pm1\sigma$.
For the year 2023, $E_{FOS}$ increased by 1.3% relative to 2022, with fossil emissions at $10.1 \pm 0.5$ GtC yr$^{-1}$ ($10.3 \pm$
$0.5$ GtC yr$^{-1}$ when the cement carbonation sink is not included), $E_{LUC}$ was $1.0 \pm 0.7$ GtC yr$^{-1}$, for a total
anthropogenic $CO_2$ emission (including the cement carbonation sink) of $11.1 \pm 0.9$ GtC yr$^{-1}$ ($40.6 \pm 3.2$ GtCO$_2$
yr$^{-1}$). Also, for 2023, $G_{ATM}$ was $5.9 \pm 0.2$ GtC yr$^{-1}$ ($2.79 \pm 0.1$ ppm yr$^{-1}$), $S_{OCEAN}$ was $2.9 \pm 0.4$ GtC yr$^{-1}$ and
$S_{LAND}$ was $2.3 \pm 1.0$ GtC yr$^{-1}$, with a near zero $B_{IM}$ (-0.02 GtC yr$^{-1}$). The global atmospheric $CO_2$ concentration
averaged over 2023 reached $419.3 \pm 0.1$ ppm. Preliminary data for 2024, suggest an increase in $E_{FOS}$ relative to
2023 of +0.8% (-0.3% to 1.9%) globally, and atmospheric $CO_2$ concentration increased by 2.8 ppm reaching
422.5 ppm, 52% above pre-industrial level (around 278 ppm in 1750). Overall, the mean and trend in the
components of the global carbon budget are consistently estimated over the period 1959-2023, with a near-zero
overall budget imbalance, although discrepancies of up to around 1 GtC yr$^{-1}$ persist for the representation of
annual to semi-decadal variability in $CO_2$ fluxes. Comparison of estimates from multiple approaches and
observations shows: (1) a persistent large uncertainty in the estimate of land-use changes emissions, (2) a low
agreement between the different methods on the magnitude of the land $CO_2$ flux in the northern extra-tropics,
and (3) a discrepancy between the different methods on the mean ocean sink.
This living data update documents changes in methods and datasets applied to this most-recent global carbon
budget as well as evolving community understanding of the global carbon cycle. The data presented in this
work are available at https://doi.org/10.18160/GCP-2024 (Friedlingstein et al., 2024).



## 2. Executive Summary

**Global fossil $CO_2$ emissions (including cement carbonation) are expected to further increase in 2024 by 0.8%.** The 2023 emission increase was 0.14 GtC yr$^{-1}$ (0.5 GtCO$_2$ yr$^{-1}$) relative to 2022, bringing 2023 fossil CO$_2$ emissions to 10.1 ± 0.5 GtC yr$^{-1}$ (36.8 ± 1.8 GtCO$_2$ yr$^{-1}$). Preliminary estimates based on data available suggest fossil CO$_2$ emissions to increase further in 2024, by 0.8% relative to 2023 (-0.3% to 1.9%), bringing emissions to 10.2 GtC yr$^{-1}$ (37.4 GtCO$_2$ yr$^{-1}$).[1]

Emissions from coal, oil and gas in 2024 are expected to be slightly above their 2023 levels (by 0.2%, 0.9% and 2.4% respectively). Regionally, fossil emissions in 2024 are expected to decrease by 3.8% in the European Union reaching 0.7 GtC (2.4 GtCO$_2$), and by 0.6% in the United States (1.3 GtC, 4.9 GtCO$_2$). Emissions in China are expected to increase in 2024 by 0.2%, reaching 3.3 GtC, (12.0 GtCO$_2$). Fossil emissions are also expected to increase by 4.6% in India (0.9 GtC, 3.2 GtCO$_2$) and by 1.1% for the rest of the world (4.0 GtC, 14.5 GtCO$_2$) in 2024. Emissions from international aviation and shipping (IAS) are also expected to increase by 7.8% (0.3 GtC, 1.2 GtCO$_2$) in 2024.

**Fossil $CO_2$ emissions decreased significantly in 22 countries with significantly growing economies during the decade 2014-2023.** Altogether, these 22 countries contribute about 2.2 GtC yr$^{-1}$ (8.1 GtCO$_2$) fossil fuel CO$_2$ emissions over the last decade, representing about 23% of world CO$_2$ fossil emissions.

**Global $CO_2$ emissions from land-use, land-use change, and forestry (LULUCF) averaged 1.1 ± 0.7 GtC yr$^{-1}$ (4.1 ± 2.6 GtCO$_2$ yr$^{-1}$) for the 2014-2023 period with a similar preliminary projection for 2024 of 1.1 ± 0.7 GtC yr$^{-1}$ (4.2 ± 2.6 GtCO$_2$ yr$^{-1}$). Since the late-1990s, emissions from LULUCF show a statistically significant decrease at a rate of around 0.2 GtC per decade.** Emissions from deforestation, the main driver of global gross sources, remain high at around 1.7 GtC yr$^{-1}$ over the 2014-2023 period, highlighting the strong potential of halting deforestation for emissions reductions. Sequestration of 1.2 GtC yr$^{-1}$ through re-/afforestation and forestry offsets two third of the deforestation emissions. Further, smaller emissions are due to other land-use transitions and peat drainage and peat fire. The highest emitters during 2014-2023 in descending order were Brazil, Indonesia, and the Democratic Republic of the Congo, with these 3 countries contributing more than half of global land-use CO$_2$ emissions.

**Total anthropogenic emissions (fossil and LULUCF, including the carbonation sink) were 11.1 GtC yr$^{-1}$ (40.6 GtCO$_2$ yr-1) in 2023, with a marginally higher preliminary estimate of 11.3 GtC yr$^{-1}$ (41.6 GtCO$_2$ yr$^{-1}$) for 2024. Total anthropogenic emissions have been stable over the last decade (zero growth rate over the 2014-2023 period), much slower than over the previous decade (2004-2013) with an average growth rate of 2.0% yr$^{-1}$.**

**The remaining carbon budget for a 50% likelihood to limit global warming to 1.5°C, 1.7°C and 2°C above the 1850-1900 level has respectively been reduced to 65 GtC (235 GtCO$_2$), 160 GtC (585 GtCO$_2$) and 305**

---

[1] All 2024 growth rates use a leap year adjustment that corrects for the extra day in 2024.



**GtC (1110 GtCO₂) from the beginning of 2025, equivalent to around 6, 14 and 27 years, assuming 2024**
**emissions levels.**
**The concentration of $CO_2$ in the atmosphere is set to reach 422.5 ppm in 2024, 52% above pre-industrial**
**levels.** The atmospheric $CO_2$ growth was $5.2 \pm 0.02$ GtC yr$^{-1}$ (2.5 ppm) during the decade 2014-2023 (48% of
total $CO_2$ emissions) with a preliminary 2024 growth rate estimate of around 5.9 GtC (2.8 ppm).
**The ocean $CO_2$ sink has been stagnant since 2016 after rapid growth during 2002-2016, largely in**
**response to large inter-annual climate variability.** The ocean $CO_2$ sink was $2.9 \pm 0.4$ GtC yr$^{-1}$ during the
decade 2014-2023 (26% of total $CO_2$ emissions). A slightly higher value of 3.0 GtC yr$^{-1}$ is preliminarily
estimated for 2024, which marks an increase in the sink since 2023 due to the prevailing El Niño and neutral
conditions in 2024.
**The land $CO_2$ sink continued to increase during the 2014-2023 period primarily in response to increased**
**atmospheric $CO_2$, albeit with large interannual variability.** The land $CO_2$ sink was $3.2 \pm 0.9$ GtC yr$^{-1}$ during
the 2014-2023 decade (30% of total $CO_2$ emissions). The land sink in 2023 was $2.3 \pm 1$ GtC yr$^{-1}$, 1.6 GtC lower
than in 2022, and the lowest estimate since 2015. This reduced sink is primarily driven by a response of tropical
land ecosystems to the onset of the 2023-2024 El Niño event, combined with large wildfires in Canada in 2023.
The preliminary 2024 estimate is around 3.2 GtC yr$^{-1}$, similar to the decadal average, consistent with a land sink
emerging from the El Niño state.
**So far in 2024, global fire $CO_2$ emissions have been 11-32% higher than the 2014-2023 average due to**
**high fire activity in both North and South America, reaching 1.6-2.2 GtC during January-September.** In
Canada, emissions through September were 0.2-0.3 GtC yr$^{-1}$, down from 0.5-0.8 GtC yr$^{-1}$ in 2023 but still more
than twice the 2014-2023 average. In Brazil, fires through September emitted 0.2-0.3 GtC yr$^{-1}$, 91-118% above
the 2014-2023 average due to intense drought. These fire emissions estimates should not be directly compared
with the land use emissions or the land sink, because they represent a gross carbon flux to the atmosphere and
do not account for post-fire recovery or distinguish between natural, climate-driven, and land-use-related fires.






## 1    Introduction

The concentration of carbon dioxide ($CO_2$) in the atmosphere has increased from approximately 278 parts per
million (ppm) in 1750 (Gulev et al., 2021), the beginning of the Industrial Era, to $419.3 \pm 0.1$ ppm in 2023 (Lan
et al., 2024; Figure 1). The atmospheric $CO_2$ increase above pre-industrial levels was, initially, primarily caused
by the release of carbon to the atmosphere from deforestation and other land-use change activities (Canadell et
al., 2021). While emissions from fossil fuels started before the Industrial Era, they became the dominant source
of anthropogenic emissions to the atmosphere from around 1950 and their relative share has continued to
increase until present. Anthropogenic emissions occur on top of an active natural carbon cycle that circulates
carbon between the reservoirs of the atmosphere, ocean, and terrestrial biosphere on time scales from sub-daily
to millennial, while exchanges with geologic reservoirs occur on longer timescales (Archer et al., 2009).
The global carbon budget (GCB) presented here refers to the mean, variations, and trends in the perturbation of
$CO_2$ in the environment, referenced to the beginning of the Industrial Era (defined here as 1750). This paper
describes the components of the global carbon cycle over the historical period with a stronger focus on the
recent period (since 1958, onset of robust atmospheric $CO_2$ measurements), the last decade (2014-2023), the last
year (2023) and the current year (2024). Finally, it provides cumulative emissions from fossil fuels and land-use
change since the year 1750, and since the year 1850 (the reference year for historical simulations in IPCC AR6)
(Eyring et al., 2016).
We quantify the input of $CO_2$ to the atmosphere by emissions from human activities, the growth rate of
atmospheric $CO_2$ concentration, and the resulting changes in the storage of carbon in the land and ocean
reservoirs in response to increasing atmospheric $CO_2$ levels, climate change and variability, and other
anthropogenic and natural changes (Figure 2). An understanding of this perturbation budget over time and the
underlying variability and trends of the natural carbon cycle is necessary to understand the response of natural
sinks to changes in climate, $CO_2$ and land-use change drivers, and to quantify emissions compatible with a given
climate stabilisation target.
The components of the $CO_2$ budget that are reported annually in this paper include separate and independent
estimates for the $CO_2$ emissions from (1) fossil fuel combustion and oxidation from all energy and industrial
processes; also including cement production and carbonation ($E_{FOS}$; GtC yr$^{-1}$) and (2) the emissions resulting
from deliberate human activities on land, including those leading to land-use change ($E_{LUC}$; GtC yr$^{-1}$); and their
partitioning among (3) the growth rate of atmospheric $CO_2$ concentration ($G_{ATM}$; GtC yr$^{-1}$), and the uptake of
$CO_2$ (the '$CO_2$ sinks') in (4) the ocean ($S_{OCEAN}$; GtC yr$^{-1}$) and (5) on land ($S_{LAND}$; GtC yr$^{-1}$). The $CO_2$ sinks as
defined here conceptually include the response of the land (including inland waters and estuaries) and ocean
(including coastal and marginal seas) to elevated $CO_2$ and changes in climate and other environmental
conditions, although in practice not all processes are fully accounted for (see Section 2.10). Global emissions
and their partitioning among the atmosphere, ocean and land are in balance in the real world. Due to the
combination of imperfect spatial and/or temporal data coverage, errors in each estimate, and smaller terms not



included in our budget estimate (discussed in Section 2.10), the independent estimates (1) to (5) above do not
necessarily add up to zero. We hence estimate a budget imbalance ($B_{IM}$), which is a measure of the mismatch
between the estimated emissions and the estimated changes in the atmosphere, land and ocean, as follows:
$B_{IM} = E_{FOS} + E_{LUC} - (G_{ATM} + S_{OCEAN} + S_{LAND})$                     (1)
$G_{ATM}$ is usually reported in ppm yr$^{-1}$, which we convert to units of carbon mass per year, GtC yr$^{-1}$, using 1 ppm
= 2.124 GtC (Ballantyne et al., 2012; Table 1). Units of gigatonnes of $CO_2$ (or billion tonnes of $CO_2$) used in
policy are equal to 3.664 multiplied by the value in units of GtC.
We also assess a set of additional lines of evidence derived from global atmospheric inversion system results
(Section 2.7), observed changes in oxygen concentration (Section 2.8) and Earth System Models (ESMs)
simulations (Section 2.9), all of these methods closing the global carbon balance (zero $B_{IM}$).
We further quantify $E_{FOS}$ and $E_{LUC}$ by country, including both territorial and consumption-based accounting for
$E_{FOS}$ (see Section 2), and discuss missing terms from sources other than the combustion of fossil fuels (see
Section 2.10, Supplement S1 and S2). We also assess carbon dioxide removal (CDR) (see Sect. 2.2 and 2.3).
Land-based CDR is significant, but already accounted for in $E_{LUC}$ in equation (1) (Sect 3.2.2). Other CDR
methods, not based on vegetation, are currently several orders of magnitude smaller than the other components
of the budget (Sect. 3.3), hence these are not included in equation (1), or in the global carbon budget tables or
figures (with the exception of Figure 2 where CDR is shown primarily for illustrative purpose).
The global $CO_2$ budget has been assessed by the Intergovernmental Panel on Climate Change (IPCC) in all
assessment reports (Prentice et al., 2001; Schimel et al., 1995; Watson et al., 1990; Denman et al., 2007; Ciais et
al., 2013; Canadell et al., 2021), and by others (e.g. Ballantyne et al., 2012). The Global Carbon Project (GCP,
www.globalcarbonproject.org, last access: 28 October 2024) has coordinated this cooperative community effort
for the annual publication of global carbon budgets for the year 2005 (Raupach et al., 2007; including fossil
emissions only), year 2006 (Canadell et al., 2007), year 2007 (GCP, 2008), year 2008 (Le Quéré et al., 2009),
year 2009 (Friedlingstein et al., 2010), year 2010 (Peters et al., 2012a), year 2012 (Le Quéré et al., 2013; Peters
et al., 2013), year 2013 (Le Quéré et al., 2014), year 2014 (Le Quéré et al., 2015a; Friedlingstein et al., 2014),
year 2015 (Jackson et al., 2016; Le Quéré et al., 2015b), year 2016 (Le Quéré et al., 2016), year 2017 (Le Quéré
et al., 2018a; Peters et al., 2017a), year 2018 (Le Quéré et al., 2018b; Jackson et al., 2018), year 2019
(Friedlingstein et al., 2019; Jackson et al., 2019; Peters et al., 2020), year 2020 (Friedlingstein et al., 2020; Le
Quéré et al., 2021), year 2021 (Friedlingstein et al., 2022a; Jackson et al., 2022), year 2022 (Friedlingstein et al.,
2022b), and most recently the year 2023 (Friedlingstein et al., 2023). Each of these papers updated previous
estimates with the latest available information for the entire time series.
We adopt a range of ±1 standard deviation (σ) to report the uncertainties in our global estimates, representing a
likelihood of 68% that the true value will be within the provided range if the errors have a gaussian distribution,
and no bias is assumed. This choice reflects the difficulty of characterising the uncertainty in the $CO_2$ fluxes
between the atmosphere and the ocean and land reservoirs individually, particularly on an annual basis, as well



as the difficulty of updating the $CO_2$ emissions from land-use change. A likelihood of 68% provides an
indication of our current capability to quantify each term and its uncertainty given the available information.
The uncertainties reported here combine statistical analysis of the underlying data, assessments of uncertainties
in the generation of the datasets, and expert judgement of the likelihood of results lying outside this range. The
limitations of current information are discussed in the paper and have been examined in detail elsewhere
(Ballantyne et al., 2015; Zscheischler et al., 2017). We also use a qualitative assessment of confidence level to
characterise the annual estimates from each term based on the type, amount, quality, and consistency of the
different lines of evidence as defined by the IPCC (Stocker et al., 2013).
This paper provides a detailed description of the datasets and methodology used to compute the global carbon
budget estimates for the industrial period, from 1750 to 2024, and in more detail for the period since 1959. This
paper is updated every year using the format of 'living data' to keep a record of budget versions and the changes
in new data, revision of data, and changes in methodology that lead to changes in estimates of the carbon
budget. Additional materials associated with the release of each new version will be posted at the Global Carbon
Project (GCP) website (http://www.globalcarbonproject.org/carbonbudget, last access: 28 October 2024), with
fossil fuel emissions also available through the Global Carbon Atlas (http://www.globalcarbonatlas.org, last
access: 28 October 2024). All underlying data used to produce the budget can also be found at
https://globalcarbonbudget.org/ (last access: 28 October 2024). With this approach, we aim to provide the
highest transparency and traceability in the reporting of $CO_2$, the key driver of climate change.
**2    Methods**
Multiple organisations and research groups around the world generated the original measurements and data used
to complete the global carbon budget. The effort presented here is thus mainly one of synthesis, where results
from individual groups are collated, analysed, and evaluated for consistency. We facilitate access to original
data with the understanding that primary datasets will be referenced in future work (see Table 2 for how to cite
the datasets, and Section on data availability). Descriptions of the measurements, models, and methodologies
follow below, with more detailed descriptions of each component provided as Supplementary Information (S1 to
S5).
This is the 19[th] version of the global carbon budget and the 13[th] revised version in the format of a living data
update in Earth System Science Data. It builds on the latest published global carbon budget of Friedlingstein et
al. (2023). The main changes this year are: the inclusion of (1) data to year 2023 and a projection for the global
carbon budget for year 2024; and (2) an estimate of the 2024 projection of fossil emissions from Carbon
Monitor. Other methodological differences between recent annual carbon budgets (2020 to 2024) are
summarised in Table 3 and previous changes since 2006 are provided in Table S9.



### 2.1 Fossil $CO_2$ emissions ($E_{FOS}$)

#### 2.1.1 Historical period 1850-2023

The estimates of global and national fossil $CO_2$ emissions ($E_{FOS}$) include the oxidation of fossil fuels through both combustion (e.g., transport, heating) and chemical oxidation (e.g. carbon anode decomposition in aluminium refining) activities, and the decomposition of carbonates in industrial processes (e.g. the production of cement). We also include $CO_2$ uptake from the cement carbonation process. Several emissions sources are not estimated or not fully covered: coverage of emissions from lime production are not global, and decomposition of carbonates in glass and ceramic production are included only for the "Annex 1" countries of the United Nations Framework Convention on Climate Change (UNFCCC) for lack of activity data. These omissions are considered to be minor. Short-cycle carbon emissions - for example from combustion of biomass - are not included here but are accounted for in the $CO_2$ emissions from land use (see Section 2.2).

Our estimates of fossil $CO_2$ emissions rely on data collection by many other parties. Our goal is to produce the best estimate of this flux, and we therefore use a prioritisation framework to combine data from different sources that have used different methods, while being careful to avoid double counting and undercounting of emissions sources. The CDIAC-FF emissions dataset, derived largely from UN energy data, forms the foundation, and we extend emissions to 2023 using energy growth rates reported by the Energy Institute (a dataset formerly produced by BP). We then proceed to replace estimates using data from what we consider to be superior sources, for example Annex 1 countries' official submissions to the UNFCCC. All data points are potentially subject to revision, not just the latest year. For full details see Andrew and Peters (2024).

Other estimates of global fossil $CO_2$ emissions exist, and these are compared by Andrew (2020a). The most common reason for differences in estimates of global fossil $CO_2$ emissions is a difference in which emissions sources are included in the datasets. Datasets such as those published by the Energy Institute, the US Energy Information Administration, and the International Energy Agency's '$CO_2$ emissions from fuel combustion' are all generally limited to emissions from combustion of fossil fuels. In contrast, datasets such as PRIMAP-hist, CEDS, EDGAR, and GCP's dataset aim to include all sources of fossil $CO_2$ emissions. See Andrew (2020a) for detailed comparisons and discussion.

Cement absorbs $CO_2$ from the atmosphere over its lifetime, a process known as 'cement carbonation'. We estimate this $CO_2$ sink, from 1931 onwards, as the average of two studies in the literature (Cao et al., 2020; Guo et al., 2021). Both studies use the same model, developed by Xi et al. (2016), with different parameterisations and input data, with the estimate of Guo and colleagues being a revision of Xi et al. (2016). The trends of the two studies are very similar. Since carbonation is a function of both current and previous cement production, we extend these estimates to 2023 by using the growth rate derived from the smoothed cement emissions (10-year smoothing) fitted to the carbonation data. In the present budget, we always include the cement carbonation carbon sink in the fossil $CO_2$ emission component ($E_{FOS}$).





We use the Kaya Identity for a simple decomposition of $CO_2$ emissions into the key drivers (Raupach et al.,
2007). While there are variations (Peters et al., 2017a), we focus here on a decomposition of $CO_2$ emissions into
population, GDP per person, energy use per GDP, and $CO_2$ emissions per energy. Multiplying these individual
components together returns the $CO_2$ emissions. Using the decomposition, it is possible to attribute the change
in $CO_2$ emissions to the change in each of the drivers. This method gives a first-order understanding of what
causes $CO_2$ emissions to change each year.

### 2.1.2    2024 projection

We provide a projection of global fossil $CO_2$ emissions in 2024 by combining separate projections for China,
USA, EU, India, and for all other countries combined. The methods are different for each of these. For China we
combine monthly fossil fuel production data from the National Bureau of Statistics and trade data from the
Customs Administration, giving us partial data for the growth rates to date of natural gas, petroleum, and
cement, and of the apparent consumption itself for raw coal. We then use a regression model to project full-year
emissions based on historical observations. For the USA our projection is taken directly from the Energy
Information Administration's (EIA) Short-Term Energy Outlook (EIA, 2024), combined with the year-to-date
growth rate of cement clinker production. For the EU we use monthly energy data from Eurostat to derive
estimates of monthly $CO_2$ emissions through July, with coal emissions extended through September using a
statistical relationship with reported electricity generation from coal and other factors. For natural gas we use
Holt-Winters to project the last four months of the year. EU emissions from oil are derived using the EIA's
projection of oil consumption for Europe. EU cement emissions are based on available year-to-date data from
three of the largest producers, Germany, Poland, and Spain. India's projected emissions are derived from
estimates through August (July for coal) using the methods of Andrew (2020b) and extrapolated assuming
seasonal patterns from before 2019. Emissions from international transportation (bunkers) are estimated
separately for aviation and shipping. Changes in aviation emissions are derived primarily from OECD monthly
estimates, extrapolated using the growth rates of global flight miles from Airportia, and then the final months
are projected assuming normal patterns from previous years. Changes in shipping emissions are derived from
OECD monthly estimates for global shipping. Emissions for the rest of the world are derived for coal and
cement using projected growth in economic production from the IMF (2023) combined with extrapolated
changes in emissions intensity of economic production; for oil using a global constraint from EIA; and for
natural gas using a global constraint from IEA. More details on the $E_{FOS}$ methodology and its 2024 projection
can be found in Supplement S.1.
For the first time this year, we cross check our 2024 projection with a 2024 projection from Carbon Monitor.
Carbon Monitor is an open access dataset (https://carbonmonitor.org/) of daily emissions constructed using
hourly to daily proxy data (e.g., electricity consumption, travel patterns, etc) instead of energy use data.
Available Carbon Monitor estimated emissions from January to August are combined to a new projection for
September to December to give a full year 2024 estimate. The September to December projections are estimated
by leveraging seasonal patterns from 2019-2023 daily $CO_2$ emission data from Carbon Monitor. A regression
model is applied separately for individual countries to obtain their respective 4-month forecast. First, the



seasonality component for each month is assessed based on daily average emissions from 2019 to 2023,
excluding 2020 due to the COVID-19 pandemic. Then, a linear regression model is constructed using the
calculated seasonal components and the daily average emissions for the months from January to August 2024.
The resulting model is used to project carbon emissions for the remaining months of 2024. The uncertainty
range is calculated by using historical monthly variance of seasonal components.
**2.2    $CO_2$ emissions from land-use, land-use change and forestry ($E_{LUC}$)**
**2.2.1    Historical period 1850-2023**
The net $CO_2$ flux from land-use, land-use change and forestry ($E_{LUC}$, called land-use change emissions in the
rest of the text) includes $CO_2$ fluxes from deforestation, afforestation, logging and forest degradation (including
harvest activity), shifting cultivation (cycle of cutting forest for agriculture, then abandoning), regrowth of
forests (following wood harvest or agriculture abandonment), peat burning, and peat drainage.
Four bookkeeping approaches (updated estimates each of BLUE (Hansis et al., 2015), OSCAR (Gasser et al.,
2020), and H&C2023 (Houghton and Castanho, 2023), and new estimates of LUCE (Qin et al. 2024) were used
to quantify gross emissions and gross removals and the resulting net $E_{LUC}$. Emissions from peat burning and peat
drainage are added from external datasets, peat drainage being averaged from three spatially explicit
independent datasets (see Supplement S.2.1). Uncertainty estimates were derived from the Dynamic Global
Vegetation Models (DGVMs) ensemble for the time period prior to 1960, and using for the recent decades an
uncertainty range of $\pm 0.7$ GtC yr$^{-1}$, which is a semi-quantitative measure for annual and decadal emissions and
reflects our best value judgement that there is at least 68% chance ($\pm 1\sigma$) that the true land-use change emission
lies within the given range, for the range of processes considered here.
The GCB $E_{LUC}$ estimates follow the $CO_2$ flux definition of global carbon cycle models and differ from IPCC
definitions adopted in National GHG Inventories (NGHGI) for reporting under the UNFCCC. The latter
typically include terrestrial fluxes occurring on all land that countries define as managed, following the IPCC
managed land proxy approach (Grassi et al., 2018). This partly includes fluxes due to environmental change
(e.g. atmospheric $CO_2$ increase), which are part of $S_{LAND}$ in our definition. As a result, global emission estimates
are smaller for NGHGI than for the global carbon budget definition (Grassi et al., 2023). The same is the case
for the Food Agriculture Organization (FAO) estimates of carbon fluxes on forest land, which include both
anthropogenic and natural fluxes on managed land (Tubiello et al., 2021). We translate the GCB and NGHGI
definitions to each other, to provide a comparison of the anthropogenic carbon budget as reported in GCB to the
official country reporting to the UNFCCC convention. We further compare these estimates with the net
atmosphere-to-land flux from atmospheric inversion systems (see Section 2.7), averaged over managed land
only.
$E_{LUC}$ contains a range of fluxes that are related to Carbon Dioxide Removal (CDR). CDR is defined as the set of
anthropogenic activities that remove $CO_2$ from the atmosphere, additional to the Earth's natural processes, and
store it in durable form, such as in forest biomass and soils, long-lived products, or in geological or ocean



reservoirs. Here, we quantify vegetation-based CDR that is implicitly or explicitly captured by land-use fluxes
(CDR not based on vegetation is discussed in Section 2.3; IPCC, 2023). We quantify re/afforestation from the
four bookkeeping estimates by separating forest regrowth in shifting cultivation cycles from permanent
increases in forest cover (see Supplement S.2.1). The latter count as CDR, but it should be noted that the
permanence of the storage under climate risks such as fire is increasingly questioned. Other CDR activities
contained in $E_{LUC}$ include the transfer of carbon to harvested wood products (HWP), bioenergy with carbon
capture and storage (BECCS); and biochar production. Note that the different bookkeeping models represent
HWP with varying details concerning product usage and their lifetimes. Bookkeeping and TRENDY models
currently only represent BECCS and biochar with regard to the $CO_2$ removal through photosynthesis, but do not
account for the durable storage. HWP, BECCS, and biochar are typically counted as CDR when the transfer to
the durable storage site occurs and not when the $CO_2$ is removed from the atmosphere, which complicates a
direct comparison to the GCB approach to quantify annual fluxes to and from the atmosphere. Estimates for
CDR through HWP, BECCS, and biochar are thus not indicated in this budget, but can be found elsewhere (see
Section 3.2.2).

**2.2.2    2024 Projection**

We project the 2024 land-use emissions for BLUE, H&C2023, OSCAR, and LUCE based on their $E_{LUC}$
estimates for 2023 and adding the change in carbon emissions from peat fires and tropical deforestation and
degradation fires (2024 emissions relative to 2023 emissions) estimated using active fire data (MCD14ML;
Giglio et al., 2016). Peat drainage is assumed to be unaltered as it has low interannual variability. More details
on the $E_{LUC}$ methodology can be found in Supplement S.2.

**2.3    Carbon Dioxide Removal (CDR) not based on vegetation**

While some CDR involves $CO_2$ fluxes via land-use and is included in $E_{LUC}$, (such as afforestation, biochar,
HWP, and BECCS) other CDR occurs through fluxes of $CO_2$ directly from the air to the geosphere. The
majority of this derives from enhanced weathering through the application of crushed rock to soils, with a
smaller contribution from Direct Air Carbon Capture and Storage (DACCS). We use data from the State of
CDR Report (Smith et al., 2024), which compiles and harmonises reported removal rates from a combination of
existing databases, surveys and novel research. Currently there are no internationally agreed methods for
reporting these types of CDR, meaning estimates are based on self-disclosure by projects following their own
protocols. As such, the fractional uncertainty on these numbers should be viewed as substantial, and they are
liable to change in future years as protocols are harmonised and improved.

**2.4    Growth rate in atmospheric $CO_2$ concentration ($G_{ATM}$)**

**2.4.1    Historical period 1850-2023**

The rate of growth of the atmospheric $CO_2$ concentration is provided for years 1959-2023 by the US National
Oceanic and Atmospheric Administration Global Monitoring Laboratory (NOAA/GML; Lan et al., 2024),



which includes recent revisions to the calibration scale of atmospheric $CO_2$ measurements (WMO-CO2-X2019;
Hall et al., 2021). For the 1959-1979 period, the global growth rate is based on measurements of atmospheric
$CO_2$ concentration averaged from the Mauna Loa and South Pole stations, as observed by the $CO_2$ Program at
Scripps Institution of Oceanography (Keeling et al., 1976). For the 1980-2021 time period, the global growth
rate is based on the average of multiple stations selected from the marine boundary layer sites with well-mixed
background air (Ballantyne et al., 2012), after fitting a smooth curve through the data for each station as a
function of time, and averaging by latitude band (Masarie and Tans, 1995). The annual growth rate is estimated
by Lan et al. (2024) from atmospheric $CO_2$ concentration by taking the average of the most recent December-
January months corrected for the average seasonal cycle and subtracting this same average one year earlier. The
growth rate in units of ppm yr$^{-1}$ is converted to units of GtC yr$^{-1}$ by multiplying by a factor of 2.124 GtC per
ppm, assuming instantaneous mixing of $CO_2$ throughout the atmosphere (Ballantyne et al., 2012; Table 1).
The uncertainty around the atmospheric growth rate is due to four main factors. First, the long-term
reproducibility of reference gas standards (around 0.03 ppm for 1σ from the 1980s; Lan et al., 2024). Second,
small unexplained systematic analytical errors that may have a duration of several months to two years come
and go. They have been simulated by randomising both the duration and the magnitude (determined from the
existing evidence) in a Monte Carlo procedure. Third, the network composition of the marine boundary layer
with some sites coming or going, gaps in the time series at each site, etc (Lan et al., 2024). The latter uncertainty
was estimated by NOAA/GML with a Monte Carlo method by constructing 100 "alternative" networks (Masarie
and Tans, 1995; NOAA/GML, 2019). The second and third uncertainties, summed in quadrature, add up to
0.085 ppm on average (Lan et al., 2024). Fourth, the uncertainty associated with using the average $CO_2$
concentration from a surface network to approximate the true atmospheric average $CO_2$ concentration (mass-
weighted, in 3 dimensions) as needed to assess the total atmospheric $CO_2$ burden. In reality, $CO_2$ variations
measured at the stations will not exactly track changes in total atmospheric burden, with offsets in magnitude
and phasing due to vertical and horizontal mixing. This effect must be very small on decadal and longer time
scales, when the atmosphere can be considered well mixed. The CO2 increase in the stratosphere lags the
increase (meaning lower concentrations) that we observe in the marine boundary layer, while the continental
boundary layer (where most of the emissions take place) leads the marine boundary layer with higher
concentrations. These effects nearly cancel each other. In addition, the growth rate is nearly the same
everywhere (Ballantyne et al., 2012). We therefore maintain an uncertainty around the annual growth rate based
on the multiple stations dataset ranges between 0.11 and 0.72 GtC yr$^{-1}$, with a mean of 0.61 GtC yr$^{-1}$ for 1959-
1979 and 0.17 GtC yr$^{-1}$ for 1980-2023, when a larger set of stations were available as provided by Lan et al.
(2024). We estimate the uncertainty of the decadal averaged growth rate after 1980 at 0.02 GtC yr$^{-1}$ based on the
calibration and the annual growth rate uncertainty but stretched over a 10-year interval. For years prior to 1980,
we estimate the decadal averaged uncertainty to be 0.07 GtC yr$^{-1}$ based on a factor proportional to the annual
uncertainty prior and after 1980 (0.02 * [0.61/0.17] GtC yr$^{-1}$).
We assign a high confidence to the annual estimates of $G_{ATM}$ because they are based on direct measurements
from multiple and consistent instruments and stations distributed around the world (Ballantyne et al., 2012; Hall
et al., 2021).





To estimate the total carbon accumulated in the atmosphere since 1750 or 1850, we use an atmospheric $CO_2$
concentration of $278.3 \pm 3$ ppm or $285.1 \pm 3$ ppm, respectively (Gulev et al., 2021). For the construction of the
cumulative budget shown in Figure 3, we use the fitted estimates of $CO_2$ concentration from Joos and Spahni
(2008) to estimate the annual atmospheric growth rate using the conversion factors shown in Table 1. The
uncertainty of $\pm 3$ ppm (converted to $\pm 1\sigma$) is taken directly from the IPCC's AR5 assessment (Ciais et al., 2013).
Typical uncertainties in the growth rate in atmospheric $CO_2$ concentration from ice core data are equivalent to
$\pm 0.1$-$0.15$ GtC yr$^{-1}$ as evaluated from the Law Dome data (Etheridge et al., 1996) for individual 20-year intervals
over the period from 1850 to 1960 (Bruno and Joos, 1997).
**2.4.2    2024 projection**
We provide an assessment of $G_{ATM}$ for 2024 as the average of two methods. The GCB regression method
models monthly global-average atmospheric $CO_2$ concentrations and derives the increment and annual average
from these. The model uses lagged observations of concentration (Lan et al., 2024): both a 12-month lag, and
the lowest lag that will allow model prediction to produce an estimate for the following January, recalling that
the $G_{ATM}$ increment is derived from December/January pairs. The largest driver of interannual changes is the
ENSO signal (Betts et al., 2016), so the monthly ENSO 3.4 index (Huang et al., 2023) is included in the model.
Given the natural lag between sea-surface temperatures and effects on the biosphere, and in turn effects on
globally mixed atmospheric $CO_2$ concentration, a lagged ENSO index is used, and we use both a 5-month and a
6-month lag. The combination of the two lagged ENSO values helps reduce possible effects of noise in a single
month. To help characterise the seasonal variation, we add month as a categorical variable. Finally, we flag the
period affected by the Pinatubo eruption (August 1991 - November 1993) as a categorical variable. Note that
while emissions of $CO_2$ are the largest driver of the trend in atmospheric $CO_2$ concentration, our goal here is to
predict divergence from that trend. Because changes in emissions from year to year are relatively minor in
comparison to total emissions, this has little effect on the variation of concentration from the trend line. Even the
relatively large drop in emissions in 2020 due to the COVID-19 pandemic does not cause any problems for the
model.
We also use the multi-model mean and uncertainty of the 2024 $G_{ATM}$ estimated by the ESMs prediction system
(see Section 2.9). We then take the average of the GCB regression and ESMs $G_{ATM}$ estimates, with their
respective uncertainty combined quadratically.
Similarly, the projection of the 2024 global average $CO_2$ concentration (in ppm), is calculated as the average of
the estimates from the two methods. For the GCB regression method, it is the annual average of global
concentration over the 12 months of 2024; for the ESMs, it is the observed global average $CO_2$ concentration for
2023 plus the annual increase in 2024 of the global average $CO_2$ concentration predicted by the ESMs multi-
model mean.



### 2.5    Ocean CO₂ sink

#### 2.5.1    Historical period 1850-2023

The reported estimate of the global ocean anthropogenic $CO_2$ sink $S_{OCEAN}$ is derived as the average of two estimates. The first estimate is derived as the mean over an ensemble of ten global ocean biogeochemistry models (GOBMs, Table 4 and Table S2). The second estimate is obtained as the mean over an ensemble of eight surface ocean $fCO_2$-observation-based data-products (Table 4 and Table S3). A ninth $fCO_2$-product (UExP-FFN-U) is shown but is not included in the ensemble average as it differs from the other products by adjusting the flux to a cool, salty ocean surface skin. In previous editions of the GCB, this product was following the Watson et al. (2020) method but has been updated following the method of Dong et al. (2022, see Supplement S.3.1 for a discussion). The GOBMs simulate both the natural and anthropogenic $CO_2$ cycles in the ocean. They constrain the anthropogenic air-sea $CO_2$ flux (the dominant component of $S_{OCEAN}$) by the transport of carbon into the ocean interior, which is also the controlling factor of present-day ocean carbon uptake in the real world. They cover the full globe and all seasons and were evaluated against surface ocean carbon observations, suggesting they are suitable to estimate the annual ocean carbon sink (Hauck et al., 2020). The $fCO_2$-products are tightly linked to observations of $fCO_2$ (fugacity of $CO_2$, which equals $pCO_2$ corrected for the non-ideal behaviour of the gas; Pfeil et al., 2013), which carry imprints of temporal and spatial variability, but are also sensitive to uncertainties in gas-exchange parameterizations and data-sparsity (Fay et al., 2021, Gloege et al., 2021, Hauck et al., 2023a). Their asset is the assessment of the mean spatial pattern of variability and its seasonality (Hauck et al., 2020, Gloege et al. 2021, Hauck et al., 2023a). To benchmark trends derived from the $fCO_2$-products, we additionally performed a model subsampling exercise following Hauck et al. (2023a, see section S3). In addition, two diagnostic ocean models are used to estimate $S_{OCEAN}$ over the industrial era (1781-1958).

The global $fCO_2$-based flux estimates were adjusted to remove the pre-industrial ocean source of $CO_2$ to the atmosphere of $0.65 \pm 0.3$ GtC yr$^{-1}$ from river input to the ocean (Regnier et al., 2022), to satisfy our definition of $S_{OCEAN}$ (Hauck et al., 2020). The river flux adjustment was distributed over the latitudinal bands using the regional distribution of Lacroix et al. (2020; North: 0.14 GtC yr$^{-1}$, Tropics: 0.42 GtC yr$^{-1}$, South: 0.09 GtC yr$^{-1}$). Acknowledging that this distribution is based on only one model, the advantage is that a gridded field is available, and the river flux adjustment can be calculated for the three latitudinal bands and the RECCAP regions (REgional Carbon Cycle Assessment and Processes (RECCAP2; Ciais et al., 2020, Poulter et al., 2022, DeVries et al., 2023). This dataset suggests that more of the riverine outgassing is located in the tropics than in the Southern Ocean and is thus opposed to the previously used dataset of Aumont et al. (2001). Accordingly, the regional distribution is associated with a major uncertainty in addition to the large uncertainty around the global estimate (Crisp et al., 2022; Gruber et al., 2023). Anthropogenic perturbations of river carbon and nutrient transport to the ocean are not considered (see Section 2.10 and Supplement S.6.3).

We derive $S_{OCEAN}$ from GOBMs by using a simulation (sim A) with historical forcing of climate and atmospheric $CO_2$ from GCB (Section 2.4), accounting for model biases and drift from a control simulation (sim B) with constant atmospheric $CO_2$ and normal year climate forcing. A third simulation (sim C) with historical



atmospheric $CO_2$ increase and normal year climate forcing is used to attribute the ocean sink to $CO_2$ (sim C
minus sim B) and climate (sim A minus sim C) effects. A fourth simulation (sim D; historical climate forcing
and constant atmospheric $CO_2$) is used to compare the change in anthropogenic carbon inventory in the interior
ocean (sim A minus sim D) to the observational estimate of Gruber et al. (2019) with the same flux components
(steady state and non-steady state anthropogenic carbon flux). The $f$CO$_2$-products are adjusted with respect to
their original publications to represent the full ice-free ocean area, including coastal zones and marginal seas,
when the area coverage is below 99%. This is done by either area filling following Fay et al. (2021) or a simple
scaling approach. GOBMs and $f$CO$_2$-products fall within the observational constraints over the 1990s ($2.2 \pm 0.7$
GtC yr$^{-1}$, Ciais et al., 2013) before and after applying adjustments.
S$_{OCEAN}$ is calculated as the average of the GOBM ensemble mean and the $f$CO$_2$-product ensemble mean from
1990 onwards. Prior to 1990, it is calculated as the GOBM ensemble mean plus half of the offset between
GOBMs and $f$CO$_2$-products ensemble means over 1990-2001.
We assign an uncertainty of $\pm 0.4$ GtC yr$^{-1}$ to the ocean sink based on a combination of random (ensemble
standard deviation) and systematic uncertainties (GOBMs bias in anthropogenic carbon accumulation,
previously reported uncertainties in $f$CO$_2$-products; see Supplement S.3.4). While this approach is consistent
within the GCB, an independent uncertainty assessment of the $f$CO$_2$-products alone suggests a somewhat larger
uncertainty of up to 0.7 GtC yr$^{-1}$ (Ford et al. 2024, accepted). We assess a medium confidence level to the
annual ocean $CO_2$ sink and its uncertainty because it is based on multiple lines of evidence, it is consistent with
ocean interior carbon estimates (Gruber et al., 2019, see Section 3.6.5) and the interannual variability in the
GOBMs and data-based estimates is largely consistent and can be explained by climate variability. We refrain
from assigning a high confidence because of the deviation between the GOBM and $f$CO$_2$-product trends
between around 2002 and 2020. More details on the S$_{OCEAN}$ methodology can be found in Supplement S.3.
**2.5.2    2024 Projection**
The ocean $CO_2$ sink forecast for the year 2024 is based on the annual historical time-series and our estimated
2024 atmospheric $CO_2$ concentration (Lan et al 2024), the historical and our estimated 2024 annual global fossil
fuel emissions from this year's carbon budget, and the spring (March, April, May) Oceanic Niño Index (ONI)
(NCEP, 2024). Using a non-linear regression approach, i.e., a feed-forward neural network, atmospheric $CO_2$,
ONI, and the fossil fuel emissions are used as training data to best match the annual ocean $CO_2$ sink (i.e.
combined S$_{OCEAN}$ estimate from GOBMs and data products) from 1959 through 2023 from this year's carbon
budget. Using this relationship, the 2024 S$_{OCEAN}$ can then be estimated from the projected 2024 input data using
the non-linear relationship established during the network training. To avoid overfitting, the neural network was
trained with a variable number of hidden neurons (varying between 2-5) and 20% of the randomly selected
training data were withheld for independent internal testing. Based on the best output performance (tested using
the 20% withheld input data), the best performing number of neurons was selected. In a second step, we trained
the network 10 times using the best number of neurons identified in step 1 and different sets of randomly
selected training data. The mean of the 10 trainings is considered our best forecast, whereas the standard





deviation of the 10 ensembles provides a first order estimate of the forecast uncertainty. This uncertainty is then
combined with the $S_{OCEAN}$ uncertainty (0.4 GtC yr$^{-1}$) to estimate the overall uncertainty of the 2024 projection.
As an additional line of evidence, we also assess the 2024 atmosphere-ocean carbon flux from the ESM
prediction system (see Section 2.9).

### 2.6    Land CO₂ sink


#### 2.6.1    Historical Period 1850-2023

The terrestrial land sink ($S_{LAND}$) is thought to be due to the combined effects of rising atmospheric $CO_2$,
increasing N inputs, and climate change, on plant growth and terrestrial carbon storage. $S_{LAND}$ does not include
land sinks directly resulting from land-use and land-use change (e.g., regrowth of vegetation) as these are part of
the land-use flux ($E_{LUC}$), although system boundaries make it difficult to attribute exactly $CO_2$ fluxes on land
between $S_{LAND}$ and $E_{LUC}$ (Erb et al., 2013).
$S_{LAND}$ is estimated from the multi-model mean of 20 DGVMs (Table 4 and Table S1). DGVMs simulations
include all climate variability and $CO_2$ effects over land. In addition to the carbon cycle represented in all
DGVMs, 14 models also account for the nitrogen cycle and hence can include the effect of N inputs on $S_{LAND}$.
The DGVMs estimate of $S_{LAND}$ does not include the export of carbon to aquatic systems or its historical
perturbation, which is discussed in Supplement S.6.3. DGVMs need to meet several criteria to be included in
this assessment. In addition, we use the International Land Model Benchmarking system (ILAMB; Collier et al.,
2018) for the DGVMs evaluation (see Supplement S.4.2), with an additional comparison of DGVMs with a
data-informed, Bayesian model-data fusion framework (CARDAMOM) (Bloom and Williams, 2015; Bloom et
al., 2016). The uncertainty on $S_{LAND}$ is taken from the DGVMs standard deviation. More details on the $S_{LAND}$
methodology can be found in Supplement S.4.

#### 2.6.2    2024 Projection

Like for the ocean forecast, the land CO₂ sink ($S_{LAND}$) forecast for the year 2024 is based on the annual
historical (Lan et al., 2024) and our estimated 2024 atmospheric $CO_2$ concentration, historical and our estimated
2024 annual global fossil fuel emissions from this year's carbon budget, and the summer (June, July, August)
ONI (NCEP, 2024). All training data are again used to best match $S_{LAND}$ from 1959 through 2023 from this
year's carbon budget using a feed-forward neural network. To avoid overfitting, the neural network was trained
with a variable number of hidden neurons (varying between 2-15), larger than for $S_{OCEAN}$ prediction due to the
stronger land carbon interannual variability. As done for $S_{OCEAN}$, a pre-training selects the optimal number of
hidden neurons based on 20% withheld input data, and in a second step, an ensemble of 10 forecasts is produced
to provide the mean forecast plus uncertainty. This uncertainty is then combined with the $S_{LAND}$ uncertainty for
2023 (1.0 GtC yr$^{-1}$) to estimate the overall uncertainty of the 2024 projection.




### 2.7 Atmospheric inversion estimate

The world-wide network of in-situ atmospheric measurements and satellite derived atmospheric $CO_2$ column (x$CO_2$) observations put a strong constraint on changes in the atmospheric abundance of $CO_2$. This is true globally (hence our large confidence in $G_{ATM}$), but also in regions with sufficient observational density found mostly in the extra-tropics. This allows atmospheric inversion methods to constrain the magnitude and location of the combined total surface $CO_2$ fluxes from all sources, including fossil and land-use change emissions and land and ocean $CO_2$ fluxes. The inversions assume $E_{FOS}$ to be well known, and they solve for the spatial and temporal distribution of land and ocean fluxes from the residual gradients of $CO_2$ between stations that are not explained by fossil fuel emissions. By design, such systems thus close the carbon balance ($B_{IM} = 0$) and thus provide an additional perspective on the independent estimates of the ocean and land fluxes.

This year's release includes fourteen inversion systems that are described in Table S4. Each system is rooted in Bayesian inversion principles but uses different methodologies. These differences concern the selection of atmospheric $CO_2$ data or x$CO_2$, and the choice of a-priori fluxes to refine. They also differ in spatial and temporal resolution, assumed correlation structures, and mathematical approach of the models (see references in Table S4 for details). Importantly, the systems use a variety of transport models, which was demonstrated to be a driving factor behind differences in atmospheric inversion-based flux estimates, and specifically their distribution across latitudinal bands (Gaubert et al., 2019; Schuh et al., 2019). Eight inversion systems used surface observations from the global measurement network (Schuldt et al., 2023, 2024). Six inversion systems (CAMS-FT24r1, CMS-flux, GONGGA, COLA, GCASv2, NTFVAR) used satellite x$CO_2$ retrievals from GOSAT and/or OCO-2, scaled to the WMO 2019 calibration scale, of which three inversions this year (CMS-Flux, COLA, NTFVAR) used these xCO2 datasets in addition to the in-situ observational $CO_2$ mole fraction records.

The original products delivered by the inverse modellers were modified to facilitate the comparison to the other elements of the budget, specifically on two accounts: (1) global total fossil fuel emissions including cement carbonation $CO_2$ uptake, and (2) riverine $CO_2$ transport. We note that with these adjustments the inverse results no longer represent the net atmosphere-surface exchange over land/ocean areas as sensed by atmospheric observations. Instead, for land, they become the net uptake of $CO_2$ by vegetation and soils that is not exported by fluvial systems, similar to the DGVMs estimates. For oceans, they become the net uptake of anthropogenic $CO_2$, similar to the GOBMs estimates.

The inversion systems prescribe global fossil fuel emissions based on e.g. the GCP's Gridded Fossil Emissions Dataset versions 2024.0 (GCP-GridFED; Jones et al., 2024a), which are updates to GCP-GridFEDv2021 presented by Jones et al. (2021b). GCP-GridFEDv2024.0 scales gridded estimates of $CO_2$ emissions from EDGARv4.3.2 (Janssens-Maenhout et al., 2019) within national territories to match national emissions estimates provided by the GCB for the years 1959-2023, which were compiled following the methodology described in Section 2.1. Small differences between the systems due to for instance regridding to the transport model resolution, or use of different fossil fuel emissions than GCP-GridFEDv2024.0, are adjusted in the



latitudinal partitioning we present, to ensure agreement with the estimate of $E_{FOS}$ in this budget. We also note
that the ocean fluxes used as prior by 8 out of 14 inversions are part of the suite of the ocean process model or
$fCO_2$-products listed in Section 2.5. Although these fluxes are further adjusted by the atmospheric inversions
(except for Jena CarboScope), it makes the inversion estimates of the ocean fluxes not completely independent
of $S_{OCEAN}$ assessed here.
To facilitate comparisons to the independent $S_{OCEAN}$ and $S_{LAND}$, we used the same adjustments for transport and
outgassing of carbon transported from land to ocean, as done for the observation-based estimates of $S_{OCEAN}$ (see
Supplement S.3).
The atmospheric inversions are evaluated using vertical profiles of atmospheric $CO_2$ concentrations (Figure S5).
More than 30 aircraft programs over the globe, either regular programs or repeated surveys over at least 9
months (except for SH programs), have been used to assess system performance (with space-time observational
coverage sparse in the SH and tropics, and denser in NH mid-latitudes; Table S8). The fourteen systems are
compared to the independent aircraft $CO_2$ measurements between 2 and 7 km above sea level between 2001 and
2023. Results are shown in Figure S5 and discussed in Supplement S.5.2.
With a relatively small ensemble of systems that cover at least one full decade (N=10), and which moreover
share some a-priori fluxes used with one another, or with the process-based models, it is difficult to justify using
their mean and standard deviation as a metric for uncertainty across the ensemble. We therefore report their full
range (min-max) without their mean. More details on the atmospheric inversion methodology can be found in
Supplement S.5.
**2.8 Atmospheric oxygen based estimate**
Long-term atmospheric $O_2$ and $CO_2$ observations allow estimation of the global ocean and land carbon sinks,
due to the coupling of $O_2$ and $CO_2$ with distinct exchange ratios for fossil fuel emissions and land uptake, and
uncoupled $O_2$ and $CO_2$ ocean exchange (Keeling and Manning, 2014). The global ocean and net land carbon
sinks were calculated following methods and constants used in Keeling and Manning (2014) but modified to
also include the effective $O_2$ source from metal refining (Battle et al., 2023). For the exchange ratio of the net
land sink at value of 1.05 is used, following Resplandy et al. (2019). For fossil fuels, the following values are
used: gas: 1.95 (+/-) 0.04, liquid: 1.44 (+/-) 0.03, solid: 1.17 (+/-) 0.03, cement: 0 (+/-) 0, gas flaring: 1.98 (+/-)
0.07 (Keeling, 1988). Atmospheric $O_2$ is observed as $\delta(O_2/N_2)$ and combined with $CO_2$ mole fraction
observations into Atmospheric Potential Oxygen (APO, Stephens et al., 1998). The APO observations from
1990 to 2024 were taken from a weighted average of flask records from three stations in the Scripps $O_2$ program
network (Alert, Canada (ALT), La Jolla, California (LJO), and Cape Grim, Australia (CGO), weighted per
Keeling and Manning (2014). Observed $CO_2$ was taken from the globally averaged marine surface annual mean
growth rate from the NOAA/GML Global Greenhouse Gas Reference Network (Lan et al., 2024). The $O_2$ source
from ocean warming is based on ocean heat content from updated data from NOAA/NCEI (Levitus et al., 2012).
The effective $O_2$ source from metal refining is based on production data from Bray (2020), Flanagan (2021), and
Tuck (2022). Uncertainty was determined through a Monte Carlo approach with 20,000 iterations, using





uncertainties prescribed in Keeling and Manning (2014), including observational uncertainties from Keeling et
al. (2007) and autoregressive errors in fossil fuel emissions (Ballantyne et al., 2015). The reported uncertainty is
1 standard deviation of the ensemble. The difference between the atmospheric $O_2$ estimate for GCB2023 is due
to a revision to the Scripps $O_2$ program $CO_2$ data. As for the atmospheric inversions, the $O_2$ based estimates also
closes the carbon balance ($B_{IM} = 0$) by design and provides another independent estimate of the ocean and land
fluxes. Note that the $O_2$ method requires a correction for global air-sea $O_2$ flux, which has the largest uncertainty
at annual time scales, but which is still non negligible for decadal estimates (Nevison et al., 2008).
**2.9      Earth System Models estimate**
Reconstructions and predictions from decadal prediction systems based on Earth system models (ESMs) provide
a novel line of evidence in assessing the atmosphere-land and atmosphere-ocean carbon fluxes in the past
decades and predicting their changes for the current year. The decadal prediction systems based on ESMs used
here consist of three sets of simulations: (i) uninitialized freely evolving historical simulations (1850-2014); (ii)
assimilation reconstruction incorporating observational data into the model (1960-2023); (iii) initialised
prediction simulations for the 1981-2024 period, starting every year from initial states obtained from the above
assimilation simulations. The assimilations are designed to reconstruct the actual evolution of the Earth system
by assimilating essential fields from data products. The assimilations' states, which are expected to be close to
observations, are used to start the initialised prediction simulations used for the current year (2024) global
carbon budget. Similar initialised prediction simulations starting every year (Nov. 1st or Jan. 1st) over the 1981-
2023 period (i.e., hindcasts) are also performed for predictive skill quantification and for bias correction. More
details on the illustration of a decadal prediction system based on an ESM can refer to Figure 1 of Li et al.
756    (2023).

By assimilating physical atmospheric and oceanic data products into the ESMs, the models are able to reproduce
the historical variations of the atmosphere-sea $CO_2$ fluxes, atmosphere-land $CO_2$ fluxes, and atmospheric $CO_2$
growth rate (Li et al., 2016, 2019; Lovenduski et al., 2019a,b; Ilyina et al., 2021; Li et al., 2023). Furthermore,
the ESM-based predictions have proven their skill in predicting the air-sea $CO_2$ fluxes for up to 6 years, the air-
land $CO_2$ fluxes and atmospheric $CO_2$ growth for 2 years (Lovenduski et al., 2019a,b; Ilyina et al., 2021; Li et
al., 2023). The reconstructions from the fully coupled model simulations ensure a closed budget within the Earth
system, i.e., no budget imbalance term.
Five ESMs, i.e., CanESM5 (Swart et al., 2019; Sospedra-Alfonso et al., 2021), EC-Earth3-CC (Döscher et al.
2021; Bilbao et al., 2021; Bernardello et al., 2024), IPSL-CM6A-CO2-LR (Boucher et al., 2020), MIROC-ES2L
(Watanabe et al., 2020), and MPI-ESM1-2-LR (Mauritsen et al., 2019; Li et al., 2023), have performed the set of
prediction simulations. Each ESM uses a different assimilation method and combination of data products
incorporated in the system, more details on the models configuration can be found in Table 4 and Supplementary
Table S5. The ESMs use external forcings from the Coupled Model Intercomparison Project Phase 6 (CMIP6)
historical (1960-2014) plus SSP2-4.5 baseline and CovidMIP two-year blip scenario (2015-2024) (Eyring et al.,
2016; Lamboll et al., 2021). The $CO_2$ emissions forcing from 2015-2024 are substituted by GCB-GridFED



(v2024.0, Jones et al., 2024a) to provide a consistent $CO_2$ forcing. Reconstructions of atmosphere-ocean $CO_2$
fluxes ($S_{OCEAN}$) and atmosphere-land $CO_2$ fluxes ($S_{LAND}$-$E_{LUC}$) for the time period from 1960-2023 are assessed
here. Predictions of the atmosphere-ocean $CO_2$ flux, atmosphere-land $CO_2$ flux, and atmospheric $CO_2$ growth for
2024 are calculated based on the predictions at a lead time of 1 year. The predictions are bias corrected using the
1985-2014 climatology mean of GCB2022 (Friedlingstein et al., 2022), more details on methods can be found in
Boer et al. (2016) and Li et al. (2023). The ensemble size of initialized prediction simulations is 10, and the
ensemble mean for each individual model is used here. The ESMs are used here to support the assessment of
$S_{OCEAN}$ and net atmosphere-land $CO_2$ flux ($S_{LAND}$ - $E_{LUC}$) over the 1960-2023 period, and to provide an estimate
of the 2024 projection of $G_{ATM}$.
**2.10   Processes not included in the global carbon budget**
The contribution of anthropogenic CO and $CH_4$ to the global carbon budget is not fully accounted for in Eq. (1)
and is described in Supplement S.6.1. The contributions to $CO_2$ emissions of decomposition of carbonates not
accounted for is described in Supplement S.6.2. The contribution of anthropogenic changes in river fluxes is
conceptually included in Eq. (1) in $S_{OCEAN}$ and in $S_{LAND}$, but it is not represented in the process models used to
quantify these fluxes. This effect is discussed in Supplement S.6.3. Similarly, the loss of additional sink capacity
from reduced forest cover is missing in the combination of approaches used here to estimate both land fluxes
($E_{LUC}$ and $S_{LAND}$) and its potential effect is discussed and quantified in Supplement S.6.4.
**3      Results**
For each component of the global carbon budget, we present results for three different time periods: the full
historical period, from 1850 to 2023, the decades in which we have atmospheric concentration records from
Mauna Loa (1960-2023), a specific focus on last year (2023), and the projection for the current year (2024).
Subsequently, we assess the estimates of the budget components of the last decades against the top-down
constraints from inverse modelling of atmospheric observations, the land/ocean partitioning derived from the
atmospheric $O_2$ measurements, and the budget components estimates from the ESMs assimilation simulations.
Atmospheric inversions further allow for an assessment of the budget components with a regional breakdown of
land and ocean sinks.
**3.1    Fossil $CO_2$ Emissions**
**3.1.1    Historical period 1850-2023**
Cumulative fossil $CO_2$ emissions for 1850-2023 were 490 ± 25 GtC, including the cement carbonation sink
(Figure 3, Table 8, with all cumulative numbers rounded to the nearest 5GtC). In this period, 46% of global
fossil $CO_2$ emissions came from coal, 35% from oil, 15% from natural gas, 3% from decomposition of
carbonates, and 1% from flaring. In 1850, the UK stood for 62% of global fossil $CO_2$ emissions. In 1893 the
combined cumulative emissions of the current members of the European Union reached and subsequently
surpassed the level of the UK. Since 1917 US cumulative emissions have been the largest. Over the entire



period 1850-2023, US cumulative emissions amounted to 120GtC (24% of world total), the EU's to 80 GtC
(16%), China's to 75 GtC (15%), and India's to 15 GtC (3%).
In addition to the estimates of fossil $CO_2$ emissions that we provide here (see Section 2.1), there are three global
datasets with long time series that include all sources of fossil $CO_2$ emissions: CDIAC-FF (Hefner and Marland,
2024), CEDS version 2024_07_08 (Hoesly et al., 2024) and PRIMAP-hist version 2.6 (Gütschow et al., 2016;
Gütschow et al., 2024), although these datasets are not entirely independent from each other (Andrew, 2020a).
CEDS has cumulative emissions over 1750-2022 at 480 GtC, CDIAC-FF has 481 GtC, GCP 484 GtC,
PRIMAP-hist CR 490 GtC, and PRIMAP-hist TR 492 GtC. CDIAC-FF excludes emissions from lime
production. CEDS estimates higher emissions from international shipping in recent years, while PRIMAP-hist
has higher fugitive emissions than the other datasets. However, in general these four datasets are in relative
agreement as to total historical global emissions of fossil $CO_2$.

### 817    3.1.2    Recent period 1960-2023

Global fossil $CO_2$ emissions, $E_{FOS}$ (including the cement carbonation sink), have increased every decade from an
average of $3.0 \pm 0.2$ GtC yr$^{-1}$ for the decade of the 1960s to an average of $9.7 \pm 0.5$ GtC yr$^{-1}$ during 2014-2023
(Table 7, Figure 2 and Figure 5). The growth rate in these emissions decreased between the 1960s and the
1990s, from 4.3% yr$^{-1}$ in the 1960s (1960-1969), 3.2% yr$^{-1}$ in the 1970s (1970-1979), 1.6% yr$^{-1}$ in the 1980s
(1980-1989), to 1.0% yr$^{-1}$ in the 1990s (1990-1999). After this period, the growth rate began increasing again in
the 2000s at an average growth rate of 2.8% yr$^{-1}$, decreasing to 0.6% yr$^{-1}$ for the last decade (2014-2023).
China's emissions increased by +1.9% yr$^{-1}$ on average over the last 10 years dominating the global trend, and
India's emissions increased by +3.6% yr$^{-1}$, while emissions decreased in EU27 by 2.1% yr$^{-1}$, and in the USA by
1.2% yr$^{-1}$. Figure 6 illustrates the spatial distribution of fossil fuel emissions for the 2014-2023 period.
$E_{FOS}$ reported here includes the uptake of $CO_2$ by cement via carbonation which has increased with increasing
stocks of cement products, from an average of 20 MtC yr$^{-1}$ (0.02 GtC yr$^{-1}$) in the 1960s to an average of 200MtC
yr$^{-1}$ (0.2 GtC yr$^{-1}$) during 2014-2023 (Figure 5).

### 830    3.1.3    Final year 2023

Global fossil $CO_2$ emissions were slightly higher, 1.4%, in 2023 than in 2022, with an increase of 0.14 GtC to
reach $10.1 \pm 0.5$ GtC (including the 0.21 GtC cement carbonation sink) in 2023 (Figure 5), distributed among
coal (41%), oil (32%), natural gas (21%), cement (4%), flaring (<1%), and others (<1%). Compared to 2022, the
2023 emissions from coal, oil, and gas increased by 1.4%, 2.5%, and 0.1% respectively, while emissions from
cement decreased by 2%. All annual growth rates presented are adjusted for the leap year, unless stated
otherwise.
In 2023, the largest absolute contributions to global fossil $CO_2$ emissions were from China (31%), the USA
(13%), India (8%), and the EU27 (7%). These four regions account for 59% of global fossil $CO_2$ emissions,
while the rest of the world contributed 41%, including international aviation and marine bunker fuels (3% of the

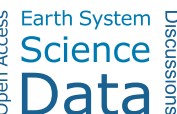

total). Growth rates for these countries from 2022 to 2023 were 4.9% (China), -3.3% (USA), -8.4% (EU27), and
8.2% (India), with +0.7% for the rest of the world, including international aviation and marine bunker fuels
(+9.5%). The per-capita fossil $CO_2$ emissions in 2023 were 1.3 tC person$^{-1}$ yr$^{-1}$ for the globe, and were 3.9
(USA), 2.3 (China), 1.5 (EU27) and 0.6 (India) tC person$^{-1}$ yr$^{-1}$ for the four highest emitters (Figure 5).
**3.1.4    Year 2024 Projection**
Globally, we estimate that global fossil $CO_2$ emissions (including cement carbonation, -0.21 GtC) will grow by
0.8% in 2024 (-0.3% to +1.9%) to 10.2 GtC (37.4 GtCO$_2$), an historical record high[2]. Carbon Monitor projects a
similar 2024 increase of 0.6% (-0.7% to 1.9%). GCB estimates of changes in 2024 emissions per fuel types,
relative to 2023, are projected to be 0.2% (range -1.0% to 1.4%) for coal, +0.9% (range 0.0% to 1.8%) for oil,
+2.4% (range 1.1% to 3.8%) for natural gas, and -2.8% (range -4.7% to -0.9%) for cement.
For China, projected fossil emissions in 2024 are expected to increase slightly by 0.2% (range -1.6% to 2.0%)
compared with 2023 emissions, bringing 2023 emissions for China around 3.3 GtC yr$^{-1}$ (12.0 GtCO$_2$ yr$^{-1}$). In
comparison, the Carbon Monitor estimate projects a 2024 decrease of 0.8% (range -3.8% to 1.9%). Our
projected changes by fuel for China are +0.3% for coal, -0.8% for oil, +8%.0 natural gas, and -8.1% for cement.
For the USA, using the Energy Information Administration (EIA) emissions projection for 2024 combined with
cement clinker data from USGS, we project a decrease of 0.6% (range -2.9% to 1.7%) compared to 2023,
bringing USA 2023 emissions to around 1.3 GtC yr$^{-1}$ (4.9 GtCO$_2$ yr$^{-1}$). Carbon Monitor projects a 2024 increase
in USA emissions of 1.2% (-1.0% to 3.5%). Our projected changes by fuel are -3.5% for coal, -0.7% for oil,
+1.0% for natural gas, and -5.8% for cement.
For the European Union, our projection for 2024 is for a decrease of 3.8% (range -6.2% to -1.4%) relative to
2023, with 2024 emissions around 0.7 GtC yr$^{-1}$ (2.4 GtCO$_2$ yr$^{-1}$). The Carbon Monitor projection for the EU27 is
slightly lower than GCB with a decrease of 5.5% (-9.2% to -1.9%). Our projected changes by fuel are -15.8%
for coal, +0.2% for oil, -1.3% for natural gas, and -3.5% for cement.
For India, our projection for 2024 is an increase of 4.6% (range of 3.0% to 6.1%) over 2023, with 2024
emissions around 0.9 GtC yr$^{-1}$ (3.2 GtCO$_2$ yr$^{-1}$). The Carbon Monitor projection for India is an increase of 5.5%
(1.9% to 9.1%). Our projected changes by fuel are +4.5% for coal, +3.6% for oil, +11.8% for natural gas, and
+4.0% for cement.
International aviation and shipping are projected to increase by 7.8% in 2024, with international aviation
projected to be up 14% over 2023, continuing to recover from pandemic lows, and international shipping
projected to rise by 3%. The Carbon Monitor projects international aviation and shipping to increase by 3.3% in

870 2024.

---

[2] Growth rates in this section use a leap year adjustment that corrects for the extra day in 2024.



For the rest of the world, the expected change for 2024 is an increase of 1.1% (range -1.0% to 3.3%) with 2024
emissions around 4.0 GtC yr$^{-1}$ (14.5 GtCO$_2$ yr$^{-1}$), similar to the Carbon Monitor projection of 1.1% (range -0.1%
to 2.3%). The fuel-specific projected 2024 growth rates for the rest of the world are: +0.5% for coal, +0.5% for
oil, +2.2% for natural gas, +2.0% for cement.
For traceability, Table S6 provides a comparison of annual projections from GCB since 2015 with the actual
emissions assessed in the subsequent GCB annual report.

### 3.2 Emissions from Land Use Change

#### 3.2.1 Historical period 1850-2023

Cumulative CO$_2$ emissions from land-use change (E$_{LUC}$) for 1850-2023 were 225 ± 65 GtC (Table 8; Figure 3;
Figure 16). The cumulative emissions from E$_{LUC}$ show a large spread among individual estimates of 150 GtC
(H&C2023), 205 GtC (OSCAR), 250 GtC (LUCE) and 285 GtC (BLUE) for the four bookkeeping models and a
similar wide estimate of 250 ± 85 GtC for the DGVMs (all cumulative numbers are rounded to the nearest 5
GtC). Vegetation biomass observations provide independent constraints on the E$_{LUC}$ estimates (Li et al., 2017).
Over the 1901-2012 period, the GCB bookkeeping models cumulative E$_{LUC}$ amounts to 165 GtC [105 to 210
GtC], similar to the observation-based estimate of 155 ± 50 GtC (Li et al., 2017).

#### 3.2.2 Recent period 1960-2023

In contrast to growing fossil emissions, CO$_2$ emissions from land-use, land-use change, and forestry remained
relatively constant (around 1.5 GtC yr$^{-1}$) over the 1960-1999 period. Since then, they have shown a statistically
significant decrease of about 0.2 GtC per decade, reaching 1.1 ± 0.7 GtC yr$^{-1}$ for the 2014-2023 period (Table
7), but with significant spread, from 0.8 to 1.3 GtC yr$^{-1}$ across the four bookkeeping models (Table 5, Figure 7).
Different from the bookkeeping average, the DGVMs average grows slightly larger over the 1980-2010 period
and shows no sign of decreasing emissions in the recent decades, apart from in the most recent decade (Table 5,
Figure 7). This is, however, expected as DGVM-based estimates include the loss of additional sink capacity,
which grows with time, while the bookkeeping estimates do not (Supplement S.6.4).
We separate net E$_{LUC}$ into five component fluxes to gain further insight into the drivers of net emissions:
deforestation, forest (re-)growth, wood harvest and other forest management, peat drainage and peat fires, and
all other transitions (Figure 7c; supplemental Sec. S.2.1). We further decompose the deforestation and the forest
(re-)growth term into contributions from shifting cultivation vs permanent forest cover changes (Figure 7d).
Averaged over the 2014-2023 period and over the four bookkeeping estimates, fluxes from deforestation amount
to 1.7 [1.4 to 2.3] GtC yr$^{-1}$ (Table 5), of which 1.0 [0.8, 1.1] GtC yr$^{-1}$ are from permanent deforestation. Fluxes
from forest (re-)growth amount to -1.2 [-1.5, -0.9] GtC yr$^{-1}$ (Table 5), of which -0.5 [-0.7, -0.3] GtC yr$^{-1}$ are from
re/afforestation and the remainder from forest regrowth in shifting cultivation cycles. Emissions from wood
harvest and other forest management (0.3 [0.0, 0.6] GtC yr$^{-1}$), peat drainage and peat fires (0.2 [0.2, 0.3] GtC yr$^{-1}$
$^{-1}$) and the net flux from other transitions (0.1 [0.0, 0.1] GtC yr$^{-1}$) are substantially less important globally (Table





5). However, the small net flux from wood harvest and other forest management contains substantial gross
fluxes that largely compensate each other (see Figure S8): 1.4 [0.9, 2.0] GtC yr$^{-1}$ emissions result from the
decomposition of slash and the decay of wood products and -1.1 [-1.4, -0.8] GtC yr$^{-1}$ removals result from
regrowth after wood harvesting.
The split into component fluxes clarifies the potentials for emission reduction and carbon dioxide removal: the
emissions from permanent deforestation - the largest of our component fluxes - could be halted (largely) without
compromising carbon uptake by forests, contributing substantially to emissions reduction. By contrast, reducing
wood harvesting would have limited potential to reduce emissions as it would be associated with less forest
regrowth; removals and emissions cannot be decoupled here on long timescales. A similar conclusion applies to
removals and emissions from shifting cultivation, which we have therefore separated out. Carbon Dioxide
Removal (CDR) in forests could instead be increased by permanently increasing the forest cover through
re/afforestation. Our estimate of about -0.5 GtC yr$^{-1}$ removed on average each year during 2014-2023 by
re/afforestation is similar to independent estimates that were derived from NGHGIs for CDR in managed forests
(through re/afforestation plus forest management) for 2013-2022 (-0.5 GtC yr$^{-1}$, Pongratz et al., 2024).
Re/afforestation constitutes the vast majority of all current CDR (Pongratz et al., 2024). Though they cannot be
compared directly to annual fluxes from the atmosphere, CDR through transfers between non-atmospheric
reservoirs such as in durable HWPs, biochar, or BECCS comprise much smaller amounts of carbon. 218 MtC
yr$^{-1}$ have been estimated to be transferred to HWPs, averaged over 2013-2022. The net flux of HWPs,
considering the re-release of $CO_2$ through their decay, amounts to 91 MtC yr$^{-1}$ over that period (Pongratz et al.,
2024). Note that some double-counting between the CDR through HWPs and the CDR through re/afforestation
exists if the HWPs are derived from newly forested areas. BECCS projects have been estimated to store 0.1 MtC
yr$^{-1}$ in geological projects worldwide in 2023, biochar projects 0.2 MtC yr$^{-1}$ (Pongratz et al., 2024). "Blue
carbon", i.e. coastal wetland management such as restoration of mangrove forests, saltmarshes and seagrass
meadows, though at the interface of land and ocean carbon fluxes, are counted towards the land-use sector as
well. Currently, bookkeeping models do not include blue carbon; however, current CDR deployment in coastal
wetlands is small globally, less than 0.003MtC yr$^{-1}$ (Powis et al., 2023).
The statistically significant decrease in $E_{LUC}$ since the late-1990s, including the larger drop within the most
recent decade, is due to the combination of decreasing emissions from deforestation (in particular permanent
deforestation) and increasing removals from forest regrowth (with those from re/afforestation stagnating
globally in the last decade). Emissions in 2014-2023 are 28% lower than in the late-1990s (1995-2004) and 20%
lower than in 2004-2013. The steep drop in $E_{LUC}$ after 2015 is due to the combined effect from a peak in peat
fire emissions in 2015 and a long-term decline in deforestation emissions in many countries over the 2010-2020
period with largest declines in the Democratic Republic of the Congo, Brazil, China, and Indonesia. Since the
processes behind gross removals, foremost forest regrowth and soil recovery, are all slow, while gross emissions
include a large instantaneous component, short-term changes in land-use dynamics, such as a temporary
decrease in deforestation, influences gross emissions dynamics more than gross removals dynamics, which
rather are a response to longer-term dynamics. Component fluxes often differ more across the four bookkeeping
estimates than the net flux, which is expected due to different process representation; in particular, the treatment





of shifting cultivation, which increases both gross emissions and removals, differs across models, but also net and gross wood harvest fluxes show high uncertainty. By contrast, models agree relatively well for emissions from permanent deforestation.

Overall, highest land-use emissions occur in the tropical regions of all three continents. The top three emitters (both cumulatively 1959-2023 and on average over 2014-2023) are Brazil (in particular the Amazon Arc of Deforestation), Indonesia and the Democratic Republic of the Congo, with these 3 countries contributing 0.7 GtC yr$^{-1}$ or 60% of the global net land-use emissions (average over 2014-2023) (Figure 6b, Figure 7b). This is related to massive expansion of cropland, particularly in the last few decades in Latin America, Southeast Asia, and sub-Saharan Africa (Hong et al., 2021), to a substantial part for export of agricultural products (Pendrill et al., 2019). Emission intensity is high in many tropical countries, particularly of Southeast Asia, due to high rates of land conversion in regions of carbon-dense and often still pristine, undegraded natural forests (Hong et al., 2021). Emissions are further increased by peat fires in equatorial Asia (GFED4s, van der Werf et al., 2017). Our estimates of high E$_{LUC}$ in China has been revised down since the 1980s as compared to GCB2023 related to the update of the land-use forcing, which is now based on the cropland dataset by Yu et al. (2022) (see Supplement S.2.2), which suggests lower cropland expansion and thus less deforestation than the previous datasets assumed. Uptake due to land-use change occurs in several regions of the world (Figure 6b) particularly because of re/afforestation. Highest CDR in the last decade is seen in China, where our estimates show an even larger uptake since 2010 compared to GCB2023 related to the updated land-use forcing, in the EU27, partly related to expanding forest area as a consequence of the forest transition in the 19[th] and 20[th] century and subsequent regrowth of forest (Mather 2001; McGrath et al., 2015), and in the U.S. Substantial uptake through re/afforestation also exists in other regions such as Brazil, Myanmar or Russia, where, however, emissions from deforestation and other land-use changes dominate the net flux.

While the mentioned patterns are robust and supported by independent literature, we acknowledge that model spread is substantially larger on regional than global levels, as has been shown for bookkeeping models (Bastos et al., 2021) as well as DGVMs (Obermeier et al., 2021). Assessments for individual regions are being performed as part of REgional Carbon Cycle Assessment and Processes (RECCAP2; Ciais et al., 2020, Poulter et al., 2022) or already exist for selected regions (e.g., for Europe by Petrescu et al., 2020, for Brazil by Rosan et al., 2021, for 8 selected countries/regions in comparison to inventory data by Schwingshackl et al., 2022). The revisions since GCB2023 reflect such uncertainties: The integration of a fourth bookkeeping model alters our estimates, though only to a limited extent given that the new model LUCE lies in between the other three models for the global E$_{LUC}$ estimates. Larger changes are obvious at regional level due to the revisions of the land-use forcing with a general update to more recent FAO input for agricultural areas and wood harvest, new MapBiomas input for Brazil and Indonesia and the updated cropland dataset in China.

The NGHGI data under the LULUCF sector and the LULUCF estimates from FAOSTAT differ from the global models' definition of E$_{LUC}$ (see Section 2.2.1). In the NGHGI reporting, the natural fluxes (S$_{LAND}$) are counted towards E$_{LUC}$ when they occur on managed land (Grassi et al., 2018). To compare our results to the NGHGI approach, we perform a translation of our E$_{LUC}$ estimates by adding S$_{LAND}$ in managed forest from the DGVMs



simulations (following the methodology described in Grassi et al., 2023) to the bookkeeping $E_{LUC}$ estimate (see
Supplement S.2.3). For the 2014-2023 period, we estimate that 1.8 GtC yr$^{-1}$ of $S_{LAND}$ occurred in managed
forests. Adding this sink to $E_{LUC}$ changes $E_{LUC}$ from being a source of 1.1 GtC yr$^{-1}$ to a sink of 0.7 GtC yr$^{-1}$, very
similar to the NGHGI estimate that yields a sink of 0.8 GtC yr$^{-1}$ (Figure 8, Table S10). We further apply a mask
of managed land to the net atmosphere-to-land flux estimate from atmospheric inversions to obtain inverse
estimates that are comparable to the NGHGI estimates and to the translated $E_{LUC}$ estimates from bookkeeping
models (see Supplement S.2.3). The inversion-based net flux in managed land indicates a sink of 0.7 GtC yr$^{-1}$
for 2014-2023, which agrees very well with the NGHGI and the translated $E_{LUC}$ estimates (Figure 8, Table S10).
Additionally, the interannual variability of the inversion estimates and the translated $E_{LUC}$ estimates show a
remarkable agreement (Pearson correlation of 0.81 in 2000-2023), which supports the suggested translation
approach.
The translation approach has been shown to be generally applicable also at the country-level (Grassi et al., 2023;
Schwingshackl et al., 2022). Country-level analysis suggests, e.g., that the bookkeeping method estimates higher
deforestation emissions than the national report in Indonesia, but less $CO_2$ removal by afforestation than the
national report in China. The fraction of the natural $CO_2$ sinks that the NGHGI estimates include differs
substantially across countries, related to varying proportions of managed vs total forest areas (Schwingshackl et
al., 2022). By comparing $E_{LUC}$ and NGHGI on the basis of the component fluxes used above, we find that our
estimates reproduce very closely the NGHGI estimates for emissions from permanent deforestation, peat
emissions, and other transitions (Figure 8), although a difference in sign for the latter (small source in
bookkeeping estimates, small sink in NGHGI) creates a notable difference between NGHGI and bookkeeping
estimates. Fluxes due to forest (re-)growth & other forest management, that is, (re-)growth from re/afforestation
plus the net flux from wood harvesting and other forest management and emissions and removals in shifting
cultivation cycles, constitute a large sink in the NGHGI (-1.9 GtC yr$^{-1}$ averaged over 2014-2023), since they
also include $S_{LAND}$ in managed forests. Summing up the bookkeeping estimates of (re-)growth from
re/afforestation, the net flux from wood harvesting and other forest management, and the emissions and
removals in shifting cultivation cycles, and adding $S_{LAND}$ in managed forests yields a flux of -2.0 GtC yr$^{-1}$
(averaged over 2014-2023), which compares well with the NGHGI estimate. Though estimates between
NGHGI, FAOSTAT and the translated budget estimates still differ in value and need further analysis, the
approach suggested by Grassi et al. (2023), which we adopt here, provides a feasible way to relate the global
models' and NGHGI approach to each other and thus link the anthropogenic carbon budget estimates of land
$CO_2$ fluxes directly to the Global Stocktake, as part of the UNFCCC Paris Agreement.
**3.2.3    Final year 2023**
The global $CO_2$ emissions from land-use change are estimated as 1.0 ± 0.7 GtC in 2023, similar to the 2022
estimate. However, confidence in the annual change remains low. Despite El Niño conditions, which in general
lead to more fires in deforestation areas, peat fire emissions in Indonesia remained below average (GFED4.1s;
updated from van der Werf et al., 2017). In South America, emissions from tropical deforestation and



degradation fires have been about average, as effects of the El Niño in the Amazon, such as droughts, are not
expected before 2024.

#### 3.2.4   Year 2024 Projection

In Southeast Asia, peat fire emissions have further dropped (from 27 Tg C in 2023 to 1 Tg C in 2024 through
October 17 2024; GFED4.1s, van der Werf et al., 2017), as have tropical deforestation and degradation fires
(from 33 Tg C to 6 Tg C) as the El Niño conditions ceased. By contrast, emissions from tropical deforestation
and degradation fires in South America have risen from 121 Tg C in 2023 to 324 Tg C in 2024 up until October
17, as the impacts of the El Niño unfold, in particular drought conditions since 2023. The 2024 South American
fire emissions are among the highest values in the record, which started in 1997. Part of the increase is due to
elevated fire activity in the wetlands of the Pantanal. Disentangling the degree to which interannual variability in
rainfall patterns and stronger environmental protection measures in both Indonesia after their 2015 high fire
season and in Brazil after the change in government play a role in fire trends is an important research topic.
Cumulative 2024 fire emission estimates through October 17 2024 are 422 Tg C for global deforestation and
degradation fires and 1 Tg C for peatland fires in Southeast Asia.
Based on these estimates, we expect $E_{LUC}$ emissions of around 1.1 GtC (4.2 GtCO$_2$) in 2024, slightly above the
2023 level. Note that although our extrapolation includes tropical deforestation and degradation fires, the
degradation attributable to selective logging, edge-effects or fragmentation is not captured. Further,
deforestation and fires in deforestation zones may become more disconnected, partly due to changes in
legislation in some regions. For example, Van Wees et al. (2021) found that the contribution from fires to forest
loss decreased in the Amazon and in Indonesia over the period of 2003-2018.

#### 3.3   CDR not based on vegetation

Besides the CDR through land use (Sec. 3.2), the atmosphere to geosphere flux of carbon resulting from carbon
dioxide removal (CDR) activity in 2023 is estimated at 0.011 MtC/yr. This results primarily from 0.009 MtC/yr
of enhanced weathering projects and 0.001 MtC/yr of DACCS. While it represents a growth of 200% in the
anthropogenic sink, from the 0.0036 MtC/yr estimate in 2022, it remains about a million times smaller than
current fossil CO$_2$ emissions.

#### 3.4   Total anthropogenic emissions

Cumulative anthropogenic CO$_2$ emissions (fossil and land use) for 1850-2023 totalled 710 ± 70 GtC (2605 ±
260 GtCO$_2$), of which 70% (500 GtC) occurred since 1960 and 34% (240 GtC) since 2000 (Table 7 and 8).
Total anthropogenic emissions more than doubled over the last 60 years, from 4.6 ± 0.7 GtC yr$^{-1}$ for the decade
of the 1960s to an average of 10.8 ± 0.9 GtC yr$^{-1}$ during 2014-2023, and reaching 11.1 ± 0.9 GtC (40.6 ± 3.2
GtCO$_2$) in 2023. However, total anthropogenic CO$_2$ emissions have been stable over the last decade (zero
growth rate over the 2014-2023 period), much slower than the 2.0% growth rate over the previous decade
1049   (2004-2013).



During the historical period 1850-2023, 31% of historical emissions were from land use change and 69% from fossil emissions. However, fossil emissions have grown significantly since 1960 while land use changes have not, and consequently the contributions of land use change to total anthropogenic emissions were smaller during recent periods, 18% during the period 1960-2023 and down to 10% over the last decade (2014-2023).

For 2024, we project global total anthropogenic $CO_2$ emissions from fossil and land use changes to be around 11.3 GtC (41.6 $GtCO_2$), 2% above the 2023 level. All values here include the cement carbonation sink (currently about 0.2 GtC $yr^{-1}$).

## 3.5    Atmospheric $CO_2$

### 3.5.1    Historical period 1850-2023

Atmospheric $CO_2$ concentration was approximately 278 parts per million (ppm) in 1750, reaching 300 ppm in the late 1900s, 350 ppm in the late 1980s, and reaching 419.31 ± 0.1 ppm in 2023 (Lan et al., 2024; Figure 1). The mass of carbon in the atmosphere increased by 51% from 590 GtC in 1750 to 890 GtC in 2023. Current $CO_2$ concentrations in the atmosphere are unprecedented in the last 2 million years and the current rate of atmospheric $CO_2$ increase is at least 10 times faster than at any other time during the last 800,000 years (Canadell et al., 2021).

### 3.5.2    Recent period 1960-2023

The growth rate in atmospheric $CO_2$ level increased from 1.7 ± 0.07 GtC $yr^{-1}$ in the 1960s to 5.2 ± 0.02 GtC $yr^{-1}$ during 2014-2023 with important decadal variations (Table 7, Figure 3 and Figure 4). During the last decade (2014-2023), the growth rate in atmospheric $CO_2$ concentration continued to increase, albeit with large interannual variability (Figure 4).

The airborne fraction (AF) is defined as the ratio of atmospheric $CO_2$ growth rate to total anthropogenic emissions:

$$AF = G_{ATM} / (E_{FOS} + E_{LUC}) \hspace{3cm} (2)$$

It provides a diagnostic of the relative strength of the land and ocean carbon sinks in removing part of the anthropogenic $CO_2$ perturbation. The evolution of AF over the last 60 years shows no significant trend, remaining at around 44%, albeit showing a large interannual and decadal variability driven by the year-to-year variability in $G_{ATM}$ (Figure 10). The observed stability of the airborne fraction over the 1960-2023 period indicates that the ocean and land $CO_2$ sinks have been increasing in pace with the total anthropogenic emissions over that period, removing on average about 56% of the emissions (see Sections 3.6.2 and 3.7.2).



### 3.5.3 Final year 2023

The growth rate in atmospheric $CO_2$ concentration was $5.9 \pm 0.2$ GtC ($2.79 \pm 0.08$ ppm) in 2023 (Figure 4; Lan et al., 2024), well above the 2022 growth rate ($4.6 \pm 0.2$ GtC) or the 2014-2023 average ($5.2 \pm 0.02$ GtC), as to be expected during an El Niño year. The 2023 atmospheric $CO_2$ growth rate was the 4th largest over the 1959-2023 atmospheric observational record, closely following 2015, 2016 and 1998, all strong El Niño years.

### 3.5.4 Year 2024 Projection

The 2024 growth in atmospheric $CO_2$ concentration ($G_{ATM}$) is projected to be about 5.9 GtC (2.76 ppm), still high, which is common for the year after a strong El Niño year. This is the average of the GCB regression method (5.6 GtC, 2.64 ppm) and ESMs the multi-model mean (6.1 GtC, 2.88 ppm). The 2024 atmospheric $CO_2$ concentration, averaged over the year, is expected to reach the level of 422.5 ppm, 52% over the pre-industrial level.

## 3.6 Ocean Sink

### 3.6.1 Historical period 1850-2023

Cumulated since 1850, the ocean sink adds up to $185 \pm 35$ GtC, with more than two thirds of this amount ($130 \pm 25$ GtC) being taken up by the global ocean since 1960. Over the historical period, the ocean sink increased in pace with the anthropogenic emissions exponential increase (Figure 3). Since 1850, the ocean has removed 26% of total anthropogenic emissions.

### 3.6.2 Recent period 1960-2023

The ocean $CO_2$ sink increased from $1.2 \pm 0.4$ GtC yr$^{-1}$ in the 1960s to $2.9 \pm 0.4$ GtC yr$^{-1}$ during 2014-2023 (Table 7), with interannual variations of the order of a few tenths of GtC yr$^{-1}$ (Figure 4, Figure 11). The ocean-borne fraction ($S_{OCEAN}/(E_{FOS}+E_{LUC})$) has been remarkably constant around 25% on average (Figure 10c), with variations around this mean illustrating the decadal variability of the ocean carbon sink. So far, there is no indication of a decrease in the ocean-borne fraction from 1960 to 2022. The increase of the ocean sink is primarily driven by the increased atmospheric $CO_2$ concentration, with the strongest $CO_2$ induced signal in the North Atlantic and the Southern Ocean (Figure 12a). The effect of climate change is much weaker, reducing the ocean sink globally by $0.17 \pm 0.05$ GtC yr$^{-1}$ (-5.9% of $S_{OCEAN}$) during 2014-2023 (all models simulate a weakening of the ocean sink by climate change, range -3.4 to -10.7%), and does not show clear spatial patterns across the GOBMs ensemble (Figure 12b). This is the combined effect of change and variability in all atmospheric forcing fields, previously attributed to wind and temperature changes (LeQuéré et al., 2010, Bunsen et al., 2024). The effect of warming is smaller than expected from offline calculation due to a stabilising feedback from limited exchange between surface and deep waters (Bunsen et al., 2024).

The global net air-sea $CO_2$ flux is a residual of large natural and anthropogenic $CO_2$ fluxes into and out of the ocean with distinct regional and seasonal variations (Figure 6 and S1). Natural fluxes dominate on regional





scales, but largely cancel out when integrated globally (Gruber et al., 2009). Mid-latitudes in all basins and the
high-latitude North Atlantic dominate the ocean $CO_2$ uptake where low temperatures and high wind speeds
facilitate $CO_2$ uptake at the surface (Takahashi et al., 2009). In these regions, formation of mode, intermediate
and deep-water masses transport anthropogenic carbon into the ocean interior, thus allowing for continued $CO_2$
uptake at the surface. Outgassing of natural $CO_2$ occurs mostly in the tropics, especially in the equatorial
upwelling region, and to a lesser extent in the North Pacific and polar Southern Ocean, mirroring a well-
established understanding of regional patterns of air-sea $CO_2$ exchange (e.g., Takahashi et al., 2009, Gruber et
al., 2009). These patterns are also noticeable in the Surface Ocean $CO_2$ Atlas (SOCAT) dataset, where an ocean
$f$CO₂ value above the atmospheric level indicates outgassing (Figure S1). This map further illustrates the data-
sparsity in the Indian Ocean and the southern hemisphere in general.
The largest variability in the ocean sink occurs on decadal time-scales (Figure 11). The ensemble means of
GOBMs and $f$CO₂-products show the same patterns of decadal variability, although with a larger amplitude of
variability in the $f$CO₂-products than in the GOBMs. The ocean sink stagnated in the 1990s and strengthened
between the early 2000s and the mid-2010s (Figure 11; Le Quéré et al., 2007; Landschützer et al., 2015, 2016;
DeVries et al., 2017; Hauck et al., 2020; McKinley et al., 2020, Gruber et al., 2023). More recently, the sink
seems to have entered a phase of stagnation since 2016, largely in response to large inter-annual climate
variability. Different explanations have been proposed for the decadal variability in the 1990s and 2000s,
ranging from the ocean's response to changes in atmospheric wind systems (e.g., Le Quéré et al., 2007, Keppler
and Landschützer, 2019), including variations in upper ocean overturning circulation (DeVries et al., 2017) to
the eruption of Mount Pinatubo in the 1990s (McKinley et al., 2020). The main origin of the decadal variability
is a matter of debate with a number of studies initially pointing to the Southern Ocean (see review in Canadell et
al., 2021), but also contributions from the North Atlantic and North Pacific (Landschützer et al., 2016, DeVries
et al., 2019), or a global signal (McKinley et al., 2020) were proposed.
On top of the decadal variability, interannual variability of the ocean carbon sink is driven by climate variability
with a first-order effect from a stronger ocean sink during large El Niño events (e.g., 1997-1998) (Figure 11;
Rödenbeck et al., 2014, Hauck et al., 2020; McKinley et al. 2017) leading to a reduction in $CO_2$ outgassing from
the Tropical Pacific. During 2010-2016, the ocean $CO_2$ sink appears to have intensified in line with the expected
increase from atmospheric $CO_2$ (McKinley et al., 2020). This effect is similar in the $f$CO₂-products (Figure 11,
ocean sink 2016 minus 2010, GOBMs: +0.42 ± 0.11 GtC yr⁻¹, $f$CO₂-products: +0.44 GtC yr⁻¹, range 0.18 to 0.72
GtC yr⁻¹). The reduction of -0.18 GtC yr⁻¹ (range: -0.41 to -0.03 GtC yr⁻¹) in the ocean $CO_2$ sink in 2017 is
consistent with the return to normal conditions after the El Niño in 2015/16, which caused an enhanced sink in
previous years. After an increasing S_OCEAN in 2018 and 2019, the GOBM and $f$CO₂-product ensemble means
suggest a decrease of S_OCEAN, related to the triple La Niña event 2020-2022, followed by a rebound in 2023
linked to the onset of an El Niño event.
Although all individual GOBMs and $f$CO₂-products fall within the observational constraint, the ensemble means
of GOBMs, and $f$CO₂-products (adjusted for the riverine flux) show a mean offset increasing from 0.31 GtC yr⁻¹
in the 1990s to 0.49 GtC yr⁻¹ in the decade 2014-2023 and a slightly lower offset of 0.3 GtC yr⁻¹ in 2023. In this



version of the GCB, the S$_{OCEAN}$ positive trend diverges over time by a factor of 1.4 since 2002 (GOBMs: 0.25 ±
0.04 GtC yr$^{-1}$ per decade, $f$CO$_2$-products: 0.35 GtC yr$^{-1}$ per decade [0.17 to 0.79 GtC yr$^{-1}$ per decade], S$_{OCEAN}$:
0.30 GtC yr$^{-1}$ per decade), but the uncertainty ranges overlap. This divergence is smaller than reported in
previous GCB versions, because of the updated lower sink estimates by the $f$CO$_2$-products for recent years. This
also leads to agreement on the trend since 2010 (GOBMs: 0.18 ± 0.06 GtC yr$^{-1}$ per decade, $f$CO$_2$-products: 0.18
GtC yr$^{-1}$ per decade [-0.36 to 0.73 GtC yr$^{-1}$ per decade] S$_{OCEAN}$: 0.18 GtC yr$^{-1}$ per decade). A hybrid approach
recently constrained the trend 2000-2022 to 0.42 ± 0.06 GtC yr$^{-1}$ decade$^{-1}$ (Mayot et al., 2024), which aligns
with the updated trends of S$_{OCEAN}$ (0.39 GtCyr$^{-1}$ decade$^{-1}$) and of the $f$CO$_2$-products (0.45 [0.28,0.84] GtCyr$^{-1}$
decade$^{-1}$), while the GOBMs result in a lower trend (0.32 ± 0.04 GtC yr$^{-1}$ per decade) over the same period.
In the current dataset, the discrepancy between the two types of estimates stems from a persistently larger
S$_{OCEAN}$ in the $f$CO$_2$-products in the northern extra-tropics since around 2002 and an intermittently larger S$_{OCEAN}$
in the southern extra-tropics in the period 2008-2020 (Figure 14). Note that the discrepancy in the mean flux,
which was located in the Southern Ocean in GCB 2022 and earlier, was reduced due to the choice of the
regional river flux adjustment (Lacroix et al., 2020 instead of Aumont et al., 2001). This comes at the expense of
a discrepancy in the mean S$_{OCEAN}$ of about 0.2 GtC yr$^{-1}$ in the tropics. Likely explanations for the discrepancy in
the trends and decadal variability in the high-latitudes are data sparsity and uneven data distribution (Bushinsky
et al., 2019, Gloege et al., 2021, Hauck et al., 2023a, Mayot et al., 2024). In particular, two $f$CO$_2$-products were
shown to overestimate the Southern Ocean CO$_2$ flux trend by 50 and 130% based on current sampling in a
model subsampling experiment (Hauck et al., 2023a) and the largest trends in the $f$CO$_2$-products occurred in a
data void region in the North Pacific (Mayot et al., 2024). In this respect it is highly worrisome that the coverage
of $f$CO$_2$ observations continues to decline (Dong et al 2024), and is now down to that of the early 2000s (Fig.
11). Another likely contributor to the discrepancy between GOBMs and $f$CO$_2$-products are model biases (as
indicated by the comparison with Mayot et al., 2024, by the large model spread in the South, Figure 14, and the
larger model-data $f$CO$_2$ mismatch, Figure S2).
The reported S$_{OCEAN}$ estimate from GOBMs and $f$CO$_2$-products is 2.2 ± 0.4 GtC yr$^{-1}$ over the period 1994 to
2007, which is in agreement with the ocean interior estimate of 2.2 ± 0.4 GtC yr$^{-1}$, which accounts for the
climate effect on the natural CO$_2$ `flux of` $-0.4 \pm 0.24$ `GtC yr`$^{-1}$ (Gruber et al., 2019) to match the
definition of S$_{OCEAN}$ used here (Hauck et al., 2020). This comparison depends critically on the estimate of the
climate effect on the natural CO$_2$ flux, which is smaller from the GOBMs (-0.1 GtC yr$^{-1}$) than in Gruber et al.
(2019). Uncertainties of these two estimates would also overlap when using the GOBM estimate of the climate
effect on the natural CO$_2$ flux. Similarly, the S$_{OCEAN}$ estimates integrated over the decades 1994-2004 (21.5 GtC
yr$^{-1}$) and 2004-2014 (25.6 GtC yr$^{-1}$) agree with the interior ocean-based estimates of Müller et al. (2023; 21.4 ±
2.8 and 26.5 ± 1.3 GtC yr$^{-1}$), but depend critically on assumptions of the climate effect on natural carbon, which
in turn, are based on the $f$CO$_2$-products in Müller et al. (2023).



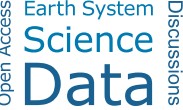

### 3.6.3 Final year 2023

The estimated ocean $CO_2$ sink is $2.9 \pm 0.4$ GtC for 2023. This is a small increase of 0.16 GtC compared to 2022, in line with the expected sink strengthening from the 2023 El Niño conditions. GOBM and $f$CO$_2$-product ensemble mean estimates consistently result in an S$_{OCEAN}$ increase in 2023 (GOBMs: $0.17 \pm 0.15$ GtC, $f$CO$_2$-products: 0.14 [-0.04,0.30] GtC). Eight GOBMs and six $f$CO$_2$-products show an increase in S$_{OCEAN}$, while only two GOBMs and two $f$CO$_2$-products show a minor decrease in S$_{OCEAN}$ of less than 0.05 GtC (Figure 11). The $f$CO$_2$-products have a larger uncertainty at the end of the reconstructed time series, potentially linked to uncertainties related to fewer available observations in the final year and the shift from La Niña to El Niño (see e.g. Watson et al 2020, Pérez et al 2024). Specifically, the $f$CO$_2$-products' estimate of the last year is regularly adjusted in the following release owing to the tail effect and an incrementally increasing data availability. While the monthly grid cells covered may have a lag of only about a year (Figure 11 inset), the values within grid cells may change with 1-5 years lag (see absolute number of observations plotted in previous GCB releases), potentially resulting in annual changes in the flux magnitude from $f$CO$_2$-products.

### 3.6.4 Year 2024 Projection

Using a feed-forward neural network method (see Section 2.5.2) we project an ocean sink of 3.0 GtC for 2024, only 0.1 GtC higher than for the year 2023, consistent with El Niño to neutral conditions in 2024. The set of ESMs predictions support this estimate with a 2024 ocean sink of around 3.0 [2.9, 3.1] GtC.

### 3.6.5 Evaluation of Ocean Models and $f$CO$_2$-products

The process-based model evaluation draws a generally positive picture with GOBMs scattered around the observational values for Southern Ocean sea-surface salinity, Southern Ocean stratification index and surface ocean Revelle factor (Section S3.3 and Table S11). However, the Atlantic Meridional Overturning Circulation at 26°N is underestimated by 8 out of 10 GOBMs and overestimated by one GOBM. It is planned to derive skill scores for the GOBMs in future releases based on these metrics.

The model simulations allow to separate the anthropogenic carbon component (steady state and non-steady state, sim D - sim A) and to compare the model flux and DIC inventory change directly to the interior ocean estimate of Gruber et al. (2019) without further assumptions (Table S11). The GOBMs ensemble average of anthropogenic carbon inventory changes 1994-2007 amounts to 2.4 GtC yr$^{-1}$ and is thus lower than the $2.6 \pm 0.3$ GtC yr$^{-1}$ estimated by Gruber et al. (2019) although within the uncertainty. Only three models fall within the range reported by Gruber et al. (2019). This suggests that the majority of the GOBMs underestimate anthropogenic carbon uptake by 10-20% and some models even more. Comparison to the decadal estimates of anthropogenic carbon accumulation (Müller et al., 2023) are close to the interior ocean data based estimate for the decade 2004-2014 (GOBMs sim D minus sim A, $24.7 \pm 3.6$ GtC yr$^{-1}$, Müller et al. $27.3 \pm 2.5$ GtC yr$^{-1}$), but do not reproduce the supposedly higher anthropogenic carbon accumulation in the earlier period 1994-2004 (GOBMs sim D minus sim A, $21.1 \pm 3.0$ GtC yr$^{-1}$, Müller et al. $29.3 \pm 2.5$ GtC yr$^{-1}$). Analysis of Earth System Models indicate that an underestimation by about 10% may be due to biases in ocean carbon transport and



mixing from the surface mixed layer to the ocean interior (Goris et al., 2018, Terhaar et al., 2021, Bourgeois et
al., 2022, Terhaar et al., 2022), biases in the chemical buffer capacity (Revelle factor) of the ocean (Vaittinada
Ayar et al., 2022; Terhaar et al., 2022) and partly due to a late starting date of the simulations (mirrored in
atmospheric $CO_2$ chosen for the preindustrial control simulation, Table S2, Bronselaer et al., 2017, Terhaar et
al., 2022; 2024). Interestingly, and in contrast to the uncertainties in the surface $CO_2$ flux, we find the largest
mismatch in interior ocean carbon accumulation in the tropics, with smaller contributions from the north and the
south. The large discrepancy in accumulation in the tropics highlights the role of interior ocean carbon
redistribution for those inventories (Khatiwala et al., 2009, DeVries et al., 2023).
The evaluation of the ocean estimates with the $fCO_2$ observations from the SOCAT v2024 dataset for the period
1990-2023 shows an RMSE from annually detrended data of 0.2 to 2.4 μatm for the eight $fCO_2$-products over
the globe (Figure S2). The GOBMs RMSEs are larger and range from 2.7 to 4.9 μatm. The RMSEs are
generally larger at high latitudes compared to the tropics, for both the $fCO_2$-products and the GOBMs. The
$fCO_2$-products have RMSEs of 0.3 to 2.9 μatm in the Tropics, 0.6 to 2.4 μatm in the North, and 0.8 to 2.4 μatm
in the South. Note that the $fCO_2$-products are based on the SOCAT v2024 database, hence SOCAT is not an
independent dataset for the evaluation of the $fCO_2$-products. The GOBMs RMSEs are more spread across
regions, ranging from 2.4 to 3.9 μatm in the tropics, 2.8 to 5.9 μatm in the North, and 2.7 to 6.0 μatm in the
South. The higher RMSEs occur in regions with stronger climate variability, such as the northern and southern
high latitudes (poleward of the subtropical gyres). Additionally, this year we evaluate the trends derived from a
subset of $fCO_2$-products by subsampling four GOBMs used in Friedlingstein et al. (2023; covering the period up
to the year 2022) following the approach of Hauck et al. (2023a) and evaluating the air-sea $CO_2$ flux trend for
the 2001-2021 period, i.e. the period of strong divergence in the air-sea $CO_2$ exchange excluding the final year
to remove the tail effect, against trend biases identified by the GOBM reconstruction. The results indicate a
relationship between reconstruction bias and strength of the decadal trends (see Figure S3), indicating a
tendency of the $fCO_2$-products ensemble to overestimate the air-sea $CO_2$ flux trends in agreement with a recent
study by Mayot et al. (2024).
**3.7     Land Sink**
**3.7.1     Historical period 1850-2023**
Cumulated since 1850, the terrestrial $CO_2$ sink amounts to $220 \pm 60$ GtC, 31% of total anthropogenic emissions.
As for the ocean, more than two thirds of this amount ($150 \pm 40$ GtC) have been taken up by terrestrial
ecosystems since 1960. Over the historical period, the sink increased in pace with the anthropogenic emissions
increase (Figure 3).
**3.7.2     Recent period 1960-2023**
The terrestrial $CO_2$ sink $S_{LAND}$ increased from $1.2 \pm 0.5$ GtC yr$^{-1}$ in the 1960s to $3.2 \pm 0.9$ GtC yr$^{-1}$ during 2014-
2023, with important interannual variations of up to 2 GtC yr$^{-1}$ generally showing a decreased land sink during
El Niño events (Figure 9), responsible for the corresponding enhanced growth rate in atmospheric $CO_2$



concentration. The larger land $CO_2$ sink during 2014-2023 compared to the 1960s is reproduced by all the
DGVMs in response to the increase in both atmospheric $CO_2$, nitrogen deposition, and the changes in climate,
and is consistent with the residual estimated from the other budget terms ($E_{FOS}+E_{LUC}-G_{ATM}-S_{OCEAN}$, Table 5).
Over the period 1960 to present the increase in the global terrestrial $CO_2$ sink is largely attributed to the $CO_2$
fertilisation effect (Prentice et al., 2001, Piao et al., 2009, Schimel et al., 2015) and increased nitrogen
deposition (Huntzinger et al., 2017, O'Sullivan et al., 2019), directly stimulating plant photosynthesis and
increased plant water use in water limited systems, with a small negative contribution of climate change (Figure
12). There is a range of evidence to support a positive terrestrial carbon sink in response to increasing
atmospheric $CO_2$, albeit with uncertain magnitude (Walker et al., 2021). As expected from theory, the greatest
$CO_2$ effect is simulated in the tropical forest regions, associated with warm temperatures and long growing
seasons (Hickler et al., 2008) (Figure 12a). However, evidence from tropical intact forest plots indicate an
overall decline in the land sink across Amazonia (1985-2011), attributed to enhanced mortality offsetting
productivity gains (Brienen et al., 2015, Hubau et al., 2020). During 2014-2023 the land sink is positive in all
regions (Figure 6) with the exception of eastern Brazil, Bolivia, northern Venezuela, Southwest USA, central
Europe and Central Asia, North and South Africa, and eastern Australia, where the negative effects of climate
variability and change (i.e. reduced rainfall and/or increased temperature) counterbalance $CO_2$ effects. This is
clearly visible on Figure 12 where the effects of $CO_2$ (Figure 12a) and climate (Figure 12b) as simulated by the
DGVMs are isolated. The negative effect of climate can be seen across the globe, and is particularly strong in
most of South America, Central America, Southwest US, Central Europe, western Sahel, southern Africa,
Southeast Asia and southern China, and eastern Australia (Figure 12b). Globally, over the 2014-2023 period,
climate change reduces the land sink by $0.87 \pm 0.56$ GtC yr$^{-1}$ (27% of $S_{LAND}$).
Most DGVMs have similar $S_{LAND}$ averaged over 2014-2023, and 14/20 models fall within the 1σ range of the
residual land sink [1.8-3.7 GtC yr$^{-1}$] (see Table 5), and all models but one are within the 2σ range [0.8-4.6 GtC
yr$^{-1}$]. The ED model is an outlier, with a land sink estimate of 5.1 GtC yr$^{-1}$ for the 2014-2023 period, driven by a
strong $CO_2$ fertilisation effect (6.3 GtC yr$^{-1}$ in the $CO_2$ only (S1) simulation). There are no direct global
observations of the land sink ($S_{LAND}$), or the $CO_2$ fertilisation effect, and so we are not yet in a position to rule
out models based on component fluxes if their net land sink ($S_{LAND}-E_{LUC}$) is within the observational uncertainty
provided by atmospheric inversions or $O_2$ measurements (Table 5). Furthermore, DGVMs were compared
against a model-data fusion based analysis of the land carbon cycle (CARDAMOM) (Bloom and Williams,
2015; Bloom et al., 2016). Results suggest good correspondence between approaches at the interannual
timescales, but divergence in the recent trend in $S_{LAND}$ with CARDAMOM simulating a stronger trend than the
DGVM multi-model mean (Figure 9).
Since 2020 the globe has experienced La Niña conditions which would be expected to lead to an increased land
carbon sink. This 3-year long period of La Niña conditions came to an end by the second half of 2023 and
transitioned to an El Niño which lasted until mid-2024. A clear transition from maximum to a minimum in the
global land sink is evident in $S_{LAND}$, from 2022 to 2023 and we find that a El Niño- driven decrease in tropical
land sink is offset by a smaller increase in the high latitude land sink. In the past years several regions



experienced record-setting fire events (see also section 3.8.3). While global burned area has declined over the
past decades mostly due to declining fire activity in savannas (Andela et al., 2017), forest fire emissions are
rising and have the potential to counter the negative fire trend in savannas (Zheng et al., 2021). Noteworthy
extreme fire events include the 2019-2020 Black Summer event in Australia (emissions of roughly 0.2 GtC; van
der Velde et al., 2021), Siberia in 2021, where emissions approached 0.4 GtC or three times the 1997-2020
average according to GFED4s, and Canada in 2023 (Byrne et al., 2024). While other regions, including Western
US and Mediterranean Europe, also experienced intense fire seasons in 2021 their emissions are substantially
lower.
Despite these regional negative effects of climate change on $S_{LAND}$, the efficiency of land to remove
anthropogenic $CO_2$ emissions has remained broadly constant over the last six decades, with a land-borne
fraction ($S_{LAND}/(E_{FOS}+E_{LUC})$) of around 30% (Figure 10b).

### 3.7.3    Final year 2023

The terrestrial $CO_2$ sink from the DGVMs ensemble $S_{LAND}$ was 2.3 ± 1.0 GtC in 2023, 41% below the 2022 La
Niña induced strong sink of 3.9 ± 1.0 GtC, and also below the 2014-2023 average of 3.2 ± 0.9 GtC yr$^{-1}$ (Figure
4, Table 7). We estimate that the 2023 land sink was the lowest since 2015. The severe reduction in the land
sink in 2023 is likely driven by the El Niño conditions, leading to a 58% reduction in $S_{LAND}$ in the tropics (30N-
30S) from 2.8 GtC in 2022 to 1.2 GtC in 2023. This is combined with intense wildfires in Canada that led to a
significant $CO_2$ source (see also Section 3.8.3). We note that the $S_{LAND}$ DGVMs estimate for 2023 of 2.3 ± 1.0
GtC is very similar to the 2.2 ± 1.0 GtC yr$^{-1}$ estimate from the residual sink from the global budget ($E_{FOS}+E_{LUC}-$
$G_{ATM}-S_{OCEAN}$, Table 5).

### 3.7.4    Year 2024 Projection

Using a feed-forward neural network method we project a land sink of 3.2 GtC for 2024, 0.9 GtC larger than the
2023 estimate. As for the ocean sink, we attribute this to the transition from the El Niño conditions in 2023 to a
neutral state. The ESMs do not provide an additional estimate of $S_{LAND}$ as they only simulate the net
atmosphere-land carbon flux ($S_{LAND}-E_{LUC}$).

### 3.7.5    Land Models Evaluation

The evaluation of the DGVMs shows generally higher agreement across models for runoff, and to a lesser extent
for GPP, and ecosystem respiration. These conclusions are supported by a more comprehensive analysis of
DGVM performance in comparison with benchmark data (Sitch et al., 2024). A relative comparison of DGVM
performance (Figure S4) suggests several DGVMs (CABLE-POP, CLASSIC, OCN, ORCHIDEE) may
outperform others at multiple carbon and water cycle benchmarks. However, results from Seiler et al., 2022,
also show how DGVM differences are often of similar magnitude compared with the range across observational
datasets. All models score high enough over the metrics tests to support their use here. There are a few

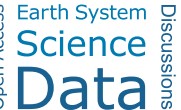

anomalously low scores for individual metrics from a single model, and these can direct the effort to improve
models for use in future budgets.

### 3.8    Partitioning the carbon sinks

#### 3.8.1    Global sinks and spread of estimates

In the period 2014-2023, the bottom-up view of global net ocean and land carbon sinks provided by the GCB,
$S_{OCEAN}$ for the ocean and $S_{LAND} - E_{LUC}$ for the land, agrees closely with the top-down global carbon sinks
delivered by the atmospheric inversions. This is shown in Figure 13, which visualises the individual decadal
mean atmosphere-land and atmosphere-ocean fluxes from each, along with the constraints on their sum offered
by the global fossil $CO_2$ emissions flux minus the atmospheric growth rate ($E_{FOS} - G_{ATM}$, $4.4 \pm 0.5$ Gt C yr$^{-1}$,
Table 7, shown as diagonal line on Figure 13). The GCB estimate for net atmosphere-to-surface flux ($S_{OCEAN}$ +
$S_{LAND} - E_{LUC}$) during 2014-2023 is $4.9 \pm 1.2$ Gt C yr$^{-1}$ (Table 7), with the difference to the diagonal representing
the budget imbalance ($B_{IM}$) of 0.4 GtC yr$^{-1}$ discussed in Section 3.9. By virtue of the inversion methodology, the
atmospheric inversions estimate of the net atmosphere-to-surface flux during 2014-2023 is 4.5 Gt C yr$^{-1}$, with a
$< 0.1$ GtC yr$^{-1}$ imbalance, and thus scatter across the diagonal, with inverse models trading land for ocean fluxes
in their solution. The independent constraint on the net atmosphere-to-surface flux based on atmospheric $O_2$ by
design also closes the balance and is $4.5 \pm 0.9$ GtC yr$^{-1}$ over the 2014-2023 period (orange symbol on Figure
13), while the ESMs estimate for the net atmosphere-to-surface flux over that period average to 4.7 [3.0, 5.8]
GtC yr$^{-1}$(Tables 5 and 6).
The distributions based on the individual models and fCO2-products reveal substantial spread but converge near
the decadal means quoted in Tables 5 to 7. Sink estimates for $S_{OCEAN}$ and from inverse systems are mostly non-
Gaussian, while the ensemble of DGVMs appears more normally distributed justifying the use of a multi-model
mean and standard deviation for their errors in the budget. Noteworthy is that the tails of the distributions
provided by the land and ocean bottom-up estimates would not agree with the global constraint provided by the
fossil fuel emissions and the observed atmospheric $CO_2$ growth rate. This illustrates the power of the
atmospheric joint constraint from $G_{ATM}$ and the global $CO_2$ observation network it derives from.

#### 3.8.1.1    Net atmosphere-to-land flux

The GCB estimate of the net atmosphere-to-land flux ($S_{LAND} - E_{LUC}$), calculated as the difference between
$S_{LAND}$ from the DGVMs and $E_{LUC}$ from the bookkeeping models, amounts to a $2.1 \pm 1.1$ GtC yr$^{-1}$ sink during
2014-2023 (Table 5). Estimates of net atmosphere-to-land flux ($S_{LAND} - E_{LUC}$) from the DGVMs alone ($1.7 \pm$
0.6 GtC yr$^{-1}$, Table 5, green symbol on Figure 13) are slightly lower, although within the uncertainty of the GCB
estimate and also within uncertainty of the global carbon budget constraint ($E_{FOS} - G_{ATM} - S_{OCEAN}$, $1.6 \pm 0.6$ GtC
yr$^{-1}$; Table 7). Also, for 2014-2023, the inversions estimate the net atmosphere-to-land flux is a 1.4 [0.3, 2.2]
GtC yr$^{-1}$ sink, slightly lower than the mean of the DGVMs estimates (purple versus grey symbols on Figure 13).
The independent constraint based on atmospheric $O_2$ is even lower, $1.0 \pm 0.8$ GtC yr$^{-1}$ (orange symbol in Figure
13), although its large uncertainty overlaps with the uncertainty range from other approaches. Last, the ESMs





estimate for the net atmosphere-to-land flux during 2014-2023 is a 2.2 [0.3, 3.6] GtC yr$^{-1}$ sink, more consistent
with the GCB estimates of $S_{LAND}$ – $E_{LUC}$ (Figure 14 top row).
As discussed in Section 3.5.3, the atmospheric growth rate of $CO_2$ was very high in 2023, 5.9 GtC (2.79 ppm)
the 4[th] largest on record. Both DGVMs and inversions assign this large $CO_2$ growth rate to a severe decrease of
the net atmosphere to land flux, and in particular in the tropics (Figure 14). DGVMs simulate a 2023 global the
net atmosphere-to-land flux of 1.1 GtC yr$^{-1}$, a 55% decline relative to the 2.4 GtC yr$^{-1}$ sink in 2022, primarily
driven by the severe reduction in $S_{LAND}$ (-41%, see Section 3.7.3). The tropics (30N-30S) are recording a
dramatic decrease in the net atmosphere-to-land flux from 1.5 GtC yr$^{-1}$ in 2022 to 0.1 GtC yr$^{-1}$ in 2023. The
atmospheric inversion shows a similar story with the global net atmosphere-to-land flux declining from 2.6 GtC
yr$^{-1}$ in 2022 to 0.9 GtC yr$^{-1}$ in 2023 (-64%), with the tropics turning from a 1.0 GtC yr$^{-1}$ sink in 2022 to a 0.4
GtC yr$^{-1}$ source in 2023. Our results are broadly consistent with the Ke et al. (2024) study which reported a
global atmosphere-to-land flux of $0.4 \pm 0.2$ GtC yr$^{-1}$ in 2023.
In addition to the large decline of the tropical land uptake, the northern extra tropics experienced warmer than
average conditions, in particular in the summer over North America and North Eurasia. In Canada alone, 2023
led to enhanced $CO_2$ release due to fires of 0.5-0.8 GtC yr$^{-1}$(see Section 3.8.3). The atmospheric inversions do
simulate a slight reduction of the atmosphere-to-land flux in the northern extra-tropics (north of 30°N) , from
1.6 GtC yr$^{-1}$ in 2022 to 1.4 GtC yr$^{-1}$ in 2023, while the DGVM fail to capture this pattern, with a simulated
northern extra-tropics net atmosphere-to-land flux larger in 2023 than in 2022 (1.0 vs 0.7 GtC yr$^{-1}$).
**3.8.1.2  Net atmosphere-to-ocean flux**
For the 2014-2023 period, the GOBMs ($2.6 \pm 0.4$ GtC yr$^{-1}$) produce a lower estimate for $S_{OCEAN}$ than the $f$CO$_2$-
products with 3.1 [2.9, 3.7] GtC yr$^{-1}$, which shows up in Figure 13 as separate peaks in the distribution from the
GOBMs (dark blue symbols) and from the $f$CO$_2$-products (light blue symbols). Atmospheric inversions (3.1
[2.4, 4.1] GtC yr$^{-1}$) suggest an ocean uptake more in line with the $f$CO$_2$-products for the recent decade (Table 7) ,
although the inversions range includes both the GOBMs and $f$CO$_2$-products estimates (Figure 14 top row) and
the inversions are not fully independent as 6 out of 10 inversions covering the last decade use $f$CO$_2$-products as
ocean priors and one uses a GOBM (Table S4) . The independent constraint based on atmospheric O$_2$ ($3.4 \pm 0.5$
GtC yr$^{-1}$ ) is at the high end of the distribution of the other methods. However, as mentioned in section 2.8, the
O$_2$ method requires a correction for global air-sea O$_2$ flux, which induces a non-negligible uncertainty on the
decadal estimates (about 0.5 GtC yr$^{-1}$). The large growth in the ocean carbon sink from O$_2$ is compatible with
the GOBMs and $f$CO$_2$-products estimates when accounting for their uncertainty ranges. Lastly, the ESMs
estimate, 2.5 [2.2, 2.8] GtC yr$^{-1}$, suggest a moderate ocean carbon sink, comparable to the GOBMs estimate with
regard to mean and spread. We caution that the riverine transport of carbon taken up on land and outgassing
from the ocean, accounted for here, is a substantial ($0.65 \pm 0.3$ GtC yr$^{-1}$) and uncertain term (Crisp et al., 2022;
Gruber et al., 2023; DeVries et al., 2023) that separates the GOBMs, ESMs and oxygen-based estimates on the
one hand from the $f$CO$_2$-products and atmospheric inversions on the other hand.



### 3.8.2 Regional partitioning

Figure 14 shows the latitudinal partitioning of the global atmosphere-to-ocean ($S_{OCEAN}$), atmosphere-to-land ($S_{LAND} - E_{LUC}$), and their sum ($S_{OCEAN} + S_{LAND} - E_{LUC}$) according to the estimates from GOBMs and ocean $f$CO$_2$-products ($S_{OCEAN}$), DGVMs ($S_{LAND} - E_{LUC}$), and from atmospheric inversions ($S_{OCEAN}$ and $S_{LAND} - E_{LUC}$).

#### 3.8.2.1 North

Despite being one of the most densely observed and studied regions of our globe, annual mean carbon sink estimates in the northern extra-tropics (north of 30°N) continue to differ. The atmospheric inversions suggest an atmosphere-to-surface sink ($S_{OCEAN}$+ $S_{LAND} - E_{LUC}$) for 2014-2023 of 2.6 [2.0 to 3.4] GtC yr$^{-1}$, which is slightly higher than the process models' estimate of 2.2 ± 0.4 GtC yr$^{-1}$ (Figure 14). The GOBMs (1.2 ± 0.2 GtC yr$^{-1}$), $f$CO$_2$-products (1.4 [1.3-1.5] GtC yr$^{-1}$), and inversion systems (1.2 [0.9 to 1.4] GtC yr$^{-1}$) produce largely consistent estimates of the ocean sink. However, the larger flux in the $f$CO$_2$-products may be related to data sparsity (Mayot et al., 2024). Thus, the difference mainly arises from the net land flux ($S_{LAND} - E_{LUC}$) estimate, which is 1.0 ± 0.4 GtC yr$^{-1}$ in the DGVMs compared to 1.5 [0.6 to 2.3] GtC yr$^{-1}$ in the atmospheric inversions (Figure 14, second row).

Discrepancies in the northern land fluxes conforms with persistent issues surrounding the quantification of the drivers of the global net land CO$_2$ flux (Arneth et al., 2017; Huntzinger et al., 2017; O'Sullivan et al., 2022) and the distribution of atmosphere-to-land fluxes between the tropics and high northern latitudes (Baccini et al., 2017; Schimel et al., 2015; Stephens et al., 2007; Ciais et al., 2019; Gaubert et al., 2019).

In the northern extra-tropics, the process models, inversions, and $f$CO$_2$-products consistently suggest that most of the interannual variability stems from the land (Figure 14). Inversions generally agree on the magnitude of interannual variations (IAV) over land, more so than DGVMs (0.29-0.32 vs 0.14-0.63 GtC yr$^{-1}$, averaged over 1990-2023).

#### 3.8.2.2 Tropics

In the tropics (30°S-30°N), both the atmospheric inversions and process models estimate a net carbon balance ($S_{OCEAN} + S_{LAND} - E_{LUC}$) that is relatively close to neutral over the past decade (inversions: 0.3 [-0.4, 0.9] GtC yr$^{-1}$, process models: 0.6±0.6 GtC yr$^{-1}$). The GOBMs (-0.03 ± 0.3 GtC yr$^{-1}$), $f$CO$_2$-products (0.3 [0.1, 0.6] GtC yr$^{-1}$), and inversion systems (0.3 [-0.1, 0.8] GtC yr$^{-1}$) indicate a neutral to positive tropical ocean flux (see Figure S1 for spatial patterns). DGVMs indicate a net land sink ($S_{LAND} - E_{LUC}$) of 0.6 ±0.4 GtC yr$^{-1}$, whereas the inversion systems indicate a neutral net land flux although with large model spread (-0.0 [-0.9, 0.8] GtC yr$^{-1}$, (Figure 14, third row).

The tropical lands are the origin of most of the atmospheric CO$_2$ interannual variability (Ahlström et al., 2015), consistently among the process models and inversions (Figure 14). The interannual variability in the tropics is similar among the ocean $f$CO$_2$-products (0.06-0.16 GtC yr$^{-1}$) and the GOBMs (0.07-0.16 GtC yr$^{-1}$, Figure S2). The DGVMs and inversions indicate that atmosphere-to-land CO$_2$ fluxes are more variable than atmosphere-to-





ocean $CO_2$ fluxes in the tropics, with interannual variability of 0.37 to 1.33 and 0.86-0.96 GtC yr$^{-1}$ for DGVMs
and inversions, respectively.

### 3.8.2.3    South

In the southern extra-tropics (south of 30°S), the atmospheric inversions suggest a net atmosphere-to-surface
sink ($S_{OCEAN}$+$S_{LAND}$-$E_{LUC}$) for 2014-2023 of 1.5 [1.2, 1.9] GtC yr$^{-1}$, identical to the process models' estimate of
1.5 ± 0.4 GtC yr$^{-1}$ (Figure 14). An approximately neutral net land flux ($S_{LAND}$-$E_{LUC}$) for the southern extra-
tropics is estimated by both the DGVMs (0.05 ± 0.1 GtC yr$^{-1}$) and the inversion systems (-0.03 [-0.11, 0.08] GtC
yr$^{-1}$). This means nearly all carbon uptake is due to oceanic sinks south of 30°S. The Southern Ocean flux in the
$f$CO$_2$-products (1.5[1.3, 1.7 GtC] yr$^{-1}$) and inversion estimates (1.6 [1.2, 1.9] GtCyr-1) is marginally higher than
in the GOBMs (1.4 ± 0.4 GtC yr$^{-1}$) (Figure 14, bottom row). This agreement is subject to the choice of the river
flux adjustment (Lacroix et al., 2020, Hauck et al., 2023b). Nevertheless, the time-series of atmospheric
inversions and $f$CO$_2$-products diverge from the GOBMs. A substantial overestimation of the trends in the $f$CO$_2$-
products could be explained by sparse and unevenly distributed observations, especially in wintertime (Figure
S1; Hauck et al., 2023a; Gloege et al., 2021). Model biases may contribute as well, with biases in mode water
formation, stratification, and the chemical buffer capacity known to play a role in Earth System Models (Terhaar
et al., 2021, Bourgeois et al., 2022, Terhaar et al., 2022).
The interannual variability in the southern extra-tropics is low because of the dominance of ocean areas with
low variability compared to land areas. The split between land ($S_{LAND}$-$E_{LUC}$) and ocean ($S_{OCEAN}$) shows a
substantial contribution to variability in the south coming from the land, with no consistency between the
DGVMs and the inversions or among inversions. This is expected due to the difficulty of separating exactly the
land and oceanic fluxes when viewed from atmospheric observations alone. The $S_{OCEAN}$ interannual variability
was found to be higher in the $f$CO$_2$-products (0.04-0.20 GtC yr$^{-1}$) compared to GOBMs (0.04 to 0.06 GtC yr$^{-1}$)
in 1990-2023 (Figure S2). Inversions give an interannual variability of 0.10 to 0.13 GtC yr$^{-1}$. Model
subsampling experiments recently illustrated that $f$CO$_2$-products may overestimate decadal variability in the
Southern Ocean carbon sink by 30% and the trend since 2000 by 50-130% due to data sparsity, based on one
and two $f$CO$_2$-products with strong variability (Gloege et al., 2021, Hauck et al., 2023a). The trend benchmark
test using the method of Hauck et al., (2023a) and a subset of 6 $f$CO$_2$-products confirms the sensitivity of the
decadal trends in $f$CO$_2$-products to reconstruction biases, particularly in the Southern Ocean, indicating an
overestimation of the ensemble mean trend. However, we also find compensating positive biases in the
ensemble so that the ensemble mean bias is smaller than the bias from some individual $f$CO$_2$-products.

### 3.8.2.4    RECCAP2 regions

Aligning with the RECCAP-2 initiative (Ciais et al., 2022; Poulter et al., 2022; DeVries et al., 2023), we
provide a breakdown of this GCB paper estimate of the $E_{LUC}$, $S_{LAND}$, Net land ($S_{LAND}$ - $E_{LUC}$), and $S_{OCEAN}$ fluxes
over the 10 land, and 5 ocean RECCAP-2 regions, averaged over the period 2014-2023 (Figure 15). The
DGVMs and inversions suggest a positive net land sink in all regions, except for South America and Africa,





where the inversions indicate a small net source of respectively -0.1 [-0.8, 0.3] GtC yr$^{-1}$ and -0.3 [-0.7, -0.1]
GtC yr$^{-1}$, compared to a small sink of 0.1±0.3 GtC yr$^{-1}$ and 0.3±0.1 GtC yr$^{-1}$ for the DGVMs. However, for
South America, there is substantial uncertainty in both products (ensembles span zero). For the DGVMs, this is
driven by uncertainty in both $S_{LAND}$ (0.5±0.4 GtC yr$^{-1}$) and $E_{LUC}$ (0.4±0.2 GtC yr$^{-1}$). The bookkeeping models
also suggest an $E_{LUC}$ source of around 0.4 GtC yr$^{-1}$ in South America and Africa, in line with the DGVMs
estimates. Bookkeeping models and DGVMs similarly estimate a source of 0.3-0.4 GtC yr$^{-1}$ in Southeast Asia,
with DGVMs suggesting a small net land sink (0.1±0.1 GtC yr$^{-1}$). This is similar to the inversion mean estimate
of a 0.1 [-0.3,0.8] GtC yr$^{-1}$ sink, although the inversion spread is substantial. The inversions suggest the largest
net land sinks are located in North America (0.5 [-0.1,1.0] GtC yr$^{-1}$), Russia (0.6 [0.1,0.9] GtC yr$^{-1}$), and East
Asia (0.4 [-0.2,1.3] GtC yr$^{-1}$). This agrees well with the DGVMs in North America (0.4±0.1 GtC yr$^{-1}$), which
indicate a large natural land sink ($S_{LAND}$) of 0.6±0.2 GtC yr$^{-1}$, being slightly reduced by land-use related carbon
losses (0.2±0.1 GtC yr$^{-1}$). The DGVMs suggest a smaller net land sink in Russia compared to inversions
(0.3±0.2 GtC yr$^{-1}$), and a similar net sink in East Asia (0.2±0.1 GtC yr$^{-1}$).
There is generally a higher level of agreement in the estimates of regional $S_{OCEAN}$ between the different data
streams (GOBMs, $f$CO$_2$-products and atmospheric inversions) on decadal scale, compared to the agreement
between the different land flux estimates. All data streams agree that the largest contribution to $S_{OCEAN}$ stems
from the Southern Ocean due to a combination of high flux density and large surface area, but with important
contributions also from the Atlantic (high flux density) and Pacific (large area) basins. In the Southern Ocean,
GOBMs suggest a sink of 1.0±0.3 GtC yr$^{-1}$, in line with the $f$CO$_2$-products (1.0 [0.8,1.3] GtC yr$^{-1}$) and
atmospheric inversions (1.0 [0.7,1.4] GtC yr$^{-1}$). There is similar agreement in the Pacific ocean, with GOBMs,
$f$CO$_2$-products, and atmospheric inversions indicating a sink of 0.6±0.2 GtC yr$^{-1}$, 0.7 [0.6,1.0] GtC yr$^{-1}$, and
0.6 [0.1,1.0] GtC yr$^{-1}$, respectively. However, in the Atlantic ocean, GOBMs simulate a sink of 0.5±0.1 GtC
yr$^{-1}$, noticeably lower than both the $f$CO$_2$-products (0.8 [0.7,1.0] GtC yr$^{-1}$) and atmospheric inversions (0.7
[0.4,1.1] GtC yr$^{-1}$). It is important to note the $f$CO$_2$-products and atmospheric inversions have a substantial and
uncertain river flux adjustment in the Atlantic ocean (0.3 GtC yr$^{-1}$) that also leads to a mean offset between
GOBMs and $f$CO$_2$-products/inversions in the latitude band of the tropics (Figure 14). The Indian Ocean due its
smaller size and the Arctic Ocean due to its size and sea-ice cover that prevents air-sea gas-exchange are
responsible for smaller but non negligible $S_{OCEAN}$ fluxes (Indian Ocean: (0.3 [0.2,0.3] GtC yr$^{-1}$, 0.3 [0.3,0.4]
GtC yr$^{-1}$, and 0.4 [0.3,0.6] GtC yr$^{-1}$ for GOBMs, $f$CO$_2$-products, and atmospheric inversions, respectively, and
Arctic Ocean: (0.1 [0.1,0.1] GtC yr$^{-1}$, 0.2 [0.1,0.2] GtC yr$^{-1}$, and 0.1 [0.1,0.2] GtC yr$^{-1}$ for GOBMs, $f$CO$_2$-
products, and atmospheric inversions, respectively). Note that the $S_{OCEAN}$ numbers presented here deviate from
numbers reported in RECCAP-2 where the net air-sea CO$_2$ flux is reported (i.e. without river flux adjustment for
$f$CO$_2$-products and inversions, and with river flux adjustment subtracted from GOBMs in most chapters, or
comparing unadjusted datasets with discussion of uncertain regional riverine fluxes as major uncertainty, e.g.
Sarma et al., 2023, DeVries et al., 2023).





#### 3.8.2.5 Tropical vs northern land uptake

A continuing conundrum is the partitioning of the global atmosphere-land flux between the northern hemisphere land and the tropical land (Stephens et al., 2017; Pan et al., 2011; Gaubert et al., 2019). It is of importance because each region has its own history of land-use change, climate drivers, and impact of increasing atmospheric $CO_2$ and nitrogen deposition. Quantifying the magnitude of each sink is a prerequisite to understanding how each individual driver impacts the tropical and mid/high-latitude carbon balance.

We define the North-South (N-S) difference as net atmosphere-land flux north of 30°N minus the net atmosphere-land flux south of 30°N. For the inversions, the N-S difference is 1.50 [0.05,3.0] GtC yr$^{-1}$ across this year's inversion ensemble. An apparent clustering of six satellite-driven solutions towards a common NH land sink noted in GCB2023 is no longer clear.

In the ensemble of DGVMs the N-S difference is 0.4 ± 0.5 GtC yr$^{-1}$, a much narrower range than the one from atmospheric inversions. Only three out of twenty DGVMs have a N-S difference larger than 1.0 GtC yr$^{-1}$, compared to half of the inversion systems simulating a difference at least this large. The smaller spread across DGVMs than across inversions is to be expected as there is no correlation between Northern and Tropical land sinks in the DGVMs as opposed to the inversions where the sum of the two regions being well-constrained by atmospheric observations leads to an anti-correlation between these two regions. This atmospheric N-S gradient could be used as an additional way to evaluate tropical and NH uptake in DGVMs, if their fluxes were combined with multiple transport models. Vice versa, the much smaller spread in the N-S difference between the DGVMs could help to scrutinise the inverse systems further. For example, a large northern land sink and a tropical land source in an inversion would suggest a large sensitivity to $CO_2$ fertilisation (the dominant factor driving the land sinks) for Northern ecosystems, which would be not mirrored by tropical ecosystems. Such a combination could be hard to reconcile with the process understanding gained from the DGVM ensembles and independent measurements (e.g. Free Air $CO_2$ Enrichment experiments).

#### 3.8.3 Fire Emissions in 2024

Fire emissions so far in 2024 have been above the average of recent decades, chiefly due to synchronous large emissions fluxes from North and South America. Figure S9 shows global and regional emissions estimates for the period 1st Jan-30th September in each year 2003-2024. Estimates derive from two global fire emissions products: the global fire emissions database (GFED, version 4.1s; van der Werf et al., 2017), and the global fire assimilation system (GFAS, operated by the Copernicus Atmosphere Service; Kaiser et al., 2012). The two products estimate that global emissions from fires were 1.6-2.2 GtC yr$^{-1}$ during January-September 2024. These estimates are 11-32% above the 2014-2023 average for the same months (1.5-1.7 GtC yr$^{-1}$). In the GFED4.1s product, the year-to-date emissions in 2024 were highest since 2003, exceeding even the large emissions estimate of 2023, whereas the GFAS product showed lower emissions in 2024 than in 2023 and six other years since 2003.



The pattern of high fire emissions from Canada in 2023, which were record-breaking (Jones et al., 2024b, Byrne
et al., 2024), continued into 2024. In January-September 2024, emissions from Canada (0.2-0.3 GtC yr-1) were
half as great as in the same months of 2023 (0.5-0.8 GtC yr$^{-1}$) but still 2.1-2.3 times the average of January-
September periods in 2014-2023 (and 4-6 times greater than the average of those months in 2003-2022
[excluding the record-breaking year in 2023]; Figure S9). The continued anomaly in Canada propagated to the
northern hemisphere, where emissions of 0.5-0.6 GtC yr$^{-1}$ were 26-44% above the average of 2014-2023.
In January-September 2024, fire emissions from South America (0.4-0.6 GtC yr$^{-1}$) were 94-164% above the
average of January-September periods in 2014-2023, marking 2024 out as a year with synchronous high fire
emissions across the Americas. Emissions from Brazil in January-September 2024 (0.2-0.3 GtC yr$^{-1}$) were 91-
118% above the average of January-September periods of 2014-2023 and were at a level not seen since the
major drought year of 2010 (Figure S9; Aragão et al., 2018, Silva Junior et al., 2019). In 2023, deforestation fire
activity in the Brazilian Amazon was below the average levels recorded in national recording systems and
attributed to renewed environmental policy implementation, however the fall in Amazon deforestation fire
activity was largely offset by above-average wildfires related to historic drought (Mataveli et al. 2024).
According to the National Center for Monitoring and Early Warning of Natural Disasters (CEMADEN), drought
conditions continued into 2024 and the current drought is the most intense and widespread Brazil has
experienced since records began in 1950 (CEMADEN, 2024), prompting large wildfires anomalies across the
Amazon, Cerrado and Pantanal regions (INPE, 2024).
Emissions anomalies in Africa strongly influence global totals because the continent typically contributed 41-
47% of global fire emissions during 2014–2023 (average of January-September periods). GFAS suggests that
fire emissions in Africa through September 2024 (0.6 GtC yr$^{-1}$) were slightly below the average of 2014-2023,
whereas GFED4.1s suggests that fire emissions through September 2024 were slightly above the average of
2014-2023 (0.8 GtC yr$^{-1}$).
Tropical fire emissions through September 2024 (1.1-1.6 GtC yr$^{-1}$) accounted for 69-74% of the global total
emissions, which is close to the average of the 2014-2023 period (1.1-1.2 GtC yr$^{-1}$; 72-75%). This marks a
return to a more typical distribution of fire emissions between the tropics and extratropics after the tropical
contribution fell to just 55-59% during January-September 2023 (Figure S9).
We caution that the fire emissions fluxes presented here should not be compared directly with other fluxes of the
budget (e.g. $S_{LAND}$ or $E_{LUC}$) due to incompatibilities between the observable fire emission fluxes and what is
quantified in the $S_{LAND}$ and $E_{LUC}$ components of the budget. The fire emission estimates from global fire
products relate to all fire types that can be observed in Earth Observations (Giglio et al., 2018; Randerson et al.,
2012; Kaiser et al., 2012), including (i) fires occurring as part of natural disturbance-recovery cycles that would
also have occurred in the pre-industrial period (Yue et al., 2016; Keeley and Pausas, 2019; Zou et al., 2019), (ii)
fires occurring above and beyond natural disturbance-recovery cycle due to changes in climate, $CO_2$ and N
fertilisation and to an increased frequency of extreme drought and heatwave events (Abatzoglou et al., 2019;
Jones et al., 2022; Zheng et al., 2021; Burton et al., 2024), and (iii) fires occurring in relation to land use and
land use change, such as deforestation fires and agricultural fires (van der Werf et al., 2010; Magi et al., 2012).





In the context of the global carbon budget, only the portion of fire emissions associated with (ii) should be
included in the $S_{LAND}$ component, and fire emissions associated with (iii) should already be accounted for in the
$E_{LUC}$ component. Emissions associated with (i) should not be included in the global carbon budget. It is not
currently possible to derive specific estimates for fluxes (i), (ii), and (iii) using global fire emission products
such as GFED or GFAS. In addition, the fire emissions estimates from global fire emissions products represent
a gross flux of carbon to the atmosphere, whereas the $S_{LAND}$ component of the budget is a net flux that should
also include post-fire recovery fluxes. Even if emissions from fires of type (ii) could be separated from those of
type (i), these fluxes may be partially or wholly offset in subsequent years by post-fire fluxes as vegetation
recovers, sequestering carbon from the atmosphere to the terrestrial biosphere (Yue et al., 2016; Jones et al.,
2024c). Increases in forest fire emissions and severity (emissions per unit area) from globally during the past
two decades have highlighted the increasing potential for fire emissions fluxes to outweigh post-fire recovery
fluxes, though long-term monitoring of vegetation recovery is required to quantify the net effect on terrestrial C
storage (Jones et al., 2024c).
**3.9    Closing the Global Carbon Cycle**
**3.9.1    Partitioning of Cumulative Emissions and Sink Fluxes**
Emissions during the period 1850-2023 amounted to $710 \pm 70$ GtC and were partitioned among the atmosphere
($285 \pm 5$ GtC; 40%), ocean ($185 \pm 35$ GtC; 26%), and land ($220 \pm 60$ GtC; 32%). The cumulative land sink is
almost equal to the cumulative land-use emissions ($225 \pm 65$ GtC), making the global land nearly neutral over
the whole 1850-2023 period (Figure 3).
The use of nearly independent estimates for the individual terms of the global carbon budget shows a cumulative
budget imbalance of 25 GtC (3% of total emissions) during 1850-2023 (Figure 3, Table 8), which, if correct,
suggests that emissions could be slightly too high by the same proportion or that the combined land and ocean
sinks are slightly underestimated (by about 6%), although these are well within the uncertainty range of each
component of the budget. Nevertheless, part of the imbalance could originate from the estimation of significant
increase in $E_{FOS}$ and $E_{LUC}$ between the mid 1920s and the mid 1960s which is unmatched by a similar growth in
atmospheric $CO_2$ concentration as recorded in ice cores (Figure 3). However, the known loss of additional sink
capacity of 30-40 GtC (over the 1850-2020 period) due to reduced forest cover has not been accounted for in
our method and would exacerbate the budget imbalance (see Section 2.10 and Supplement S.6.4).
For the more recent 1960-2023 period where direct atmospheric $CO_2$ measurements are available, total
emissions ($E_{FOS} + E_{LUC}$) amounted to $500 \pm 50$ GtC, of which $410 \pm 20$ GtC (82%) were caused by fossil $CO_2$
emissions, and $90 \pm 45$ GtC (18%) by land-use change (Table 8). The total emissions were partitioned among
the atmosphere ($220 \pm 5$ GtC; 45%), ocean ($130 \pm 26$ GtC; 25%), and the land ($150 \pm 40$ GtC; 30%), with a near
zero (<1 GtC) unattributed budget imbalance. All components except land-use change emissions have
significantly grown since 1960, with important interannual variability in the growth rate in atmospheric $CO_2$
concentration and in the land $CO_2$ sink (Figure 4), and some decadal variability in all terms (Table 7).
Differences with previous budget releases are documented in Figure S6.





The global carbon budget averaged over the last decade (2014-2023) is shown in Figure 2, Figure 16 (right
panel) and Table 7. For this period, 90% of the total emissions ($E_{FOS}$ + $E_{LUC}$) were from fossil $CO_2$ emissions
($E_{FOS}$), and 10% from land-use change ($E_{LUC}$). The total emissions were partitioned among the atmosphere
(48%), ocean (26%) and land (30%), with a small negative budget imbalance (~4%, 0.4 GtC yr$^{-1}$). For single
years, the budget imbalance can be larger (Figure 4). For 2023, the combination of our estimated sources (11.1 ±
0.9 GtC yr$^{-1}$) and sinks (11.1 ± 0.9 GtC yr$^{-1}$) leads to a $B_{IM}$ of -0.02 GtC, suggesting a near perfect closure of
the global carbon budget.
**3.9.2    Trend and Variability in the Carbon Budget Imbalance**
The carbon budget imbalance ($B_{IM}$; Eq. 1, Figure 4) quantifies the mismatch between the estimated total
emissions and the estimated changes in the atmosphere, land, and ocean reservoirs. The budget imbalance from
1960 to 2023 is very small (0.5 GtC over the period, i.e. <0.01 GtC yr$^{-1}$ on average) and shows no trend over the
full time series (Figure 4e). The process models (GOBMs and DGVMs) and $fCO_2$-products have been selected
to match observational constraints in the 1990s, but no further constraints have been applied to their
representation of trend and variability. Therefore, the near-zero mean and trend in the budget imbalance is seen
as evidence of a coherent community understanding of the emissions and their partitioning on those time scales
(Figure 4). However, the budget imbalance shows substantial variability of the order of ±1 GtC yr$^{-1}$, particularly
over semi-decadal time scales, although most of the variability is within the uncertainty of the estimates. The
positive carbon imbalance during the 1960s, and early 1990s, indicates that either the emissions were
overestimated, or the sinks were underestimated during these periods. The reverse is true for the 1970s, and to a
lesser extent for the 1980s and 2014-2023 period (Figure 4, Table 7).
We cannot attribute the cause of the variability in the budget imbalance with our analysis, we only note that the
budget imbalance is unlikely to be explained by errors or biases in the emissions alone because of its large semi-
decadal variability component, a variability that is atypical of emissions and has not changed in the past 60 years
despite a near tripling in emissions (Figure 4). Errors in $S_{LAND}$ and $S_{OCEAN}$ are more likely to be the main cause
for the budget imbalance, especially on interannual to semi-decadal timescales. For example, underestimation of
the $S_{LAND}$ by DGVMs has been reported following the eruption of Mount Pinatubo in 1991 possibly due to
missing responses to changes in diffuse radiation (Mercado et al., 2009). Although since GCB2021 we
accounted for aerosol effects on solar radiation quantity and quality (diffuse vs direct), most DGVMs only used
the former as input (i.e., total solar radiation) (Table S1). Thus, the ensemble mean may not capture the full
effects of volcanic eruptions, i.e. associated with high light scattering sulphate aerosols, on the land carbon sink
(O'Sullivan et al., 2021). DGVMs are suspected to overestimate the land sink in response to the wet decade of
the 1970s (Sitch et al., 2008). Quasi-decadal variability in the ocean sink has also been reported, with all
methods agreeing on a smaller than expected ocean $CO_2$ sink in the 1990s and a larger than expected sink in the
2000s (Figure 11; Landschützer et al., 2016, DeVries et al., 2019, Hauck et al., 2020, McKinley et al., 2020,
Gruber et al., 2023) and the climate-driven variability could be substantial but is not well constrained (DeVries
et al., 2023, Müller et al., 2023). Errors in sink estimates could also be driven by errors in the climatic forcing
data, particularly precipitation for $S_{LAND}$ and wind for $S_{OCEAN}$. Also, the $B_{IM}$ shows substantial departure from



zero on yearly time scales (Figure 4e), highlighting unresolved variability of the carbon cycle, likely in the land
sink ($S_{LAND}$), given its large year to year variability (Figure 4d and 9).
Both the budget imbalance ($B_{IM}$, Table 7) and the residual land sink from the global budget ($E_{FOS}+E_{LUC}-G_{ATM}-$
$S_{OCEAN}$, Table 5) include an error term due to the inconsistencies that arises from combining $E_{LUC}$ from
bookkeeping models with $S_{LAND}$ from DGVMs, most notably the loss of additional sink capacity (see Section
2.10 and Supplement S.6.4). Other differences include a better accounting of land use changes practices and
processes in bookkeeping models than in DGVMs, or the bookkeeping models error of having present-day
observed carbon densities fixed in the past. That the budget imbalance shows no clear trend towards larger
values over time is an indication that these inconsistencies probably play a minor role compared to other errors
in $S_{LAND}$ or $S_{OCEAN}$.
Although the budget imbalance is near zero for the recent decades, it could be due to a compensation of errors.
We cannot exclude an overestimation of $CO_2$ emissions, particularly from land-use change, given their large
uncertainty, as has been suggested elsewhere (Piao et al., 2018), and/or an underestimate of the sinks. A larger
DGVM estimate of the atmosphere-land $CO_2$ flux ($S_{LAND}-E_{LUC}$) over the extra-tropics would reconcile model
results with inversion estimates for fluxes in the total land during the past decade (Figure 14; Table 5).
Likewise, a larger $S_{OCEAN}$ is also possible given the higher estimates from the $f\!CO_2$-products, inversions and
oxygen based estimates (see Section 3.6.2, Figure 11 and Figure 14), the underestimation of interior ocean
anthropogenic carbon accumulation in the GOBMs (Section 3.6.5, Müller et al., 2023), known biases of ocean
models (e.g., Terhaar et al., 2022; 2024), the role of potential temperature bias and skin effects in $f\!CO_2$-products
(Watson et al., 2020; Dong et al., 2022; Bellenger et al., 2023, Figure 11) and regionally larger estimates based
e.g. on eddy covariance measurements and aircraft data (Dong et al., 2024a; Long et al., 2021; Jin et al., 2024).
More integrated use of observations in the Global Carbon Budget, either on their own or for further constraining
model results, should help resolve some of the budget imbalance (Peters et al., 2017a).
**4    Tracking progress towards mitigation targets**
The average growth in global fossil $CO_2$ emissions peaked at nearly +3% per year during the 2000s, driven by
the rapid growth in emissions in China. In the last decade, however, the global growth rate has slowly declined,
reaching a low +0.6% per year over 2014-2023. While this slowdown in global fossil $CO_2$ emissions growth is
welcome, global fossil $CO_2$ emissions continue to grow, far from the rapid emission decreases needed to be
consistent with the temperature goals of the Paris Agreement.
Since the 1990s, the average growth rate of fossil $CO_2$ emissions has continuously declined across the group of
developed countries of the Organisation for Economic Co-operation and Development (OECD), with emissions
peaking in around 2005 and declining at 1.4% yr$^{-1}$ in the decade 2014-2023, compared to a decline of 0.9% yr$^{-1}$
during the 2004-2013 period (Table 9). In the decade 2014-2023, territorial fossil $CO_2$ emissions decreased
significantly (at the 95% confidence level) in 22 countries/economies whose economies grew significantly (also
at the 95% confidence level): Belgium, Czechia, Denmark, Estonia, Finland, France, Germany, Jordan,
Luxembourg, Netherlands, New Zealand, Norway, Portugal, South Korea, Romania, Slovenia, Somalia, Spain,

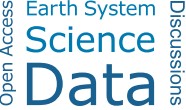

Sweden, Switzerland, United Kingdom, USA (updated from Le Quéré et al., 2019). Altogether, these 22
countries emitted 2.2 GtC yr$^{-1}$ (8.1 GtCO$_2$ yr$^{-1}$) on average over the last decade, about 23% of world CO$_2$ fossil
emissions. For comparison, 18 countries showed a significant decrease in territorial fossil CO$_2$ emissions over
the previous decade (2004-2013).
Decomposing emission changes into the components of growth, a Kaya decomposition, helps give an initial
understanding of the drivers of the changes (Peters et al., 2017b). The reduction in growth in global fossil CO$_2$
emissions in the last decade is due to slightly weaker economic growth, accelerating declines in CO$_2$ emissions
per unit energy, and sustained declines in energy per unit GDP (Figure 17). These trends are a supposition of the
trends at the national level. Fossil CO$_2$ emission declines in the USA and the EU27 are primarily driven by
slightly weaker economic growth since the Global Financial Crisis (GFC) in 2008/2009, sustained declines in
energy per GDP, and sustained declines in CO$_2$ emissions per unit energy with a slight acceleration in the USA
in the last decade. In contrast, fossil CO$_2$ emissions continue to grow in non-OECD countries, although the
growth rate has slowed from 4.9% yr$^{-1}$ during the 2004-2013 decade to 1.8% yr$^{-1}$ in the last decade (Table 9).
Representing 47% of non-OECD emissions in 2023, a large part of this slowdown is due to China, which has
seen emissions growth decline from 7.5% yr$^{-1}$ in the 2004-2013 decade to 1.9% yr$^{-1}$ in the last decade.
Excluding China, non-OECD emissions grew at 3% yr$^{-1}$ in the 2004-2013 decade compared to 1.7% yr$^{-1}$ in the
last decade. China has had weaker economic growth in the 2000s compared to the 2010s, and the rate of
reduction in the energy intensity of economic production has weakened significantly since 2015 with
accelerating declines in CO$_2$ emissions per unit energy (Figure 17). India has had strong economic growth that is
not offset by declines in energy per GDP or declines in CO$_2$ emissions per unit energy, driving up fossil CO$_2$
emissions. Despite the high deployment of renewables in some countries (e.g., China, India), fossil energy
sources continue to grow to meet growing energy demand (Le Quéré et al., 2019). In the rest of the world,
economic growth has slowed considerably in the last decade, but is only partly offset by declines in energy or
carbon intensity, leading to growing emissions.
Globally, fossil CO$_2$ emissions growth is slowing, and this is due in part to the emergence of climate policy
(Eskander and Fankhauser 2020; Le Quere et al 2019) and technological change, which is leading to a shift from
coal to gas and growth in renewable energies, and reduced expansion of coal capacity. At the aggregated global
level, decarbonisation shows a strong and growing signal in the last decade, with smaller contributions from
lower economic growth and declines in energy per GDP (Figure 17). Altogether, global fossil CO$_2$ emissions are
still growing (average of 0.6% per year over the 2014-2023 decade), far from the reductions needed to meet the
ambitious climate goals of the UNFCCC Paris agreement.
Last, we update the remaining carbon budget (RCB) based on two studies, the IPCC AR6 (Canadell et al., 2021)
and the revision of the IPCC AR6 estimates (Forster et al., 2024, Lamboll et al., 2023). We update the RCB
assessed by the IPCC AR6 (Canadell et al., 2021), accounting for the 2020 to 2024 estimated emissions from
fossil fuel combustion (E$_{FOS}$) and land use changes (E$_{LUC}$). From January 2025, the IPCC AR6 RCB (50%
likelihood) for limiting global warming to 1.5°C, 1.7°C and 2°C is estimated to amount to 85, 180, and 315 GtC
(305, 655, 1155 GtCO$_2$). The Forster et al. (2024) study proposed a significantly lower RCB than IPCC AR6,



with the largest reduction being due to an update of the climate emulator (MAGICC) used to estimate the
warming contribution of non-$CO_2$ agents, and to the warming (i.e. emissions) that occurred over the 2020-2023
period. We update the Forster et al., budget accounting for the 2024 estimated emissions from fossil fuel
combustion ($E_{FOS}$) and land use changes ($E_{LUC}$). From January 2025, the Forster et al., (2024) RCB (50%
likelihood) for limiting global warming to 1.5°C, 1.7°C and 2°C is estimated to amount to 45, 140, and 290 GtC
(160, 510, 1060 $GtCO_2$), significantly smaller than the updated IPCC AR6 estimate. Both the original IPCC
AR6 and Forster et al. (2024) estimates include the Earth System uncertainty on the climate response to
cumulative $CO_2$ emissions, which is reflected through the percent likelihood of exceeding the given temperature
threshold, an additional uncertainty of 220$GtCO_2$ due to alternative non-$CO_2$ emission scenarios, and other
sources of uncertainties (see Canadell et al., 2021). The two sets of estimates overlap when considering all
uncertainties. The IPCC AR6 estimates have the advantage of a consensus building approach, while the Forster
et al. (2024) estimates include significant update estimates but without the backing of the IPCC yet.
Here, we take the average of our 2024 update of both IPCC AR6 and Forster et al. (2024) estimates, giving a
remaining carbon (50% likelihood) for limiting global warming to 1.5°C, 1.7°C and 2°C of respectively 65, 160,
and 305 GtC (235, 585, 1110 $GtCO_2$) starting from January 2025. We emphasise the large uncertainties,
particularly when close to the global warming limit of 1.5°C. These 1.5°C, 1.7°C and 2°C remaining carbon
budgets correspond respectively to about 6, 14 and 27 years from the beginning of 2025, at the 2024 level of
total anthropogenic $CO_2$ emissions. Reaching net-zero $CO_2$ emissions by 2050 entails cutting total
anthropogenic $CO_2$ emissions by about 0.4 GtC (1.6 $GtCO_2$), 3.9% of 2024 emissions, each year on average,
comparable to the decrease in $E_{FOS}$ observed in 2020 during the COVID-19 pandemic. However, this would lead
to cumulative emissions over 2025-2050 of 145 GtC (530 $GtCO_2$), well above the remaining carbon budget of
65 GtC to limit global warming to 1.5°C, but still within the remaining budget of 160 GtC to limit warming to
1.7°C (in phase with the "well below 2°C" ambition of the Paris Agreement). Even reaching net zero $CO_2$
globally by 2040, which would require annual emissions cuts of 0.7 GtC (2.5 $GtCO_2$) on average, would still
exceed the remaining carbon budget, with 90 GtC (325 $GtCO_2$) cumulative emissions over 2025-2040, unless
the global emissions trajectory becomes net negative (i.e. more anthropogenic $CO_2$ sinks than emissions) after

1741 2040.

**5    Discussion**
Each year when the global carbon budget is published, each flux component is updated for all previous years to
consider corrections that are the result of further scrutiny and verification of the underlying data in the primary
input datasets. Annual estimates may be updated with improvements in data quality and timeliness (e.g., to
eliminate the need for extrapolation of forcing data such as land-use). Of all terms in the global budget, only the
fossil $CO_2$ emissions and the growth rate in atmospheric $CO_2$ concentration are based primarily on empirical
inputs supporting annual estimates in this carbon budget. The carbon budget imbalance, yet an imperfect
measure, provides a strong indication of the limitations in observations, in understanding and representing
processes in models, and/or in the integration of the carbon budget components.



The persistent unexplained variability in the carbon budget imbalance limits our ability to verify reported
emissions (Peters et al., 2017a) and suggests we do not yet have a complete understanding of the underlying
carbon cycle dynamics on annual to decadal timescales. Resolving most of this unexplained variability should
be possible through different and complementary approaches. First, as intended with our annual updates, the
imbalance as an error term should be reduced by improvements of individual components of the global carbon
budget that follow from improving the underlying data and statistics and by improving the models through the
resolution of some of the key uncertainties detailed in Table 10. Second, additional clues to the origin and
processes responsible for the variability in the budget imbalance could be obtained through a closer scrutiny of
carbon variability in light of other Earth system data (e.g., heat balance, water balance), and the use of a wider
range of biogeochemical observations to better understand the land-ocean partitioning of the carbon imbalance
such as the constraint from atmospheric oxygen included this year. Finally, additional information could also be
obtained through better inclusion of process knowledge at the regional level, and through the introduction of
inferred fluxes such as those based on satellite $xCO_2$ retrievals. The limit of the resolution of the carbon budget
imbalance is yet unclear, but most certainly not yet reached given the possibilities for improvements that lie
ahead.
Estimates of global fossil $CO_2$ emissions from different datasets are in relatively good agreement when the
different system boundaries of these datasets are considered (Andrew, 2020a). But while estimates of $E_{FOS}$ are
derived from reported activity data requiring much fewer complex transformations than some other components
of the budget, uncertainties remain, and one reason for the apparently low variation between datasets is precisely
the reliance on the same underlying reported energy data. The budget excludes some sources of fossil $CO_2$
emissions, which available evidence suggests are relatively small (<1%). We have added emissions from lime
production in China and the US, but these are still absent in most other non-Annex I countries, and before 1990
in other Annex I countries.
Estimates of $E_{LUC}$ suffer from a range of intertwined issues, including the poor quality of historical land-cover
and land-use change maps, the rudimentary representation of management processes in most models, and the
confusion in methodologies and boundary conditions used across methods (e.g., Arneth et al., 2017; Pongratz et
al., 2014, see also Supplement S.6.4 on the loss of sink capacity; Bastos et al., 2021). Uncertainties in current
and historical carbon stocks in soils and vegetation also add uncertainty in the $E_{LUC}$ estimates. Unless a major
effort to resolve these issues is made, little progress is expected in the resolution of $E_{LUC}$. This is particularly
concerning given the growing importance of $E_{LUC}$ for climate mitigation strategies, and the large issues in the
quantification of the cumulative emissions over the historical period that arise from large uncertainties in $E_{LUC}$.
By adding the DGVMs estimates of $CO_2$ fluxes due to environmental change from countries' managed forest
areas (part of $S_{LAND}$ in this budget) to the budget $E_{LUC}$ estimate, we successfully reconciled the large gap
between our $E_{LUC}$ estimate and the land use flux from NGHGIs using the approach described in Grassi et al.
(2021) for future scenarios and in Grassi et al. (2023) using data from the Global Carbon Budget 2021. The
updated data presented here can be used as potential adjustment in the policy context, e.g., to help assess the
collective countries' progress towards the goal of the Paris Agreement and avoiding double-accounting for the



sink in managed forests. In the absence of this adjustment, collective progress would hence appear better than it
is (Grassi et al., 2021). The application of this adjustment is also recommended in the UNFCCC Synthesis
report for the first Global Stocktake (UNFCCC, 2022) whenever a comparison between LULUCF fluxes
reported by countries and the global emission estimates of the IPCC is conducted. However, this adjustment
should be seen as a short-term and pragmatic fix based on existing data, rather than a definitive solution to
bridge the differences between global models and national inventories. Additional steps are needed to
understand and reconcile the remaining differences, some of which are relevant at the country level (Grassi, et
al., 2023, Schwingshackl, et al., 2022).
The comparison of GOBMs, $f$CO$_2$-products, and inversions highlights substantial discrepancy in the temporal
evolution of S$_{OCEAN}$ in the Southern Ocean and northern high-latitudes (Figure 14, Hauck et al., 2023a) and in
the mean S$_{OCEAN}$ in the tropics. A large part of the uncertainty in the mean fluxes stems from the regional
distribution of the river flux adjustment term. The current distribution simulates the largest share of the
outgassing to occur in the tropics (Lacroix et al., 2020). The long-standing sparse data coverage of $f$CO$_2$
observations in the Southern compared to the Northern Hemisphere (e.g., Takahashi et al., 2009) continues to
exist (Bakker et al., 2016, 2024, Figure S1) and to lead to substantially higher uncertainty in the S$_{OCEAN}$ estimate
for the Southern Hemisphere (Watson et al., 2020, Gloege et al., 2021, Hauck et al., 2023a). This discrepancy,
which also hampers model improvement, points to the need for increased high-quality $f$CO$_2$ observations
especially in the Southern Ocean. At the same time, model uncertainty is illustrated by the large spread of
individual GOBM estimates (indicated by shading in Figure 14) and highlights the need for model
improvement. The issue of diverging trends in S$_{OCEAN}$ from different methods is smaller this year as the trend in
the $f$CO$_2$-products was revised downwards with the data available in this GCB release, but remains a matter of
concern. Recent and on-going work suggests that the $f$CO$_2$-products may overestimate the trend (Hauck et al.,
2023a, Supplement section S3.4), though the full $f$CO$_2$-product ensemble remains to be tested. A data-
constrained model approach suggests that the GOBMs underestimate the amplitude of decadal variability, but
that the $f$CO$_2$-products overestimate the trend (Mayot et al., 2024). At the same time, evidence is accumulating
that GOBMs likely underestimate the mean flux (Section 3.6.2, Terhaar et al., 2022, DeVries et al., 2023,
Müller et al., 2023, Dong et al., 2024). The independent constraint from atmospheric oxygen measurements
gives a larger sink for the past decade and a steeper trend. However, the estimate is consistent within
uncertainties with S$_{OCEAN}$, with the relatively larger ocean sink in the $f$CO$_2$-products and some of the GOBMs.
The assessment of the net land-atmosphere exchange from DGVMs and atmospheric inversions also shows
substantial discrepancy, particularly for the estimate of the net land flux over the northern extra-tropic. This
discrepancy highlights the difficulty to quantify complex processes (CO$_2$ fertilisation, nitrogen deposition and
fertilisers, climate change and variability, land management, etc.) that collectively determine the net land CO$_2$
flux. Resolving the differences in the Northern Hemisphere land sink will require the consideration and
inclusion of larger volumes of observations.
We provide metrics for the evaluation of the ocean and land models and the atmospheric inversions (Figures S2
to S4, Table S11). These metrics expand the use of observations in the global carbon budget, helping 1) to
support improvements in the ocean and land carbon models that produce the sink estimates, and 2) to constrain




the representation of key underlying processes in the models and to allocate the regional partitioning of the $CO_2$
fluxes. The introduction of process-based metrics targeted to evaluate the simulation of $S_{OCEAN}$ in the ocean
biogeochemistry models is an important addition to the evaluation based on ocean carbon observations. This is
an initial step towards the introduction of a broader range of observations and more stringent model evaluation
that we hope will support continued improvements in the annual estimates of the global carbon budget.
We assessed before that a sustained decrease of –1% in global emissions could be detected at the 66%
likelihood level after a decade only (Peters et al., 2017a). Similarly, a change in behaviour of the land and/or
ocean carbon sink would take as long to detect, and much longer if it emerges more slowly. To continue
reducing the carbon imbalance on annual to decadal time scales, regionalising the carbon budget, and integrating
multiple variables are powerful ways to shorten the detection limit and ensure the research community can
rapidly identify issues of concern in the evolution of the global carbon cycle under the current rapid and
unprecedented changing environmental conditions.
**6    Conclusions**
The estimation of global $CO_2$ emissions and sinks is a major effort by the carbon cycle research community that
requires a careful compilation and synthesis of measurements, statistical estimates, and model results. The
delivery of an annual carbon budget serves two purposes. First, there is a large demand for up-to-date
information on the state of the anthropogenic perturbation of the climate system and its underpinning causes. A
broad stakeholder community relies on the datasets associated with the annual carbon budget including
scientists, policy makers, businesses, journalists, and non-governmental organisations engaged in adapting to
and mitigating human-driven climate change. Second, over the last decades we have seen unprecedented
changes in the human and biophysical environments (e.g., changes in the growth of fossil fuel emissions, impact
of COVID-19 pandemic, Earth's warming, and strength of the carbon sinks), which call for frequent
assessments of the state of the planet, a better quantification of the causes of changes in the contemporary global
carbon cycle, and an improved capacity to anticipate its evolution in the future. Building this scientific
understanding to meet the extraordinary climate mitigation challenge requires frequent, robust, transparent, and
traceable datasets and methods that can be scrutinised and replicated. This paper via 'living data' helps to keep
track of new budget updates.
**7    Data availability**
The data presented here are made available in the belief that their wide dissemination will lead to greater
understanding and new scientific insights of how the carbon cycle works, how humans are altering it, and how
we can mitigate the resulting human-driven climate change. Full contact details and information on how to cite
the data shown here are given at the top of each page in the accompanying database and summarised in Table 2.
The accompanying database includes three Excel files organised in the following spreadsheets:
File Global_Carbon_Budget_2024v1.0.xlsx includes the following:



1.   Summary
2.   The global carbon budget (1959-2023);
3.   The historical global carbon budget (1750-2023);
4.   Global $CO_2$ emissions from fossil fuels and cement production by fuel type, and the per-capita emissions
1864        (1850-2023);

5.   $CO_2$ emissions from land-use change from the individual bookkeeping models (1959-2023);
6.   Ocean $CO_2$ sink from the individual global ocean biogeochemistry models and $fCO_2$-products (1959-
1867        2023);

7.   Terrestrial $CO_2$ sink from the individual DGVMs (1959-2023);
8.   Cement carbonation $CO_2$ sink (1959-2023).
File National_Fossil_Carbon_Emissions_2024v1.0.xlsx includes the following:
1.   Summary
2.   Territorial country $CO_2$ emissions from fossil fuels and cement production (1850-2023);
3.   Consumption country $CO_2$ emissions from fossil fuels and cement production and emissions transfer from
1874        the international trade of goods and services (1990-2020) using CDIAC/UNFCCC data as reference;

4.   Emissions transfers (Consumption minus territorial emissions; 1990-2020);
5.   Country definitions.
File National_LandUseChange_Carbon_Emissions_2024v1.0.xlsx includes the following:
1.   Summary
2.   Territorial country $CO_2$ emissions from Land Use Change (1850-2023) from three bookkeeping models;
All three spreadsheets are published by the Integrated Carbon Observation System (ICOS) Carbon Portal and
are available at https://doi.org/10.18160/GCP-2024 (Friedlingstein et al., 2024). National emissions data are also
available at https://doi.org/10.5281/zenodo.13981696 (Andrew and Peters, 2024), from the Global Carbon Atlas
(http://www.globalcarbonatlas.org/, last access: 28 October 2024) and from Our World in Data
(https://ourworldindata.org/co2-emissions, last access: 28 October 2024).

**Author contributions**

PF, MO, MWJ, RMA, JH, PL, CLQ, HL, ITL, AO, GPP, WP, JP, CS, and SSi designed the study, conducted the analysis, and wrote the paper with input from JGC, PCi and RBJ. RMA, GPP and JIK produced the fossil $CO_2$ emissions and their uncertainties and analysed the emissions data. MH and GMa provided fossil fuel emission data. JP, TGa, ZQ, and CS provided the bookkeeping land-use change emissions with synthesis by JP and CS. SSm provided the estimates of non-vegetation CDR fluxes. LB, MC, ÖG, NG, TI, TJ, LR, JS, RS, and HTs provided an update of the global ocean biogeochemical models; LMD, ARF, DJF, MG, LG, YI, AJ, CR, AR, JZ, and PC provided an update of the ocean $f$CO2-data products, with synthesis on both streams by JH, PL and NMa. SRA, NRB, MB, CFB, HCB, KC, KE, WE, RAF, TGk, SKL, NL, NMe, NMM, SN, LO, TO, DP, AJS, ST, BT, CN, and RW provided ocean $fCO_2$ measurements for the year 2023, with synthesis by AO and TS. AA, VA, PCa, THC, JD, CDR, AF, JHe, AKJ, EK, JK, PCM, LM, TN, MO, QS, HTi, XYa, WY, XYu, and SZ provided an update of the Dynamic Global Vegetation Models, with synthesis by SSi and MO. HL, RSA, OT, and ET provided estimates of land and ocean sinks from Earth System Models, as well as a projection of the atmospheric growth rate for 2024. NC, FC, ARJ, FJ, ZJ, JL, SM, YN, PIP, CR, DY, and NZ provided an updated atmospheric inversion, WP, FC, and ITL developed the protocol and produced the synthesis and evaluation of the atmospheric inversions. RMA provided projections of the 2024 fossil emissions and atmospheric CO2 growth rate. PL provided the predictions of the 2024 ocean and land sinks. LPC, GCH, KKG, TMR, GRvdW, WX, and ZY provided forcing data for land-use change. FT and GG provided data for the land-use change NGHGI harmonisation. RFK provided key atmospheric CO2 data. EJM and RFK provided the atmospheric oxygen constraint on surface net carbon sinks. MWJ provided the historical atmospheric CO2 concentration and growth rate. MO and NB produced the aerosol diffuse radiative forcing for the DGVMs. IH provided the climate forcing data for the DGVMs. PCM provided the evaluation of the DGVMs. MWJ provided the emissions prior for use in the inversion systems. XD provided seasonal emissions data for most recent years for the emission prior. PF, MO and MWJ coordinated the effort, revised all figures, tables, text and numbers to ensure the update was clear from the 2023 edition and in line with the globalcarbonatlas.org.

**Competing interests.**

At least one of the (co-)authors is a member of the editorial board of Earth System Science Data






**Acknowledgements**
We thank all people and institutions who provided the data used in this global carbon budget 2024 and the Global
Carbon Project members for their input throughout the development of this publication. We thank Nigel Hawtin
for producing Figure 2 and Figure 15. We thank Alex Vermeulen and Hannah Ritchie for respectively hosting the
Global Carbon Budget datasets on the ICOS portal and the Our World in Data website. We thank Ian G. C. Ashton,
Dorothee Bakker, Sebastian Brune, Fatemeh Cheginig, Emeric Claudel, Lushanya Dayathilake, Christian Ethé,
Lonneke Goddijn-Murphy, T. Holding, Fabrice Lacroix, Yi Liu, Damian Loher, Naiqing Pan, Paridhi Rustogi, J.
D. Shutler, Richard Sims, Phillip Townsend, Jing Wang, Andrew J. Watson, Kristina Frölich, Zoé Lloret, Adrien
Martinez, Lorna Nayagam, Rajesh Janardanan, Yakun Zhu, Ram Alkama, Simone Rossi, Stefanie Falk , Pengyue
Du, Peter Lawrence, Sean Swenson, Daniel Kennedy, Sam Levis, Erik Kluzek, Lachlan Whyborn, Drew
Holzworth, Ian Harman, Dr. Shufen Pan, Jason Cole, Victoria Spada, Vladimir Lapin, Raffaele Bernardello, K.
Toyama, H. Nakano, S. L. Urakawa, Anthony P. Walker, Tristan Quaife, and David K. Woolf for their
involvement in the development, use and analysis of the models and data-products used here. We thank Kim
Currie, Siyabulela Hamnca, Mutshutshu Tsanwani, Pedro Monteiro, C. Lo Monaco, and Arne Körtzinger who
contributed to the provision of surface ocean $CO_2$ observations for the year 2023 (see Table S7). We also thank
Stephen D. Jones of the Ocean Thematic Centre of the EU Integrated Carbon Observation System (ICOS)
Research Infrastructure, Eugene Burger of NOAA's Pacific Marine Environmental Laboratory and Alex Kozyr of
NOAA's National Centers for Environmental Information, for their contribution to surface ocean $CO_2$ data and
metadata management. We thank the scientists, institutions, and funding agencies responsible for the collection
and quality control of the data in SOCAT as well as the International Ocean Carbon Coordination Project
(IOCCP), the Surface Ocean Lower Atmosphere Study (SOLAS) and the Integrated Marine Biosphere Research
(IMBeR) program for their support. We thank Nadine Goris and Lavinia Patara for support in calculating
observational ocean evaluation metrics. We thank Fortunat Joos, Samar Khatiwala and Timothy DeVries for
providing historical atmospheric and ocean data. We thank data providers ObsPack GLOBALVIEWplus v9.0 and
NRT v9.2 for atmospheric $CO_2$ observations. Ingrid T Luijkx and Wouter Peters thank the CarbonTracker Europe
team at Wageningen University, including Remco de Kok, Joram Hooghiem, Linda Kooijmans and Auke van der
Woude. Ian Harris thanks the Japan Meteorological Agency (JMA) for producing the Japanese 55-year Reanalysis
(JRA-55). Reinel Sospedra-Alfonso thanks William J. Merryfield, Woosung Lee, Jason Cole, and Victoria Spada



for their help to set up and produce CanESM5 runs. Olivier Torres thanks Patricia Cadule, Juliette Mignot, Didier
Swingedouw, and Laurent Bopp for contributions to the IPSL-CM6-LR-ESMCO2 simulations. Yosuke Niwa
thanks CSIRO, EC, EMPA, FMI, IPEN, JMA, LSCE, NCAR, NIES, NILU, NIWA, NOAA, SIO, and TU/NIPR
for providing data for NISMON-CO2. Zhe Jin thanks Xiangjun Tian, Yilong Wang, Hongqin Zhang, Min Zhao,
Tao Wang, Jinzhi Ding and Shilong Piao for their contributions to the GONGGA inversion system. Paul I. Palmer
thanks Lian Fang and acknowledges ongoing support from the National Centre for Earth Observation. Ning Zeng
thanks Zhiqiang Liu, Yun Liu, Eugenia Kalnay, and Gassem Asrar for their contributions to the COLA system.
Fei Jiang acknowledges the High-Performance Computing Center (HPCC) of Nanjing University for doing the
inversions on its blade cluster system, and thanks Weimin Ju for updating the a priori fluxes of the terrestrial
ecosystems. Meike Becker and Are Olsen thank Sparebanken Vest / Agenda Vestlandet for their support for the
observations on the Statsraad Lehmkuhl. Wiley Evans and Katie Campbell thank the Tula Foundation for funding
support. Thanos Gkritzalis and the VLIZ ICOS team are thankful to the crew of the research vessel Simon Stevin
for all the support and help they provide. Bronte Tilbrook and Craig Neill thank Australia's Integrated Marine
Observing System (IMOS) for sourcing CO2 data. FAOSTAT is funded by FAO member states through their
contributions to the FAO Regular Programme, data contributions by national experts are greatly acknowledged.
The views expressed in this paper are the authors' only and do not necessarily reflect those of FAO. Finally, we
thank all funders who have supported the individual and joint contributions to this work (see details below), as
well as the two anonymous reviewers of this manuscript, and the many researchers who have provided feedback.
**Financial and computing support**
This research was supported by the following sources of funding: The Argentinian-Uruguayan Joint Technical
Commission of the Maritime Front (Comisión Técnica Mixta del Frente Marítimo, CTMFM) (Argentina);
Instituto Nacional de Investigación y Dessarrollo Pesquero (Argentina); Australia's Integrated Marine Observing
System (IMOS) which is enabled by the National Collaborative Research Infrastructure Strategy (NCRIS)
(Australia); Australian Earth-System Simulator National Research Infrastructure (ACCESS-NRI) (Australia);
Australian National Environmental Science Program, Climate Systems Hub (Australia); Research Foundation
Flanders (ICOS Flanders, grant no. I001821N) (Belgium); Tula Foundation (Canada); National Key Research and
Development Program (grant no. 2023YFF0805400) (China); National Key Research and Development Program
(grant no. 2023YFB3907404) (China); Jiangsu Provincial Science Fund for Distinguished Young Scholars (Grant
No: BK20231530) (China); National Natural Science Foundation (grant no. 42141020) (China); Carbon
Neutrality and Energy System Transformation (CNEST) Program led by Tsinghua University, granted by National
Key R&D Program of China (Grant No. 2023YFE0113000) (China); Second Tibetan Plateau Scientific
Expedition and Research Program (Grant: 2022QZKK0101) (China); CAS Project for Young Scientists in Basic



Research, Grant No.YSBR-037 (China); grant no. 2021YFD2200405 (China); Copernicus Atmosphere Monitoring Service, implemented by ECMWF (Grant: CAMS2 55) (European Commission); Copernicus Marine Environment Monitoring Service, implemented by MOi (Grant: CMEMS-TAC-MOB) (European Commission); H2020 4C (grant no. 821003) (European Commission); H2020 ESM2025 – Earth System Models for the Future (grant no. 101003536) (European Commission); H2020 GEORGE (grant no. 101094716) (European Commission); Horizon Europe (EYE-CLIMA: grant no. 101081395) (European Commission); ERC-2022-STG OceanPeak (Grant 101077209) (European Commission); Horizon Europe Grant 101083922 (OceanICU Improving Carbon Understanding) (European Commission); Horizon 2020 research and innovation programme under Grant Agreements N° 101056939 (RESCUE project) (European Commission); COMFORT project (grant no. 820989) (European Commission); Climate Space RECCAP-2 (European Space Agency); Ocean Carbon for Climate (European Space Agency); EO-LINCS (European Space Agency); OceanSODA project, grant no. 4000112091/14/I-LG (European Space Agency); Institut National des Sciences de l'Univers (INSU) (France); Institut Polaire français, Paul-Emile Victor (IPEV) (France); Observatoire des sciences de l'univers Ecce-Terra (OSU at Sorbonne Université) (France); Institut de recherché français sur les ressources marines (IFREMER) (France); French Oceanographic Fleet (FOF) (France); ICOS-France (France); Institut de Recherche pour le Développement (IRD) (France); Agence Nationale de la Recherche - France 2030 (PEPR TRACCS programme under grant number ANR-22-EXTR-0009) (France); Institut de l'Océan and the Institut des Sciences du Calcul et des Données of Sorbonne University (IDEX SUPER 11-IDEX-0004, project-team FORMAL) (France); Federal Ministry of Education and Research, collaborative project C-SCOPE (project no. 03F0877A) (Germany); Helmholtz Association ATMO programme (Germany); Initiative and Networking Fund of the Helmholtz Association (grant no. VH-NG-19-33) (Germany); ICOS Germany (Germany); German Federal Ministry of Education and Research (BMBF), project STEPSEC (grant no. 01LS2102A) (Germany); Helmholtz Young Investigator Group Marine Carbon and Ecosystem Feedbacks in the Earth System (MarESys), grant number VH-NG-1301 (Germany); Deutsche Forschungsgemeinschaft (DFG) under Germany's Excellence Strategy – EXC 2037 'Climate, Climatic Change, and Society' – Project Number: 390683824 (Germany); Arctic Challenge for Sustainability phase II project (ArCS-II; grant no. JP-MXD1420318865) (Japan); Environment Research and Technology Development Fund (grant no. JPMEERF24S12206) (Japan); CREST, Japan Science and Technology Agency (grant no. JPMJCR23J4) (Japan); Global Environmental Research Coordination System, Ministry of the Environment (grant no. E2252) (Japan); Meteorological Agency (Japan); Ministry of Education, Culture, Sports, Science and Technology, MEXT program for the advanced studies of climate change projection (SENTAN) (grant numbers JPMXD0722680395 and JPMXD0722681344) (Japan); Meteorological Research Institute and the Environment Research and Technology Development Fund (JPMEERF24S12200) (Japan); NIES GOSAT project (Japan); Research Council of Norway (N-ICOS-2, grant no. 296012) (Norway); Research Council of Norway (grant no. 270061) (Norway); Swiss National Science Foundation (grant no. 200020-200511) (Switzerland); Natural Environmental Research Council, National Centre for Earth Observation (NE/R016518/1) (UK); Natural Environment Research Council, UK EO Climate Information Service (NE/X019071/1) (UK); Natural Environment Research Council (NE/V01417X/1) (UK); Natural Environment Research Council, National Centre for Atmospheric Science (UK); Natural Environment Research Council (NE/Y005260/1) (UK); UK Royal Society (grant no. RP\R1\191063) (UK); Natural Environment Research Council (Grant Ref: NE/V013106/1) (UK);





National Center for Atmospheric Research (NSF Cooperative Agreement No. 1852977) (USA); NOAA Ocean
Acidification Program (grant no. 100018228) (USA); NOAA Global Ocean Monitoring and Observing Program
(grant no. 100018302) (USA); NOAA cooperative agreement NA22OAR4320151 (USA); NOAA cooperative
agreement NA20OAR4310340 (USA); NOAA (grant no. 1305M322PNRMJ0338); NASA (grant no.
80NSSC22K0150); NASA (grant no. 80NM0018D0004); National Science Foundation (NSF-2019625) (USA);
National Science Foundation (NSF-831361857) (USA); NASA Terrestrial Ecology Program (USA); NASA
Carbon Monitoring System program (80NSSC21K1059) (USA); NASA Land Cover and Land Use Change
Program (80NSSC24K0920) (USA); National Science Foundation (NSF-1903722) (USA); National Science
Foundation (NSF-1852977) (USA); The U.S. Department of Energy, Office of Science, Office of Biological and
Environmental Research (USA); The Department for Education SciDac (grant number: DESC0012972) (USA);
IDS (grant number: 80NSSC17K0348) (USA); Schmidt Sciences, LLC (USA).
We also acknowledge support from the following computing facilities: The Australian Earth-System Simulator
National Research Infrastructure (ACCESS-NRI) (Australia); Deutsches Klimarechenzentrum (DKRZ) granted
by its Scientific Steering Committee (WLA) under project ID bm0891 (Germany); HPC cluster Aether at the
University of Bremen, financed by DFG within the scope of the Excellence Initiative (Germany); HPC resources
of TGCC under the allocation A0150102201 awarded by GENCI and of CCRT under the Grant CCRT2024-
p24cheva awarded by CEA/DRF (France); HPC resources of Meteo-France (France); NIES supercomputer system
(Japan); UNINETT Sigma2, National Infrastructure for High Performance Computing and Data Storage in
Norway (NN2980K/NS2980K) (Norway); UK CEDA JASMIN supercomputer (UK); UEA (University of East
Anglia) high performance computing cluster (UK); Derecho supercomputer (doi:10.5065/D6RX99HX), provided
by the Computational and Information Systems Laboratory (CISL) at NCAR (USA); Oak Ridge Leadership
Computing Facility at the Oak Ridge National Laboratory (USA); ISAM simulations were performed using
Cheyenne, NCAR HPC resources managed by CISL (doi:10.5065/D6RX99HX) (USA).



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



**Tables**
**Table 1.** Factors used to convert carbon in various units (by convention, Unit 1 = Unit 2 × conversion).

| Unit 1 | Unit 2 | Conversion | Source |
|---|---|---|---|
| GtC (gigatonnes of carbon) | ppm (parts per million) (a) | 2.124 (b) | Ballantyne et al. (2012) |
| GtC (gigatonnes of carbon) | PgC (petagrams of carbon) | 1 | SI unit conversion |
| GtCO2 (gigatonnes of carbon dioxide) | GtC (gigatonnes of carbon) | 3.664 | 44.01/12.011 in mass equivalent |
| (a) Measurements of atmospheric CO2 concentration have units of dry-air mole fraction. 'ppm' is an abbreviation for micromole/mol, dry air. | | | |
| (b) The use of a factor of 2.124 assumes that all the atmosphere is well mixed within one year. In reality, only the troposphere is well mixed and the growth rate of CO2 concentration in the less well-mixed stratosphere is not measured by sites from the NOAA network. Using a factor of 2.124 makes the approximation that the growth rate of CO2 concentration in the stratosphere equals that of the troposphere on a yearly basis. | | | |






**Table 2.** How to cite the individual components of the global carbon budget presented here.

| Component | Primary reference |
|---|---|
| Global fossil CO2 emissions (EFOS), total and by fuel type | Andrew and Peters (2024) |
| National territorial fossil CO2 emissions (EFOS) | Hefner and Marland (2023), UNFCCC (2024) |
| National consumption-based fossil CO2 emissions (EFOS) by country (consumption) | Peters et al. (2011a) updated as described in this paper |
| Net land-use change flux (ELUC) | This paper (see Table 4 for individual model references). |
| Growth rate in atmospheric CO2 concentration (GATM) | Lan et al. (2024) |
| Ocean and land CO2 sinks (SOCEAN and SLAND) | This paper (see Table 4 for individual model and data products references). |




**Table 3.** Main methodological changes in the global carbon budget since 2020. Methodological changes introduced in one year are kept for the following years unless noted. Empty cells mean there were no methodological changes introduced that year. Table S9 lists methodological changes from the first global carbon budget publication up to 2019.

| Publication year | Fossil fuel emissions | | LUC emissions | Reservoirs | | | Other changes |
|---|---|---|---|---|---|---|---|
| | Global | Country (territorial) | | Atmosphere | Ocean | Land | |
| 2020<br><br>Friedlingstein et al. (2020) GCB2020 | Cement carbonation now included in the EFOS estimate, reducing EFOS by about 0.2GtC yr-1 for the last decade | India's emissions from Andrew (2020: India); Corrections to Netherland Antilles and Aruba and Soviet emissions before 1950 as per Andrew (2020: CO2); China's coal emissions in 2019 derived from official statistics, emissions now shown for EU27 instead of EU28. Projection for 2020 based on assessment of four approaches. | Average of three bookkeeping models; use of 17 DGVMs. Estimate of gross land use sources and sinks provided | Use of six atmospheric inversions | Based on nine models. River flux revised and partitioned NH, Tropics, SH | Based on 17 models | |
| 2021<br><br>Friedlingstein et al. (2022a) GCB2021 | Projections are no longer an assessment of four approaches. | Official data included for a number of additional countries, new estimates for South Korea, added emissions from lime production in China. | ELUC estimate compared to the estimates adopted in national GHG inventories | | Average of means of eight models and means of seven data-products. Current year prediction of SOCEAN using a feed-forward neural network method | Current year prediction of SLAND using a feed-forward neural network method | |
| 2022<br><br>Friedlingstein et al. (2022) GCB2022 | | | ELUC provided at country level. Revised components decomposition of ELUC fluxes. Revision of LUC | Use of nine atmospheric inversions | Average of means of ten models and means of seven data-products | Based on 16 models. Revision of LUC maps for Brazil. | |





| | | | maps for Brazil. New datasets for peat drainage. | | | | |
|---|---|---|---|---|---|---|---|
| 2023<br><br>Friedlingstein et al. (2023) GCB2023 | | | Refined components decomposition of ELUC. Revision of LUC maps for Indonesia. Use of updated peat drainage estimates. | Use of 14 atmospheric inversions. Additional use of 4 Earth System Models to estimate current year CO2 | Additional use of 4 Earth System Models and atmospheric oxygen method to assess SOCEAN. Regional distribution of river flux adjustment revised. | Based on 20 models. Additional use of 4 Earth System Models and atmospheric oxygen method to assess the net atmosphere-land flux. | Inclusion of an estimate of Carbon Dioxide Removal |
| 2024<br><br>This study | Inclusion of 2024 projections from Carbon Monitor | Inclusion of 2024 projections from Carbon Monitor for China, USA, EU27, India, and Rest of the World | Fourth bookkeeping estimate (LUCE). Update in land-use data (HYDE3.4) including revision of LUC maps for China. Updated definition of forest (re-)growth fluxes (consistent with 2nd State of CDR Report). | Use of 14 atmospheric inversions models | Use of 10 GOBMs, 8 fCO2-products. Added evaluation for fCO2-products. | Use of 20 DGVMs | |



**Table 4.** References for the process models, bookkeeping models, ocean data products, and atmospheric inversions. All models and products are updated with new data to the end of year 2023.

| Model/data name | Reference | Change from Global Carbon Budget 2023 (Friedlingstein et al., 2023) |
|---|---|---|
| **_Bookkeeping models for land-use change emissions_** | | |
| BLUE | Hansis et al. (2015) | No change to model, but simulations performed with LUH2-GCB2024 forcing. Update in added peat drainage emissions. |
| H&C2023 | Houghton and Castanho (2023) | No change to model. Data for years after last modelled year (2020) extrapolated based on anomalies in deforestation fires from GFED. Update in added peat drainage emissions. |
| OSCAR | Gasser et al. (2020) | No change to model, but land-use forcing changed to LUH2-GCB2024 and FRA2020 extrapolated to 2023. Constraining based on GCB2023 data for SLAND over 1960-2022. Update in added peat drainage emissions. |
| LUCE | Qin et al. (2024) | New model in GCB2024. |
| **_Dynamic global vegetation models_** | | |
| CABLE-POP | Haverd et al. (2018) | Bug fix applied to land use change calculations enabling variable crop and pasture fractions; corrections to the pre-industrial primary forest fraction in Europe; minor parameter changes |
| CLASSIC | Melton et al. (2020), Asaadi et al. (2018) | Permeable soil depth reduced to 4 m ; 15 soil layers in the top 4 m permeable soil and 5 bed rock layers from 4 m to 62 m; saturated hydraulic conductivity decreases with depth in the permeable soil layers; transpiration occurs from a partially-wet canopy leaves. These changes yield better runoff seasonality and a more realistic partitioning of precipitation into runoff and evapotranspiration. |
| CLM6.0 | Lawrence et al. (2019) | Updates to surface datasets; improvement of roughness length calculation; updates to snow optical properties and snow thermal conductivity; improved excess ice; improved simulation of burial of vegetation by snow; urban updates, including transient urban, urban properties, and air conditioning; improvements to biomass heat storage; tillage and residue removal; crop penology and planting dates; improvement to irrigation methods; PFT parameter update. |





| | | |
|---|---|---|
| DLEM | Tian et al. (2015), You et al. (2022) | Incorporate mechanistic representations of dynamic crop growth and development processes, such as crop-specific phenological development, carbon allocation, yield formation, and biological N fixation. Agricultural management practices such as N fertiliser use, irrigation, tillage, manure application, dynamic crop rotation, cover cropping, and genetic improvements are also included (You et al. 2022). |
| EDv3 | Moorcroft et al. (2001), Ma et al. (2022) | Minor bug fixes, updated fire submodule |
| ELM | Yang et al.(2023), Burrows et al.(2020) | No change |
| IBIS | Xia et al., (2024) | Improved algorithm of leaf area index |
| iMAPLE | Yue et al. (2024) | The updated version of YIBs model with dynamic coupling between carbon and water cycles. |
| ISAM | Jain et al. (2013), Meiyappan et al. (2015), Shu et al. (2020) | Vertically resolved soil biogeochemistry (carbon and nitrogen) module, following Shu et al. (2020), |
| ISBA-CTRIP | Delire et al. (2020) | No change. |
| JSBACH | Mauritsen et al. (2019), Reick et al. (2021) | Minor bug fixes in post-processing |
| JULES-ES | Wiltshire et al. (2021), Sellar et al. (2019), Burton et al. (2019) | Minor bug fixes. (Using JULES v7.4) |
| LPJ-GUESS | Smith et al. (2014) | No change. |
| LPJml | Schaphoff et al., 2018, von Bloh et al., 2018, Lutz et al., 2019 (tillage), Heinke et al., 2023 (livestock grazing) | No change |
| LPJwsl | Poulter et al. (2011) (d) | Minor bug fixes, weighting of fire carbon by GFED to simulate annual cycle |
| LPX-Bern | Lienert and Joos (2018) | No change. |
| OCN | Zaehle and Friend (2010), Zaehle et al. (2011) | No change. |
| ORCHIDEEv3 | Krinner et al. (2005), Zaehle and Friend (2010), Vuichard et al. (2019) | No change. |
| SDGVM | Woodward and Lomas (2004), Walker et al. (2017) | Parameter adjustment for reducing evaporation from vegetation that intercepted precipitation, as well as other adjustments to the calculation of evapotranspiration; bug fix in output of monthly NEP, NBP, soilr, and rh; bug fix on cLeaf monthly output; further development on gross land-use transitions, tracking of carbon from wood & crop harvest, and tracking of primary & secondary vegetation. |
| VISIT | Ito and Inatomi (2012), Kato et al. (2013) | No change. |





| *Intermediate complexity land carbon cycle model* | | |
|---|---|---|
| CARDAMOM | Bloom et al. (2016), Smallman et al. (2021) | No change. |
| *Global ocean biogeochemistry models* | | |
| NEMO-PlankTOM12 | Wright et all (2021) | Minor bug fixes, change to salinity restoring and restart file. New atmospheric CO2 input for simulations A and C. |
| NEMO4.2-PISCES (IPSL) | Aumont et al. (2015) | Switch to the new model version NEMO4.2-PISCES. Update following the new protocol (with 1750 preindustrial CO2 for spin-up). New atmospheric CO2 input for simulations A and C. |
| MICOM-HAMOCC (NorESM1-OCv1.2) | Schwinger et al. (2016) | No change in model set-up. New atmospheric CO2 file for simulations A and C. Corrected diagnostic output for pco2atm; diagnostic output for sfco2 and spco2 provided at the air-sea interface (not with respect to dry air at 1 atm). |
| MPIOM-HAMOCC6 | Lacroix et al. Global Change Biology 2021 | No change; only updated atmosphere CO2 input for A and C experiments and run 1948-2023. |
| NEMO3.6-PISCESv2-gas (CNRM) | Berthet et al. (2019) Séférian et al. (2019) | Updated simulations using 1750 preindustrial conditions instead of 1850. No change in model configuration. New atmospheric CO2 input for simulations A and C |
| FESOM2.1-REcoM3 | Gürses et al. (2023) | Updated atmospheric CO2 for simulations A and C. |
| MOM6-COBALT (Princeton) | Liao et al. (2020) | No change. |
| CESM-ETHZ | Doney et al. (2009) | Compared to the 2023 submission, the spinup was extended to 1422 years before 1750. Also, starting at 1751 the new atmospheric CO2 concentrations provided by GCB have been used for simulations A and C. |
| MRI-ESM2-3 | Tsujino et al. (2024), Sakamoto et al. (2023) | Iron circulation and its limitation on primary production are introduced. Updated atmospheric CO2 for simulations A and C |
| ACCESS (CSIRO) | Law et al. (2017) | No change in model set-up (since GCB2023). Updated atmospheric CO2 for simulations A and C. |
| *fCO2-products* | | |
| VLIZ-SOMFFN (former MPI-SOM-FFN) | Landschützer et al. (2016) | Time period 1982-2023; The estimate now coveres the full open ocean and coastal domain as well as the Arctic Ocean extension by merging 2 MLD proxies for year round full coverage. Additionally, in the SOM step, the Seaflux climatology is now used |
| Jena-MLS | Rödenbeck et al. (2014) updated to | Time period extended to 2023 |



| | Rödenbeck et al (2022) | |
|---|---|---|
| CMEMS-LSCE-FFNNv2 | Chau et al. (2022) | Time period now 1985-2023 |
| UExP-FNN-U (previously Watson et al.) | Watson et al. (2020) and Ford et al. (accepted) | Updated CCI-SST to v3 (Embury et al. 2024), with cool bias with respect to global drifter observations corrected following recommendations in Dong et al. (2022). Updated SOM-FFN implementation to Python. |
| NIES-ML3 | Zeng et al. (2022) | Updated time period 1982-2023. |
| JMA-MLR | Iida et al. (2021) | Time period extended to 2023 |
| OceanSODA-ETHZv2 | Gregor et al. (2024) | Updated method following Gregor et al 2024 and time period extended to 2023 |
| LDEO-HPD | Gloege et al. 2022 and Bennington et al. 2022 | Timeperiod extended to 2023 |
| CSIR-ML6 | Gregor et al. (2019) | Time period 1982-2023. |
| ***Atmospheric inversions*** | | |
| Jena CarboScope | Rödenbeck et al. (2018), Stephens et al. (2007) | Extension to end of year 2023. Slight change in station set. In the NBE-T inversion, removal of the relaxation term, instead, filtering out decadal variations from air temperature. Adding an additive correction to the result of the NBE-T inversion, to account for CO2 flux IAV not related to air temperature, based on 8 long atmospheric records available near-continuously since at least 1976. TM3 driven by ERA5 rather than NCEP. |
| CAMS | Chevallier et al. (2005), Remaud et al. (2018) | Extension to year 2023. Increase of the 3D resolution with hexagonal prisms rather than rectangular parallelepipeds (3 times more 3D cells than the previous submission). Update of the prior fluxes. |
| CarbonTracker Europe (CTE) | van der Laan-Luijkx et al. (2017) | Extension to 2023, update of prior fluxes. |
| NISMON-CO2 | Niwa et al. (2022), Niwa et al. (2017). | Extension to 2023, update of prior fluxes. |
| CT-NOAA | Jacobson et al. (2023a), Jacobson et al. (2024), Byrne et al. (2023), Krol et al. (2005), Peiro et al. (2022) | Extended to 2023 using the CarbonTracker Near-Real Time release CT-NRT.v2024-1 (Jacobson et al. 2024). |
| CMS-Flux | Liu et al. (2021) | Extension to 2023, update of prior fluxes. |
| CAMS-Satellite | Chevallier et al. (2005), Remaud et al. (2018) | Extension to year 2023. Increase of the 3D resolution with hexagonal prisms rather than rectangular parallelepipeds (3 times more 3D cells than the previous submission). Update of the prior fluxes. |




| | | |
|---|---|---|
| GONGGA | Jin et al. (2023), Nassar et al. (2010) | Extension to 2023, update of prior fluxes. |
| COLA | Liu et al. (2022) | Extension to 2023, update of prior fluxes. |
| GCASv2 | Jiang et al. (2021) & Emmons et al. (2010) | Extension to 2023, update of prior fluxes. |
| UoE in-situ | Feng et al. (2016) & Palmer et al. (2019) | Extension to 2023, update of prior fluxes. |
| IAPCAS | Yang et al. (2021) & Feng et al. (2016) | Extension to 2023, update of prior fluxes. |
| MIROC4-ACTM | Chandra et al. (2022) & Patra et al. (2018) | Extension to 2023, update of prior fluxes using only CASA and not VISIT. Less observation sites used in the assimilation (46 instead of 50). |
| NTFVAR | Nayagam et al. (2024) & Maksyutov et al. (2021) | New this year |
| ***Earth System Models*** | | |
| CanESM5 | Swart et al. (2019), Sospedra-Alfonso et al. (2021) | Reconstructions are extended to 1960-2023, and predictions are extended to 2024. |
| EC-Earth3-CC | Döscher et al. (2021), Bilbao et al. (2021), Bernardello et al. (2024) | New this year. |
| IPSL-CM6A-CO2-LR | Boucher et al. (2020) | Reconstructions are extended to 1960-2023, and predictions are extended to 2024. No change to model, the CMIP6 CovidMIP $CO_2$ emissions after 2015 are used. |
| MIROC-ES2L | Watanabe et al. (2020) | Reconstructions are extended to 1960-2023, and predictions are extended to 2024. No change to model, the simulations were rerun including a long spinup. |
| MPI-ESM1-2-LR | Mauritsen et al. (2019), Li et al. (2023) | Reconstructions are extended to 1960-2023, and predictions are extended to 2024. |



**Table 5.** Comparison of results from the bookkeeping method and budget residuals with results from the DGVMs, as well as additional estimates from atmospheric oxygen, atmospheric inversions and Earth System Models (ESMs) for different periods, the last decade, and the last year available. All values are in GtCyr−1. See Figure 7 for explanation of the bookkeeping component fluxes. The DGVM uncertainties represent ±1σ of the decadal or annual (for 2023) estimates from the individual DGVMs: for the inverse systems the mean and range of available results is given. All values are rounded to the nearest 0.1 GtC and therefore columns do not necessarily add to zero.

| | | Mean (GtC/yr) | | | | | | |
|---|---|---|---|---|---|---|---|---|
| | | 1960s | 1970s | 1980s | 1990s | 2000s | 2014-2023 | 2023 |
| Land-use change emissions (ELUC) | Bookkeeping (BK) Net flux (1a) | 1.6±0.7 | 1.4±0.7 | 1.4±0.7 | 1.6±0.7 | 1.4±0.7 | 1.1±0.7 | 1±0.7 |
| | BK - deforestation (total) | 1.7 [1.3,2.2] | 1.6 [1.2,2] | 1.6 [1.3,1.9] | 1.8 [1.6,2] | 1.9 [1.6,2.2] | 1.7 [1.4,2.3] | 1.7 [1.3,2.3] |
| | BK - forest regrowth (total) | -0.8 [-1.1,-0.6] | -0.9 [-1.1,-0.7] | -0.9 [-1,-0.7] | -0.9 [-1.1,-0.8] | -1.1 [-1.2,-0.9] | -1.2 [-1.5,-0.9] | -1.2 [-1.5,-0.9] |
| | BK - other transitions | 0.3 [0.3,0.4] | 0.2 [0.2,0.3] | 0.2 [0.1,0.3] | 0.1 [0,0.2] | 0.1 [0,0.1] | 0.1 [0,0.1] | 0 [0,0.1] |
| | BK - peat drainage & peat fires | 0.2 [0.1,0.2] | 0.2 [0.1,0.2] | 0.2 [0.2,0.2] | 0.3 [0.2,0.3] | 0.2 [0.2,0.3] | 0.2 [0.2,0.3] | 0.2 [0.2,0.3] |
| | BK - wood harvest & forest management | 0.2 [-0.2,0.6] | 0.3 [-0.2,0.6] | 0.3 [-0.2,0.7] | 0.3 [-0.1,0.6] | 0.3 [-0.1,0.6] | 0.3 [0,0.6] | 0.3 [0,0.7] |
| | DGVMs-net flux (1b) | 1.5±0.5 | 1.5±0.5 | 1.5±0.5 | 1.7±0.5 | 1.7±0.6 | 1.5±0.6 | 1.2±0.7 |
| Terrestrial sink (SLAND) | Residual sink from global budget ($E_{FOS}+E_{LUC}$(1a)-$G_{ATM}$-$S_{OCEAN}$) (2a) | 1.7±0.8 | 1.9±0.8 | 1.6±0.9 | 2.6±0.9 | 2.8±0.9 | 2.7±0.9 | 2.3±1 |
| | DGVMs (2b) | 1.2±0.5 | 2±0.8 | 1.8±0.8 | 2.5±0.6 | 2.8±0.7 | 3.2±0.9 | 2.3±1 |
| Net land fluxes (SLAND-ELUC) | GCB2024 Budget (2b-1a) | -0.4±0.9 | 0.5±1.1 | 0.4±1.1 | 0.9±0.9 | 1.4±1 | 2.1±1.1 | 1.3±1.2 |
| | Atmospheric O₂ | --- | --- | --- | 1.3±0.7 | 1±0.7 | 1±0.8 | - |
| | DGVMs-net (2b-1b) | -0.3±0.5 | 0.5±0.7 | 0.3±0.6 | 0.8±0.4 | 1.1±0.4 | 1.7±0.6 | 1.1±0.8 |
| | Inversions* | - [-,-] | - [-,-] | 0.3 [0.3,0.4] (2) | 0.9 [0.6,1.1] (3) | 1.2 [0.6,1.5] (4) | 1.4 [0.3,2.2] (10) | 0.9 [-0.1-2.7] (14) |
| | ESMs | 0 [-0.7,0.5] | 1.5 [1.2,2] | 1 [0.5,1.4] | 1.7 [1.2,2.4] | 1.8 [0.4,2.7] | 2.2 [0.3,3.6] | 1.8 [-2.9-3.7] |

*Estimates are adjusted for the pre-industrial influence of river fluxes, for the cement carbonation sink, and adjusted to common EFOS (Sect. 2.7). The ranges given include varying numbers (in parentheses) of inversions in each decade (Table A4)

**Table 6:** Comparison of results for the ocean sink from the *f*CO2-products, from global ocean biogeochemistry models (GOBMs), the best estimate for GCB2024 as calculated from fCO2-products and GOBMs that is used in the budget Table 7, as well as additional estimates from atmospheric oxygen, atmospheric inversions and Earth System Models (ESMs) for different periods, the last decade, and the last year available. All values are in GtCyr−1. Uncertainties represent ±1σ of the estimates from the GOBMs (N>10) and range of ensemble members is given for ensembles with N<10 (*f*CO2-products, inversions, ESMs). The uncertainty of the GCB2024 budget estimate is based on expert judgement (Section 2 and Supplementary S1 to S4) and for oxygen it is the standard deviation of a Monte Carlo ensemble (Section 2.8).

| | | | | | | | |
|---|---|---|---|---|---|---|---|
| | | | | **Mean (GtC/yr)** | | | |
| Product | 1960s | 1970s | 1980s | 1990s | 2000s | 2014-2023 | 2023 |
| *f*CO$_2$-products | --- | --- | --- | 2.3 [1.9,2.9] | 2.5 [2.3,2.7] | 3.1 [2.9,3.7] | 3 [2.3,4] |
| GOBMs | 1±0.2 | 1.3±0.3 | 1.8±0.3 | 2±0.3 | 2.2±0.3 | 2.6±0.4 | 2.7±0.4 |
| GCB2024 Budget | 1.2±0.4 | 1.5±0.4 | 1.9±0.4 | 2.1±0.4 | 2.3±0.4 | 2.9±0.4 | 2.9±0.4 |
| Atmospheric O$_2$ | --- | --- | --- | 2±0.5 | 2.8±0.4 | 3.4±0.5 | - |
| Inversions | - [-,-] | - [-,-] | 1.8 [1.8,1.9] (2) | 2.3 [2.1,2.5] (3) | 2.5 [2.3,3.1] (4) | 3.1 [2.4,4.1] (10) | 3 [1.8-4.1] (14) |
| ESMs | 0.7 [0.1,1.1] | 1 [0.4,1.4] | 1.4 [0.7,1.7] | 1.7 [1.1,2] | 1.9 [1.5,2.2] | 2.5 [2.2,2.8] | 2.5 [2.2-3] |

**Table 7:** Decadal mean in the five components of the anthropogenic CO2 budget for different periods, and last year available. All values are in GtC yr$^{-1}$, and uncertainties are reported as ±1σ. Fossil $CO_2$ emissions include cement carbonation. The table also shows the budget imbalance ($B_{IM}$), which provides a measure of the discrepancies among the nearly independent estimates. A positive imbalance means the emissions are overestimated and/or the sinks are too small. All values are rounded to the nearest 0.1 GtC and therefore columns do not necessarily add to zero.

| | | Mean (GtC/yr) | | | | | | |
|---|---|---|---|---|---|---|---|---|
| | | 1960s | 1970s | 1980s | 1990s | 2000s | 2014-2023 | 2023 | 2024 (Projection) |
| Total emissions (EFOS + ELUC) | Fossil CO2 emissions (EFOS)* | 3±0.2 | 4.7±0.2 | 5.5±0.3 | 6.4±0.3 | 7.8±0.4 | 9.7±0.5 | 10.1±0.5 | 10.2±0.5 |
| | Land-use change emissions (ELUC) | 1.6±0.7 | 1.4±0.7 | 1.4±0.7 | 1.6±0.7 | 1.4±0.7 | 1.1±0.7 | 1±0.7 | 1.1±0.7 |
| | Total emissions | 4.6±0.7 | 6.1±0.7 | 6.9±0.8 | 7.9±0.8 | 9.2±0.8 | 10.8±0.9 | 11.1±0.9 | 11.3±0.9 |
| Partitioning | Growth rate in atmos CO2 (GATM) | 1.7±0.07 | 2.8±0.07 | 3.4±0.02 | 3.1±0.02 | 4±0.02 | 5.2±0.02 | 5.9±0.2 | 5.9±0.5 |
| | Ocean sink (SOCEAN) | 1.2±0.4 | 1.5±0.4 | 1.9±0.4 | 2.1±0.4 | 2.3±0.4 | 2.9±0.4 | 2.9±0.4 | 3±0.6 |
| | Terrestrial sink (SLAND) | 1.2±0.5 | 2±0.8 | 1.8±0.8 | 2.5±0.6 | 2.8±0.7 | 3.2±0.9 | 2.3±1 | 3.2±1.5 |
| Budget Imbalance | BIM=EFOS+ELUC-(GATM+SOCE | 0.5 | -0.1 | -0.2 | 0.1 | 0 | -0.4 | 0 | -0.7 |



|  | 1960s | 1970s | 1980s | 1990s | 2000s | 2014-2023 | 2023 | 2024 (Projection) |
|---|---|---|---|---|---|---|---|---|
| AN+SL AND) |  |  |  |  |  |  |  |  |

*Fossil emissions excluding the cement carbonation sink amount to 3±0.2 GtC/yr, 4.7±0.2 GtC/yr, 5.5±0.3 GtC/yr, 6.4±0.3 GtC/yr, 7.9±0.4 GtC/yr, and 9.9±0.5 GtC/yr for the decades 1960s to 2010s respectively and to 10.3±0.5 GtC/yr for 2023, and 10.4±0.5 GtC/yr for 2024.



**Table 8.** Cumulative $CO_2$ for different time periods in gigatonnes of carbon (GtC). Fossil $CO_2$ emissions include cement carbonation. The budget imbalance ($B_{IM}$) provides a measure of the discrepancies among the nearly independent estimates. All values are rounded to the nearest 5 GtC and therefore columns do not necessarily add to zero. Uncertainties are reported as follows: $E_{FOS}$ is 5% of cumulative emissions; $E_{LUC}$ prior to 1959 is $1\sigma$ spread from the DGVMs, $E_{LUC}$ post-1959 is 0.7*number of years (where 0.7 GtC/yr is the uncertainty on the annual $E_{LUC}$ flux estimate); $G_{ATM}$ uncertainty is held constant at 5 GtC for all time periods; $S_{OCEAN}$ uncertainty is 20% of the cumulative sink (20% relates to the annual uncertainty of 0.4 GtC/yr, which is ~20% of the current ocean sink); and $S_{LAND}$ is the $1\sigma$ spread from the DGVMs estimates.

| | | 1750-2023 | 1850-2014 | 1850-2023 | 1960-2023 | 1850-2024 |
|---|---|---|---|---|---|---|
| Emissions | Fossil CO2 emissions (EFOS) | 490±25 | 400±20 | 490±25 | 410±20 | 500±25 |
| | Land-use change emissions (ELUC) | 255±75 | 215±60 | 225±65 | 90±45 | 225±65 |
| | Total emissions | 745±80 | 615±65 | 710±70 | 500±50 | 725±70 |
| Partitioning | Growth rate in atmos CO2 (GATM) | 305±5 | 235±5 | 285±5 | 220±5 | 290±5 |
| | Ocean sink (SOCEAN) | 195±40 | 160±30 | 185±35 | 130±25 | 185±35 |
| | Terrestrial sink (SLAND) | 245±65 | 190±55 | 220±60 | 150±40 | 225±60 |
| Budget imbalance | BIM=EFOS +ELUC- (GATM+SOCEAN+SLAND) | 0 | 30 | 25 | 0 | 20 |





**Table 9.** Average annual growth rate in fossil $CO_2$ emissions over the most recent decade (2014-2023) and the previous decade (2004-2013). The data for the World include the cement carbonation sink. IAS are emissions from international aviation and shipping. Rest of the World is World minus China, USA, EU27, India and IAS.

|  | World | China | USA | EU27 | India | OECD | Non-OECD | IAS | Rest of the World |
|---|---|---|---|---|---|---|---|---|---|
| 2004-2013 | 2.4% | 7.5% | -1.4% | -1.8% | 6.4% | -0.9% | 4.9% | 2.6% | 1.9% |
| 2014-2023 | 0.6% | 1.9% | -1.2% | -2.1% | 3.6% | -1.4% | 1.8% | -1.6% | 0.4% |



**Table 10.** Major known sources of uncertainties in each component of the Global Carbon Budget, defined as input data or processes that have a demonstrated effect of at least ±0.3 GtC yr-1.

| Source of uncertainty | Time scale (years) | Location | Evidence |
|---|---|---|---|
| **Fossil CO2 emissions (EFOS; Section 2.1)** | | | |
| energy statistics | annual to decadal | global, but mainly China & major developing countries | (Korsbakken et al., 2016, Guan et al., 2012) |
| carbon content of coal | annual to decadal | global, but mainly China & major developing countries | (Liu et al., 2015) |
| system boundary | annual to decadal | all countries | (Andrew, 2020a) |
| **Net land-use change flux (ELUC; section 2.2)** | | | |
| land-cover and land-use change statistics | continuous | global; in particular tropics | (Houghton et al., 2012, Gasser et al., 2020, Ganzenmüller et al., 2022, Yu et al. 2022) |
| sub-grid-scale transitions | annual to decadal | global | (Wilkenskjeld et al., 2014, Bastos et al., 2021) |
| vegetation biomass | annual to decadal | global; in particular tropics | (Houghton et al., 2012, Bastos et al., 2021) |
| forest degradation (fire, selective logging) | annual to decadal | tropics; Amazon | (Aragão et al., 2018, Qin et al., 2021, Lapola et al., 2023) |
| wood and crop harvest | annual to decadal | global; SE Asia | (Arneth et al., 2017, Erb et al., 2018) |
| peat burning | multi-decadal trend | global | (van der Werf et al., 2010, 2017) |
| loss of additional sink capacity | multi-decadal trend | global | (Pongratz et al., 2014, Gasser et al., 2020; Obermeier et al., 2021; Dorgeist et al., 2024) |
| environmental effects | multi-decadal trend | global | (Gasser et al. 2020, Dorgeist et al., 2024) |

Atmospheric growth rate (GATM; section 2.4) no demonstrated uncertainties larger than ±0.3 GtC yr-1 . The uncertainties in GATM have been estimated as ±0.2 GtC yr-1, although the conversion of the growth rate into a global annual flux assuming instantaneous mixing throughout the atmosphere introduces additional errors that have not yet been quantified.

Ocean sink (SOCEAN; section 2.5)





| sparsity in surface fCO2 observations | mean, decadal variability and trend | global, in particular southern hemisphere | (Gloege et al., 2021, Denvil-Sommer et al., 2021, Hauck et al., 2023a; Dong et al., 2024b) |
|---|---|---|---|
| riverine carbon outgassing and its anthropogenic perturbation | annual to decadal | global, in particular partitioning between Tropics and South | (Aumont et al., 2001, Lacroix et al., 2020, Crisp et al., 2022) |
| Models underestimate interior ocean anthropogenic carbon storage | annual to decadal | global | (Friedlingstein et al., 2022a, this study, DeVries et al., 2023, Müller et al., 2023) |
| near-surface temperature and salinity gradients | mean on all time-scales | global | (Watson et al., 2020, Dong et al., 2022, Bellenger et al., 2023, Dong et al., 2024a) |

| Land sink (SLAND; section 2.6) | | | |
|---|---|---|---|
| strength of CO2 fertilisation | multi-decadal trend | global | (Wenzel et al., 2016; Walker et al., 2021) |
| response to variability in temperature and rainfall | annual to decadal | global; in particular tropics | (Cox et al., 2013; Jung et al., 2017; Humphrey et al., 2018; 2021) |
| nutrient limitation and supply | annual to decadal | global | (Zaehle et al., 2014) |
| carbon allocation and tissue turnover rates | annual to decadal | global | (De Kauwe et al., 2014; O'Sullivan et al., 2022) |
| tree mortality | annual | global in particular tropics | (Hubau et al., 2021; Brienen et al., 2020) |
| response to diffuse radiation | annual | global | (Mercado et al., 2009; O'Sullivan et al., 2021) |
| estimation under constant pre-industrial land cover | multi-decadal trend | global | (Gasser et al. 2020, Dorgeist et al., 2024) |



**Figures and Captions**

Figure 1. Surface average atmospheric $CO_2$ concentration (ppm). Since 1980, monthly data are from NOAA/GML (Lan et al., 2024) and are based on an average of direct atmospheric $CO_2$ measurements from multiple stations in the marine boundary layer (Masarie and Tans, 1995). The 1958-1979 monthly data are from the Scripps Institution of Oceanography, based on an average of direct atmospheric $CO_2$ measurements from the Mauna Loa and South Pole stations (Keeling et al., 1976). To account for the difference of mean $CO_2$ and seasonality between the NOAA/GML and the Scripps station networks used here, the Scripps surface average (from two stations) was de-seasonalised and adjusted to match the NOAA/GML surface average (from multiple stations) by adding the mean difference of 0.667 ppm, calculated here from overlapping data during 1980-2012.

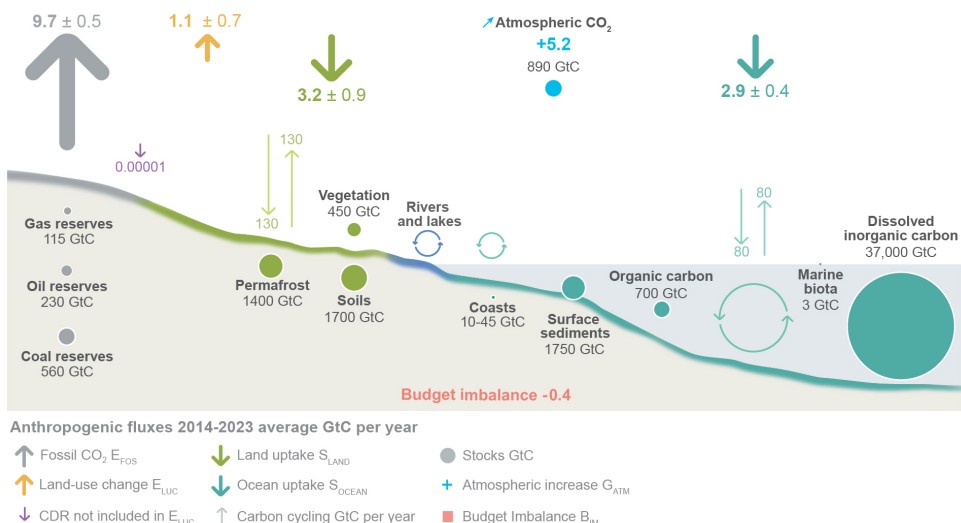

**Figure 2.** Schematic representation of the overall perturbation of the global carbon cycle caused by anthropogenic activities, averaged globally for the decade 2014-2023. See legends for the corresponding arrows. Fluxes estimates and their 1 standard deviation uncertainty are as reported in Table 7. The CDR estimate is for the year 2023 only. The uncertainty in the atmospheric $CO_2$ growth rate is very small ($\pm0.02$ GtC yr$^{-1}$) and is neglected for the figure. The anthropogenic perturbation occurs on top of an active carbon cycle, with fluxes and stocks represented in the background and taken from Canadell et al. (2021) for all numbers, except for the carbon stocks in coasts which is from a literature review of coastal marine sediments (Price and Warren, 2016). Fluxes are in GtC yr$^{-1}$ and reservoirs in GtC.





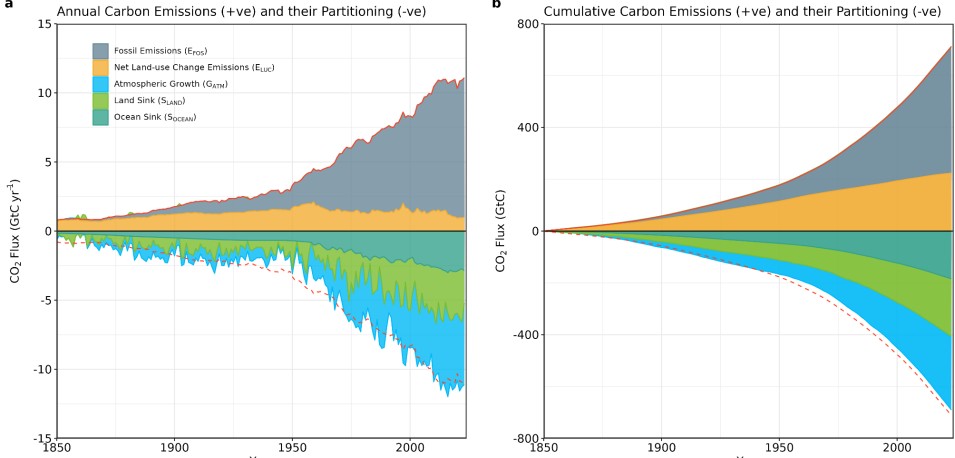

**Figure 3.** Combined components of the global carbon budget as a function of time, for fossil $CO_2$ emissions ($E_{FOS}$, including a small sink from cement carbonation; grey) and emissions from land-use change ($E_{LUC}$; brown), as well as their partitioning among the atmosphere ($G_{ATM}$; cyan), ocean ($S_{OCEAN}$; blue), and land ($S_{LAND}$; green). Panel (a) shows annual estimates of each flux (in GtC yr$^{-1}$) and panel (b) the cumulative flux (the sum of all prior annual fluxes, in GtC) since the year 1850. The partitioning is based on nearly independent estimates from observations (for $G_{ATM}$) and from process model ensembles constrained by data (for $S_{OCEAN}$ and $S_{LAND}$) and does not exactly add up to the sum of the emissions, resulting in a budget imbalance ($B_{IM}$) which is represented by the difference between the bottom red line (mirroring total emissions) and the sum of carbon fluxes in the ocean, land, and atmosphere reservoirs. All data are in GtC yr$^{-1}$ (panel a) and GtC (panel b). The $E_{FOS}$ estimate is based on a mosaic of different datasets, and has an uncertainty of ±5% (±1σ). The $E_{LUC}$ estimate is from four bookkeeping models (Table 4) with uncertainty of ±0.7 GtC yr$^{-1}$. The $G_{ATM}$ estimates prior to 1959 are from Joos and Spahni (2008) with uncertainties equivalent to about ±0.1-0.15 GtC yr-1 and from Lan et al. (2024) since 1959 with uncertainties of about +-0.07 GtC yr$^{-1}$ during 1959-1979 and ±0.02 GtC yr$^{-1}$ since 1980. The $S_{OCEAN}$ estimate is the average from Khatiwala et al. (2013) and DeVries (2014) with uncertainty of about ±30% prior to 1959, and the average of an ensemble of models and an ensemble of $fCO_2$-products (Table 4) with uncertainties of about ±0.4 GtC yr$^{-1}$ since 1959. The $S_{LAND}$ estimate is the average of an ensemble of models (Table 4) with uncertainties of about ±1 GtC yr$^{-1}$. See the text for more details of each component and their uncertainties.

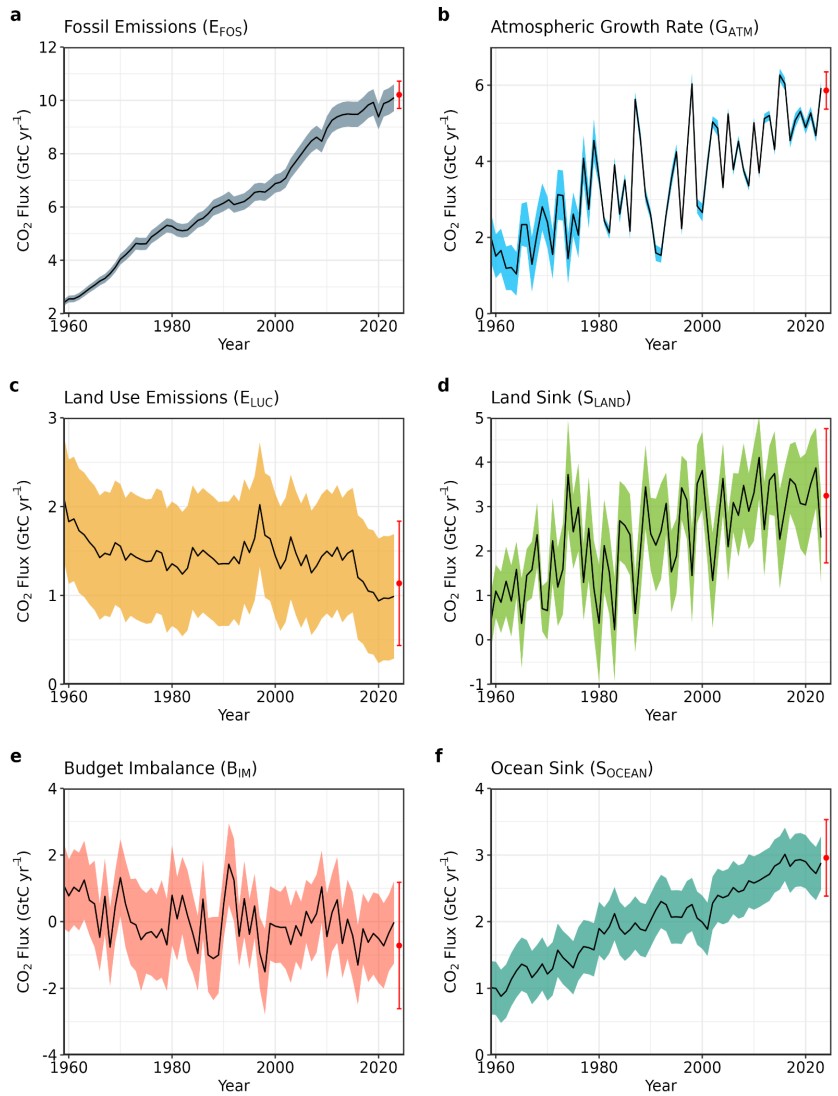

**Figure 4.** Components of the global carbon budget and their uncertainties as a function of time, presented individually for (a) fossil $CO_2$, including cement carbonation emissions ($E_{FOS}$), (b) growth rate in atmospheric $CO_2$ concentration ($G_{ATM}$), (c) emissions from land-use change ($E_{LUC}$), (d) the land $CO_2$ sink ($S_{LAND}$), (e) the ocean $CO_2$ sink ($S_{OCEAN}$), (f) the budget imbalance ($B_{IM}$) that is not accounted for by the other terms. Positive values of $S_{LAND}$ and $S_{OCEAN}$ represent a flux from the atmosphere to land or the ocean. All data are in GtC yr$^{-1}$ with the uncertainty bounds representing ±1 standard deviation in shaded colour. Data sources are as in Figure 3. The red dots indicate our projections for the year 2024 and the red error bars the uncertainty in the 2024 projections (see methods).

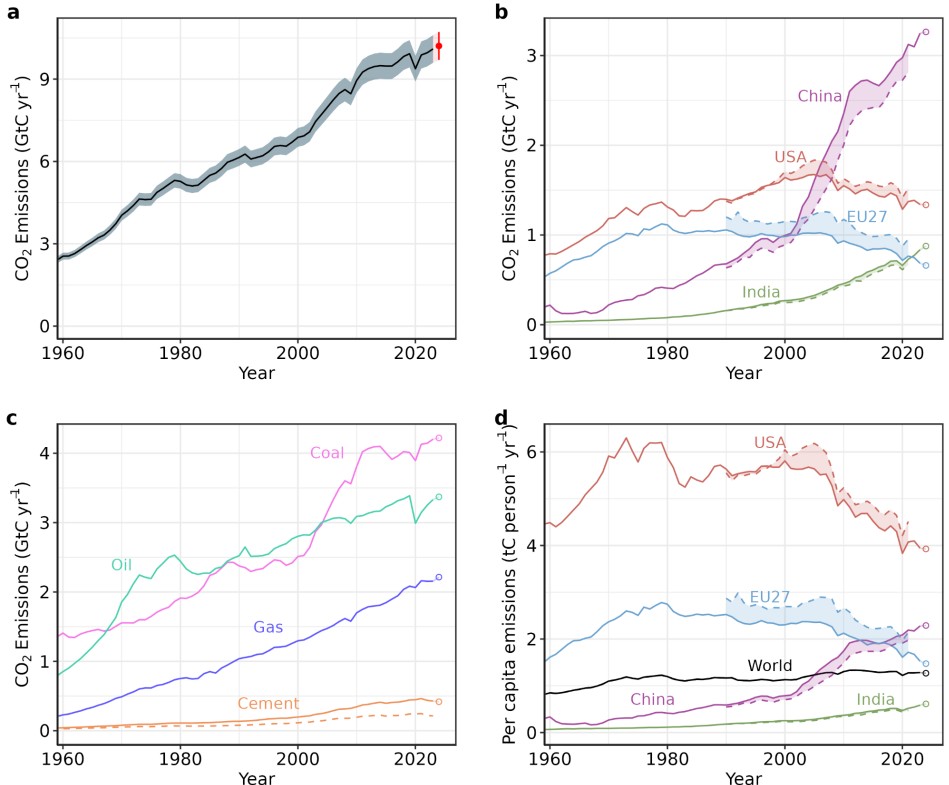

**Figure 5.** Fossil $CO_2$ emissions for (a) the globe, including an uncertainty of ± 5% (grey shading) and a projection through the year 2024 (red dot and uncertainty range), (b) territorial (solid lines) and consumption (dashed lines) emissions for the top three country emitters (USA, China, India) and for the European Union (EU27), (c) global emissions by fuel type, including coal, oil, gas, and cement, and cement minus cement carbonation (dashed), and (d) per-capita emissions the world and for the large emitters as in panel (b). Territorial emissions are primarily from a draft update of Hefner and Marland (2023) except for national data for most Annex I countries for 1990-2022, which are reported to the UNFCCC as detailed in the text, as well as some improvements in individual countries, and extrapolated forward to 2023 using data from Energy Institute. Consumption-based emissions are updated from Peters et al. (2011a). See Section 2.1 and Supplement S.1 for details of the calculations and data sources.

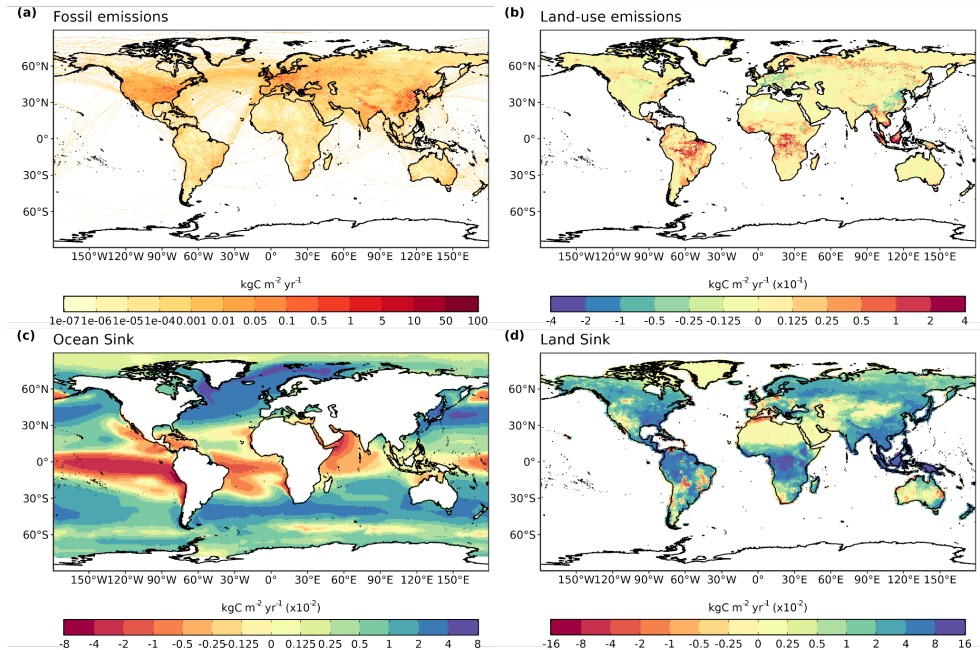

**Figure 6.** The 2014-2023 decadal mean components of the global carbon budget, presented for (a) fossil $CO_2$ emissions ($E_{FOS}$), (b) land-use change emissions ($E_{LUC}$), (c) the ocean $CO_2$ sink ($S_{OCEAN}$), and (d) the land $CO_2$ sink ($S_{LAND}$). Positive values for $E_{FOS}$ and $E_{LUC}$ represent a flux to the atmosphere, whereas positive values of $S_{OCEAN}$ and $S_{LAND}$ represent a flux from the atmosphere to the ocean or the land (carbon sink). In all panels, yellow/red colours represent a source (flux from the land/ocean to the atmosphere), green/blue colours represent a sink (flux from the atmosphere into the land/ocean). All units are in kgC $m^{-2}$ $yr^{-1}$. Note the different scales in each panel. $E_{FOS}$ data shown is from GCP-GridFEDv2024.0 and does not include cement carbonation. The $E_{LUC}$ map shows the average $E_{LUC}$ from the four bookkeeping models plus emissions from peat drainage and peat fires. BLUE and LUCE provide spatially explicit estimates at 0.25° resolution. Gridded $E_{LUC}$ estimates for H&C2023 and OSCAR are derived by spatially distributing their national data based on the spatial patterns of BLUE gross fluxes in each country (see Schwingshackl et al., 2022, for more details about the methodology). $S_{OCEAN}$ data shown is the average of GOBMs and $f$CO$_2$-products means, using GOBMs simulation A, no adjustment for bias and drift applied to the gridded fields (see Section 2.5). $S_{LAND}$ data shown is the average of the DGVMs for simulation S2 (see Section 2.6).

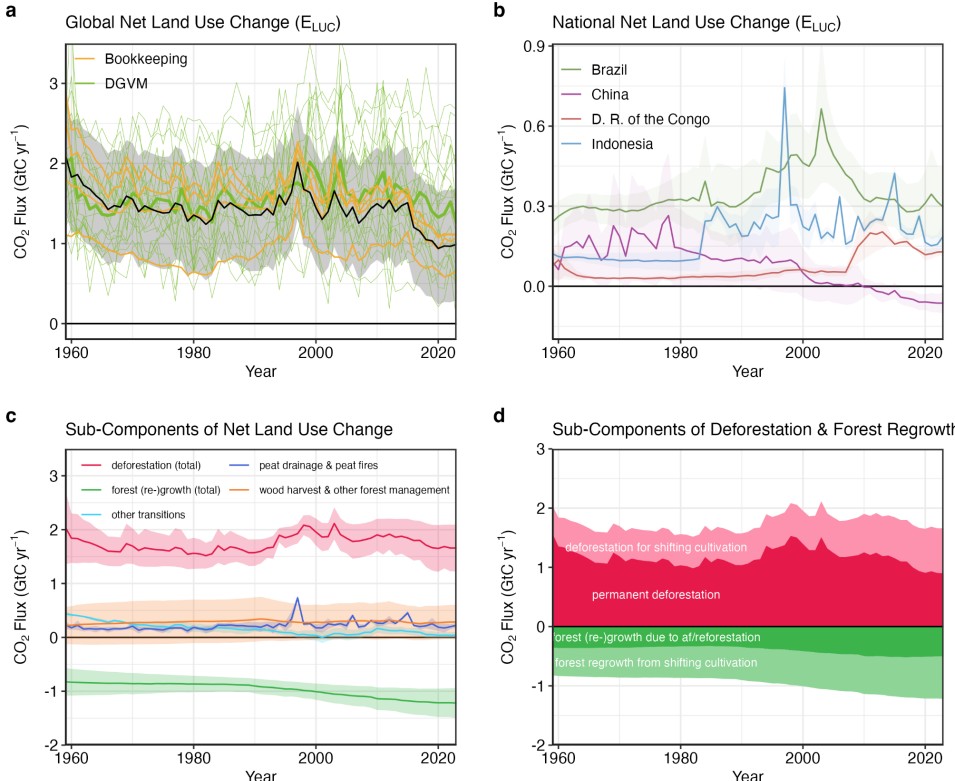

**Figure 7.** Net $CO_2$ exchanges between the atmosphere and the terrestrial biosphere related to land use change. (a) Net $CO_2$ emissions from land-use change ($E_{LUC}$) with estimates from the four bookkeeping models (yellow lines) and the budget estimate (black with ±1σ uncertainty), which is the average of the four bookkeeping models. Estimates from individual DGVMs (narrow green lines) and the DGVM ensemble mean (thick green line) are also shown. (b) Net $CO_2$ emissions from land-use change from the four countries with largest cumulative emissions since 1959. Values shown are the average of the four bookkeeping models, with shaded regions as ±1σ uncertainty. (c) Sub-components of $E_{LUC}$: (i) emissions from deforestation (including permanent deforestation and deforestation in shifting cultivation cycles), (ii) emissions from peat drainage & peat fires, (iii) removals from forest (re-)growth (including forest (re-)growth due to afforestation and reforestation and forest regrowth in shifting cultivation cycles), (iv) fluxes from wood harvest and other forest management (comprising slash and product decay following wood harvest, regrowth after wood harvest, and fire suppression), and (v) emissions and removals related to other land-use transitions. The sum of the five components is $E_{LUC}$ shown in panel (a). (d) Sub-components of 'deforestation (total)' and of 'forest (re-)growth (total)': (i) deforestation in shifting cultivation cycles, (ii) permanent deforestation, (iii) forest (re-)growth due to afforestation and/or reforestation, and (iv) forest regrowth in shifting cultivation cycles.

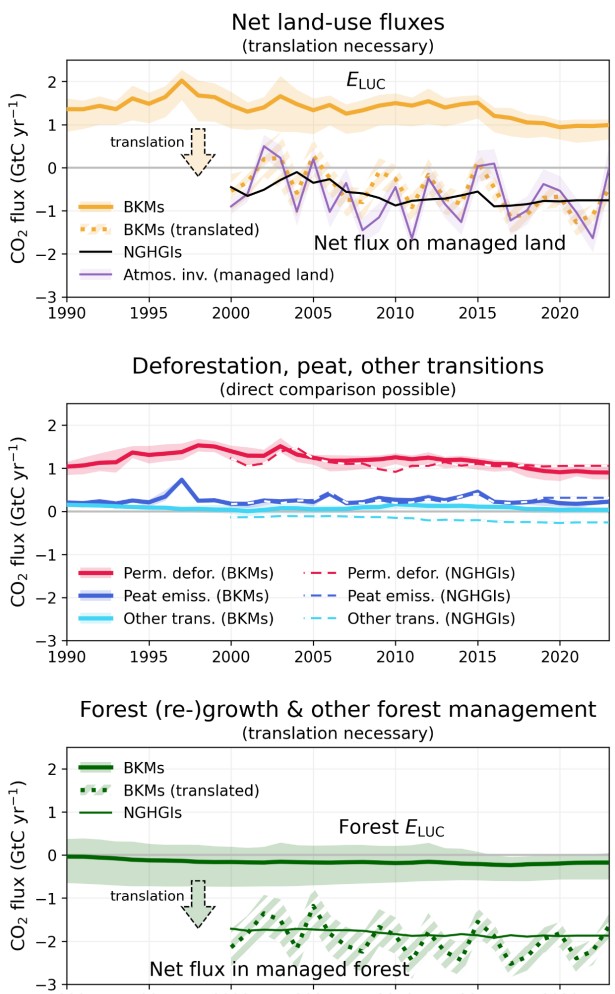

**Figure 8.** Comparison of land-use flux estimates from bookkeeping models (BKMs; following the GCB definition of $E_{LUC}$), national GHG inventories (NGHGIs; following IPCC guidelines and thus including all carbon fluxes on managed land), and atmospheric inversion systems (considering fluxes on managed land only). To compare BKM results with NGHGIs, a translation is necessary for some subcomponents. (a) Net land-use fluxes, for which a translation of BKMs is necessary, (b) subcomponents permanent deforestation, peat drainage & peat fires, and other transitions, which can be directly compared and (c) subcomponent forest (re-)growth & other forest management, for which a translation is necessary. The lines represent the mean of 4 BKMs and 14 atmospheric inversion estimates, respectively; Shaded areas denote the full range across BKM estimates and the standard deviation for atmospheric inversions, respectively. The subcomponent forest (re-)growth & other forest management includes removals from forest (re-)growth (permanent), emissions and removals from wood harvest & other forest management, and emissions and removals in shifting cultivation cycles. The translation of



BKM estimates to NGHGI estimates in (a) and (c) is done by adding the natural land sink in managed forests to the BKM estimates (see also Table S10). The GCB definition of ELUC and the NGHGI definition of land-use fluxes are equally valid, each in its own context. For illustrative purposes we only show the translation of BKM estimates to the NGHGI definition.

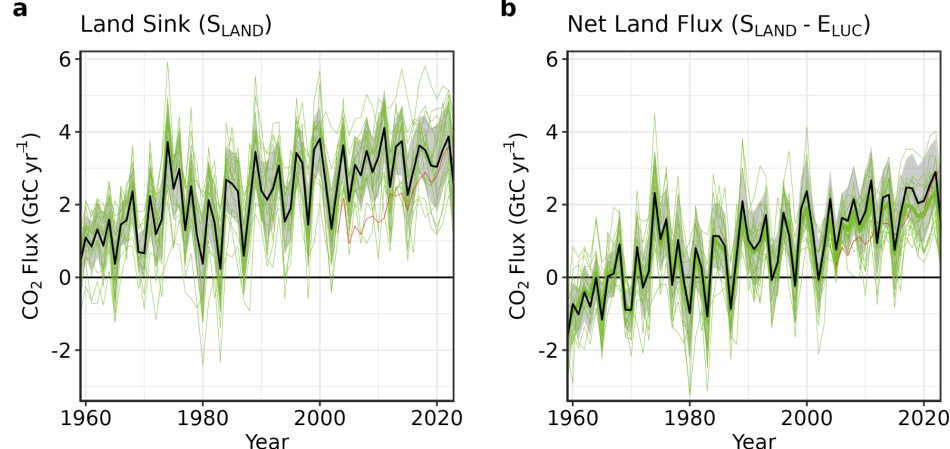

**Figure 9.** (a) The land $CO_2$ sink ($S_{LAND}$) estimated by individual DGVMs (green), and CARDAMOM (red), as well as the budget estimate (black with ±1σ uncertainty), which is the average of all DGVMs. (b) Net atmosphere-land $CO_2$ fluxes ($S_{LAND} - E_{LUC}$). The budget estimate of the net land flux (black with ±1σ uncertainty) combines the DGVM estimate of $S_{LAND}$ from panel (a) with the bookkeeping estimate of $E_{LUC}$ from Figure 7a. Uncertainties are similarly propagated in quadrature. DGVMs also provide estimates of $E_{LUC}$ (see Figure 7a), which can be combined with their own estimates of the land sink. Hence panel (b) also includes an estimate for the net land flux for individual DGVMs (thin green lines) and their multi-model mean (thick green line).

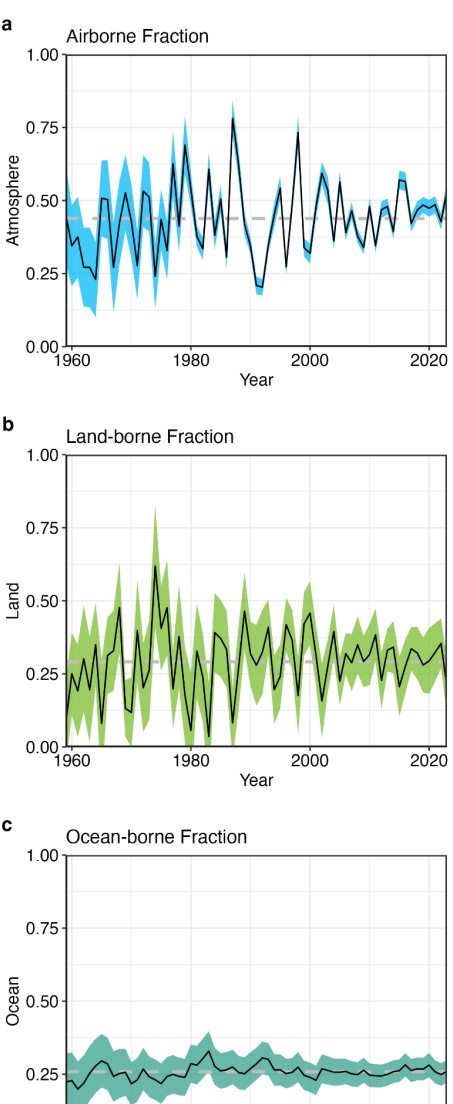

**Figure 10.** The partitioning of total anthropogenic $CO_2$ emissions ($E_{FOS}$ + $E_{LUC}$) across (a) the atmosphere (airborne fraction), (b) land (land-borne fraction), and (c) ocean (ocean-borne fraction). Black lines represent the central estimate, and the coloured shading represents the uncertainty. The grey dashed lines represent the long-term average of the airborne (44%), land-borne (30%) and ocean-borne (25%) fractions during 1960-2023 (with a $B_{IM}$ of 1%).

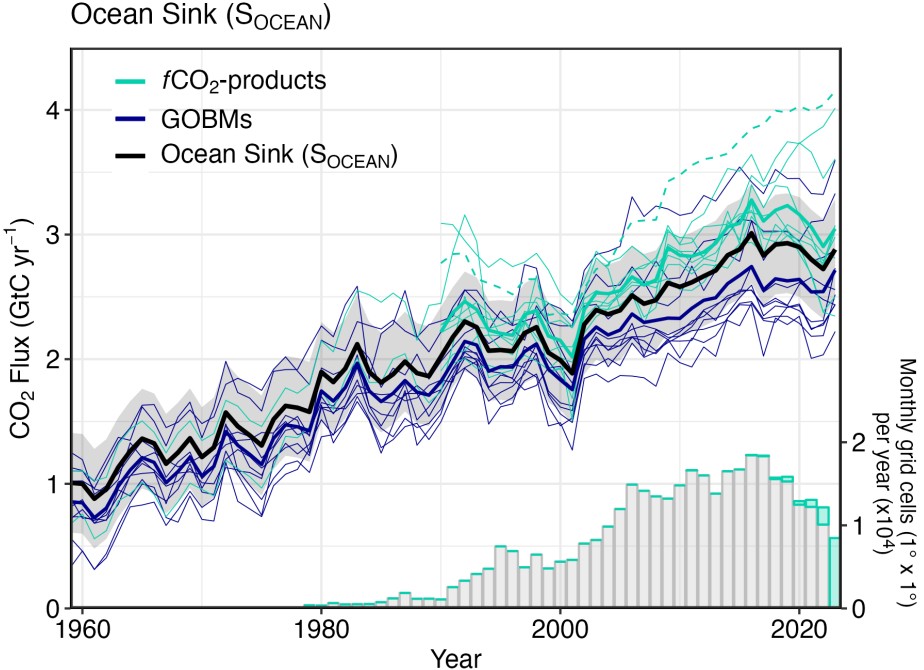

**Figure 11.** Comparison of the anthropogenic atmosphere-ocean $CO_2$ flux showing the budget values of $S_{OCEAN}$ (black; with the uncertainty in grey shading), individual ocean models (royal blue), and the ocean $fCO_2$-products (cyan; with UExP-FFN-U, previously Watson et al. (2020), in dashed line as not used for ensemble mean). Two $fCO_2$-products (Jena-MLS, LDEO-HPD) extend back to 1959. The $fCO_2$-products were adjusted for the pre-industrial ocean source of $CO_2$ from river input to the ocean, by subtracting a source of 0.65 GtC yr$^{-1}$ to make them comparable to $S_{OCEAN}$ (see Section 2.5). Bar-plot in the lower right illustrates the number of monthly gridded values in the SOCAT v2024 database (Bakker et al., 2024). Grey bars indicate the number of grid cells in SOCAT v2023, and coloured bars indicate the newly added grid cells in v2024.

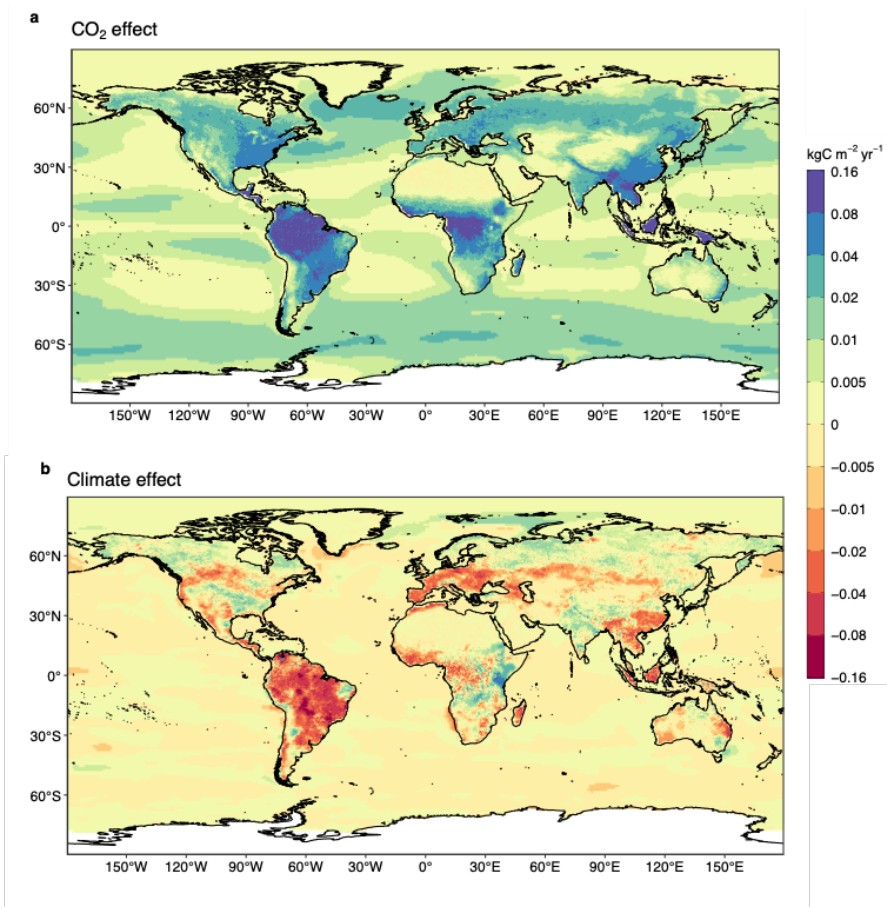

**Figure 12.** Attribution of the atmosphere-ocean ($S_{OCEAN}$) and atmosphere-land ($S_{LAND}$) $CO_2$ fluxes to (a) increasing atmospheric $CO_2$ concentrations and (b) changes in climate, averaged over the previous decade 2014-2023. All data shown is from the processed-based GOBMs and DGVMs. Note that the sum of ocean $CO_2$ and climate effects shown here will not equal the ocean sink shown in Figure 6, which includes the $f$CO_2-products. See Supplement S.3.2 and S.4.1 for attribution methodology. Units are in kgC m$^{-2}$ yr$^{-1}$ (note the non-linear colour scale). Positive values (blue) are $CO_2$ sinks, negative values (red) are $CO_2$ sources.

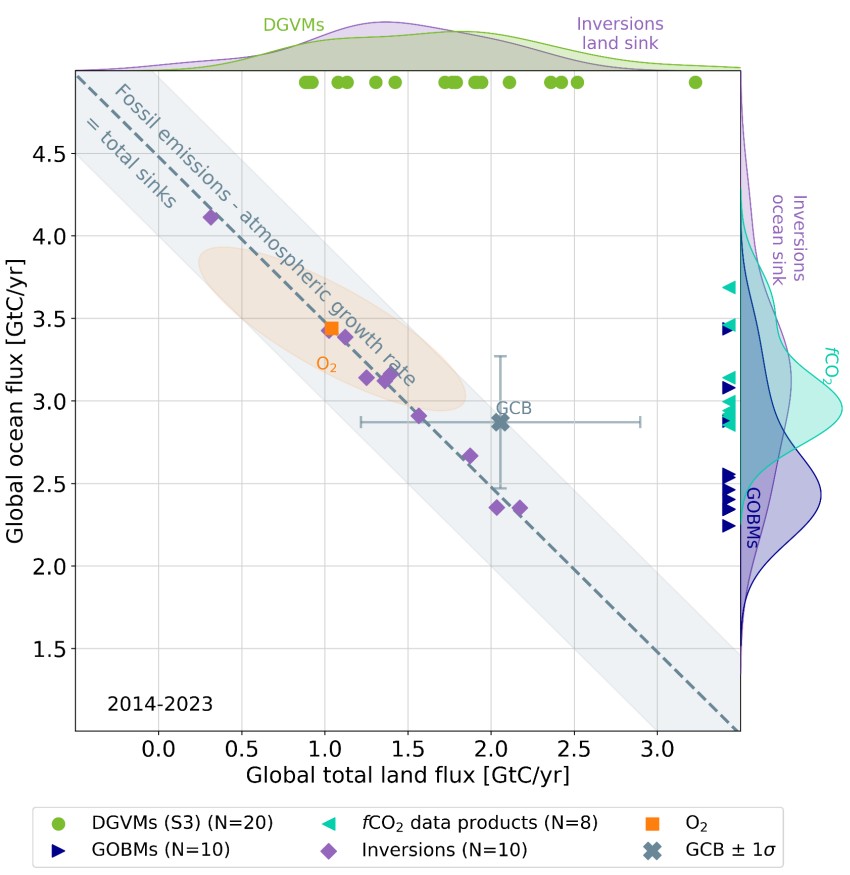

**Figure 13.** The 2014-2023 decadal mean global net atmosphere-ocean and atmosphere-land fluxes derived from the ocean models and $f$CO$_2$ products (y-axis, right and left pointing blue triangles respectively), and from the DGVMs (x-axis, green symbols), and the same fluxes estimated from the atmospheric inversions (purple symbols). The shaded distributions show the densities of the ensembles of individual estimates. The grey central cross is the mean (±1σ) of S$_{OCEAN}$ and (S$_{LAND}$ – E$_{LUC}$) as assessed in this budget. The grey diagonal line represents the constraint on the global land + ocean net flux, i.e. global fossil fuel emissions minus the atmospheric growth rate from this budget (E$_{FOS}$ – G$_{ATM}$). The orange square represents the same global net atmosphere-ocean and atmosphere-land fluxes as estimated from the atmospheric O$_2$ constraint (the ellipse drawn around the central atmospheric O$_2$ estimate is a contour representing the 1σ uncertainty of the land and ocean fluxes as a joint probability distribution). Positive values are CO$_2$ sinks. Note that the inverse estimates have been scaled for a minor difference between E$_{FOS}$ and GridFEDv2024.0 (Jones et al., 2024a).

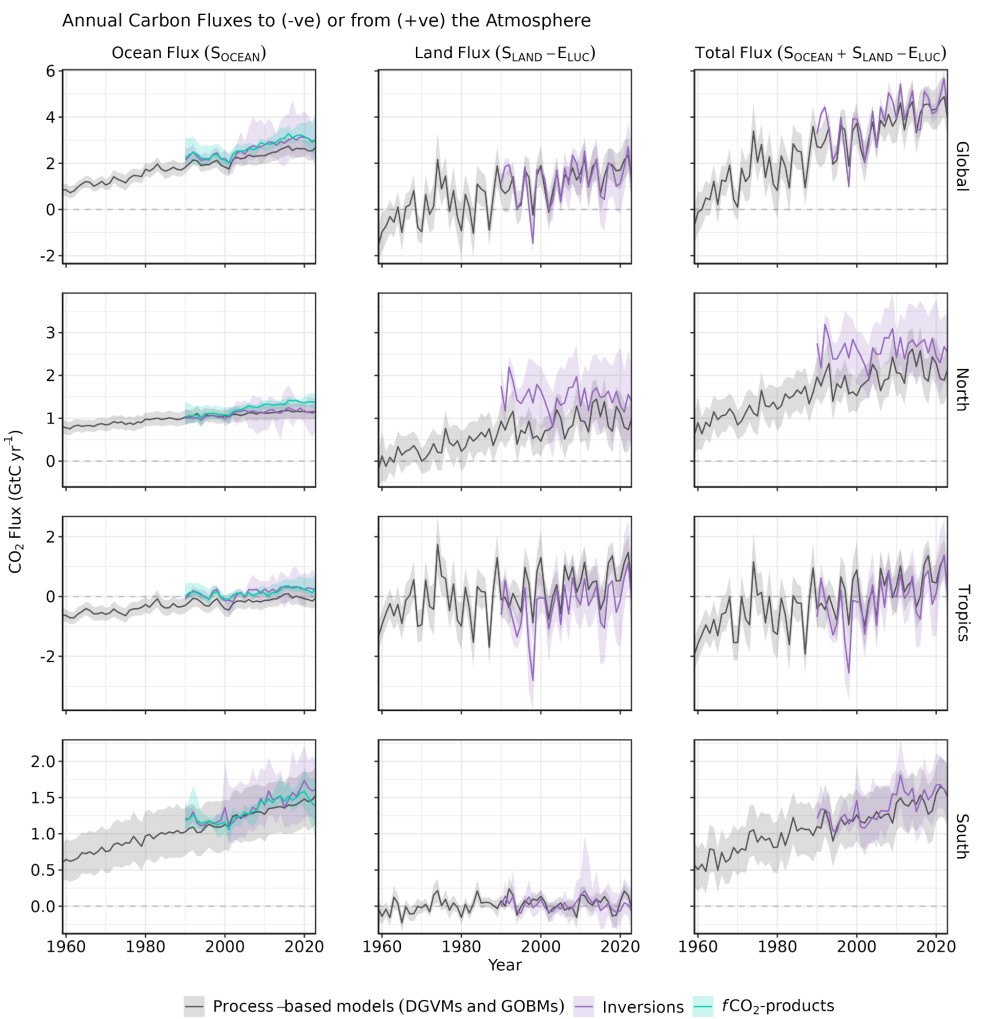

**Figure 14.** $CO_2$ fluxes between the atmosphere and the Earth's surface separated between land and oceans, globally and in three latitude bands. The ocean flux is $S_{OCEAN}$ and the land flux is the net atmosphere-land fluxes from the DGVMs. The latitude bands are (top row) global, (2nd row) north (>30°N), (3rd row) tropics (30°S-30°N), and (bottom row) south (<30°S), and over ocean (left column), land (middle column), and total (right column). Estimates are shown for: process-based models (DGVMs for land, GOBMs for oceans); inversion systems (land and ocean); and $f$CO$_2$-products (ocean only). Positive values are $CO_2$ sinks. Mean estimates from the combination of the process models for the land and oceans are shown (black line) with ±1 σ of the model ensemble (grey shading). For the total uncertainty in the process-based estimate of the total sink, uncertainties are summed in quadrature. Mean estimates from the atmospheric inversions are shown (purple lines) with their full spread (purple shading). Mean estimates from the $f$CO$_2$-products are shown for the ocean domain (light blue



lines) with full model spread (light blue shading). The global $S_{OCEAN}$ (upper left) and the sum of $S_{OCEAN}$ in all three regions represents the anthropogenic atmosphere-to-ocean flux based on the assumption that the preindustrial ocean sink was 0 GtC yr$^{-1}$ when riverine fluxes are not considered. This assumption does not hold at the regional level, where preindustrial fluxes can be significantly different from zero. Hence, the regional panels for $S_{OCEAN}$ represent a combination of natural and anthropogenic fluxes. Bias-correction and area-weighting were only applied to global $S_{OCEAN}$; hence the sum of the regions is slightly different from the global estimate (<0.07 GtC yr$^{-1}$).

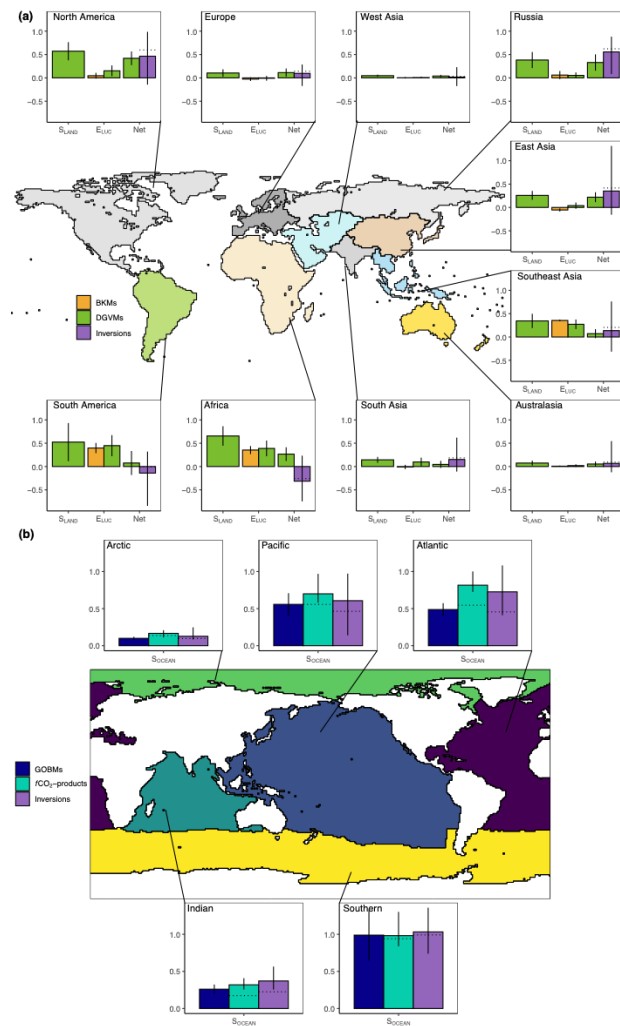

**Figure 15.** Decadal mean (a) land and (b) ocean fluxes for RECCAP-2 regions over 2014-2023. For land fluxes, $S_{LAND}$ is estimated by the DGVMs (green bars), with the error bar as $\pm1\sigma$ spread among models. A positive $S_{LAND}$ is a net transfer of carbon from the atmosphere to the land. $E_{LUC}$ fluxes are shown for both DGVMs (green) and bookkeeping models (orange), again with the uncertainty calculated as the $\pm1\sigma$ spread. Note, a positive $E_{LUC}$ flux indicates a loss of carbon from the land. The net land flux is shown for both DGVMs (green) and atmospheric inversions (purple), including the full model spread for inversions. The net ocean sink ($S_{OCEAN}$) is estimated by GOBMs (royal blue), $f$CO₂-products (cyan), and atmospheric inversions (purple). Uncertainty is estimated as the $\pm1\sigma$ spread for GOBMs, and the full model spread for the other two datasets. The dotted lines show the $f$CO₂-products and inversion results without river flux adjustment. Positive values are CO₂ sinks.

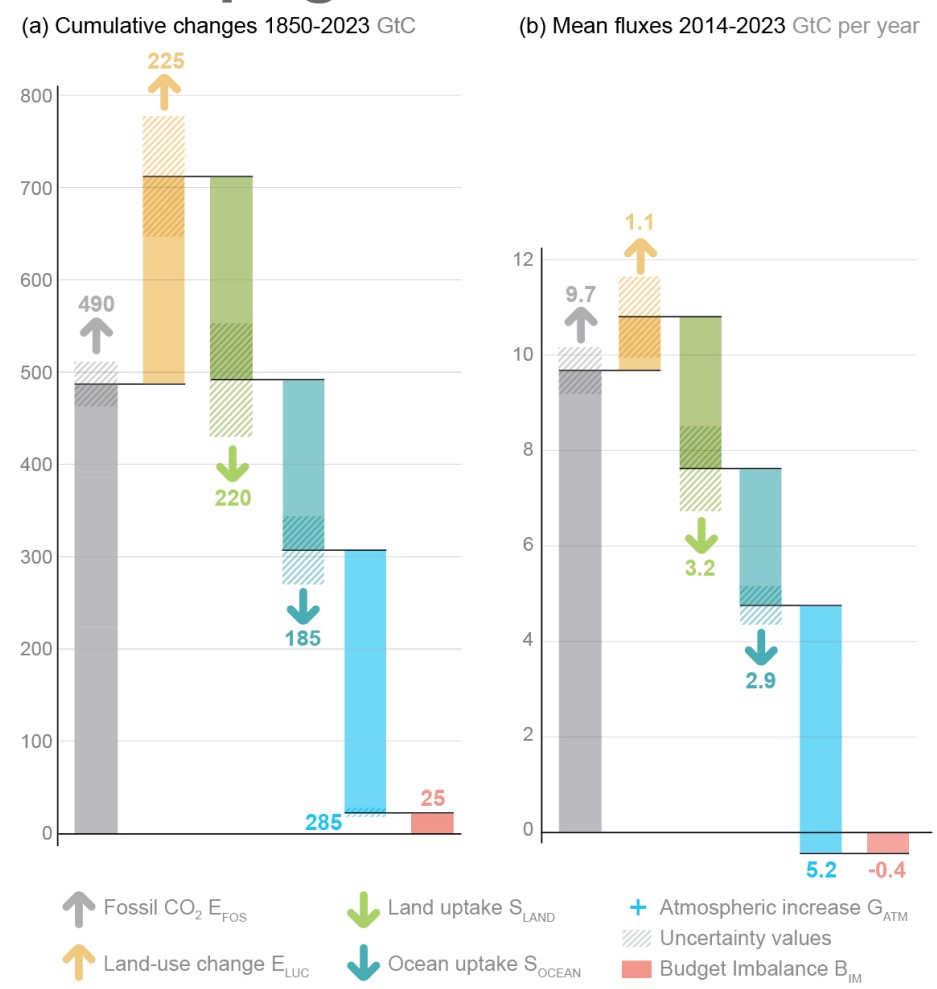

**Figure 16.** Cumulative changes over the 1850-2023 period (left) and average fluxes over the 2014-2023 period (right) for the anthropogenic perturbation of the global carbon cycle. See the caption of Figure 3 for key information and the methods in text for full details.

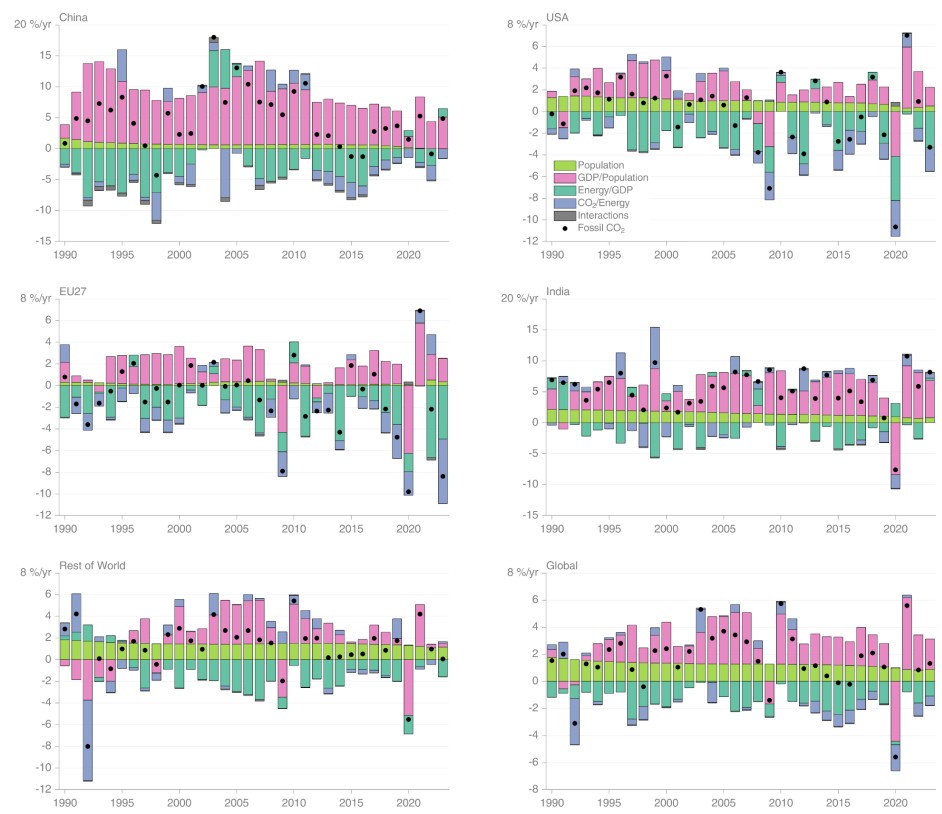

**Figure 17.** Kaya decomposition of the main drivers of fossil CO$_2$ emissions, considering population, GDP per person, Energy per GDP, and CO$_2$ emissions per energy, for China (top left), USA (top right), EU27 (middle left), India (middle right), Rest of the World (bottom left), and World (bottom right). Black dots are the annual fossil CO$_2$ emissions growth rate, coloured bars are the contributions from the different drivers to this growth rate. A general trend is that population and GDP growth put upward pressure on emissions (positive values), while energy per GDP and, more recently, CO$_2$ emissions per energy put downward pressure on emissions (negative values). Both the COVID-19 induced drop during 2020 and the recovery in 2021 led to a stark contrast to previous years, with different drivers in each region.