# Peer review of "Global Carbon Budget 2024"

_Earth System Science Data, 2024_

## Author Comment (AC1)

Global Carbon Budget 2024

Response to Referee 1

I have focussed my review here on the sections of the paper for which I have decent expertise (notably land carbon fluxes and the remaining carbon budget). My (minor) comments are:

Line 210: What does "forestry" refer to here? I don't see how forestry (the vast majority of which is deforestation) leads to carbon removal, so mentioning it here seems like it will cause confusion.

**Clarified now, by forestry we meant forest regrowth in shifting cultivation cycles.**

Lines 452-455: In the definition of CDR, it would be worth again mentioning that this definition follows the scientific convention of not including passive carbon sinks. Otherwise, this rather confuses the definition of net zero and its ability to lead to stable global temperatures.

**Done, the sentence now reads: "CDR is defined as the set of anthropogenic activities that remove CO2 from the atmosphere, in addition to the Earth's natural processes (such as carbon uptake in response to atmospheric $CO_2$ increase), and store it in durable form, such as in forest biomass, soils, long-lived products, ocean or geological reservoirs."**

Line 460: I don't think that transfer of carbon to harvested wood products should count as CDR. This is at best a delayed emission from deforestation. I am glad to see that this flux is not included in the budget (lines 466-468) though the preceding text suggests that they are. For this opening text, I would suggest being more clear at the outset what fluxes are considered to the CDR in the budget (and in the 1.2 GtC/yr number given in the summary). This comment applies also to lines 922-925 which uncritically refers to HWP as a form of CDR. If HWP is to be included in the potential CDR pathways discussed here, I think it is really important to present some evidence that all of the carbon flows involved actually lead to net removal, and under what conditions this is the case.

**We agree that the finite lifetime of HWPs leads to emissions eventually. There is no authoritative definition of how long a product needs to store CO2 to count as CDR, but the IPCC definition of CDR accounts for product storage in general: "Anthropogenic activities removing carbon dioxide (CO2) from the atmosphere and durably storing it in geological, terrestrial, or ocean reservoirs, or in products." (IPCC glossary). We follow the State of CDR report (Smith et al., 2024) in counting HWPs towards durable products accountable as CDR. We have added references to IPCC and State of CDR reports to justify our choice of counting HWPs to CDR.**
**We have further adjusted the text in lines 460ff, 476ff, 920-922 to make it very clear that our estimate of CDR does not include activities other than re/afforestation. As we do not include HWPs in our estimate of ELUC we find an in-depth discussion under which conditions HWPs are net removals or not beyond the scope of our paper.**
**With these clarifications to re/afforestation and the clarification to Line 210 above ("by forestry we meant forest regrowth in shifting cultivation cycles"), the 1.2 GtC/yr in the summary are unambiguous.**

**Changes in ms:**
*New text from line 460:*
*Other CDR activities related to land use but not fully accounted for in our ELUC estimate include the transfer of carbon to harvested wood products (HWP), bioenergy with carbon capture and storage*

*(BECCS), and biochar production (Babiker et al., 2022; Smith et al., 2024). The different bookkeeping models all represent HWP but with varying details concerning product usage and their lifetimes. BECCS and biochar are currently only represented in bookkeeping and TRENDY models with regard to the CO2 removal through photosynthesis, without accounting for the durable storage. HWP, BECCS, and biochar are typically counted as CDR once the transfer to the durable storage site occurs and not when the CO2 is removed from the atmosphere, which complicates a direct comparison to the GCB approach to quantify annual fluxes to and from the atmosphere. We provide estimates for CDR through HWP, BECCS, and biochar based on independent studies in Section 3.2.2, but do not add them to our ELUC estimate to avoid potential double-counting that arises from the partial consideration of HWP, BECCS, and biochar in the bookkeeping and TRENDY models and to avoid inconsistencies from the temporal discrepancy between transfer to storage and removal from the atmosphere.*

*New text from line 476:*
*While some CDR involves CO2 fluxes via land-use and is included in our estimate of ELUC (re/afforestation) or provided from other data sources (biochar, HWP, and BECCS), other CDR occurs through fluxes of CO2 directly from the air to the geosphere.*

*New text from line 920:*
*Though they cannot be compared directly to annual fluxes from the atmosphere and are thus not included in our estimate of ELUC, CDR through transfers between non-atmospheric reservoirs such as in durable HWPs, biochar, or BECCS comprise much smaller amounts of carbon: 218 MtC yr-1 have been estimated to be transferred to HWPs, averaged over 2013-2022 (Pongratz et al., 2024).*

Line 1723: Should this be ±220 or a range of 220?

**Corrected, it is ±220.**

Line 1725-1726: Of course, the authorship of the IPCC chapter and the Forster et al paper is basically the same, so the "backing of the IPCC" seems to evoke some mythical other identity that is a bit of an artifact. I would suggest removing this sentence.

**Agreed, sentence removed now.**

---

## Author Comment (AC2)

Global Carbon Budget 2024

Response to Referee 2

It is an impressive annual update and an increasingly significant body of work. I have some minor comments below to improve readability. On a minor note, the style of the figures is quite different and could be harmonised, but this doesn't detract from the results, discussion and conclusion.

Specific comments:

Line 431 remove "(" and add comma

**Done, sentence rephrased to: "*Four bookkeeping approaches were used to quantify gross emissions and gross removals and the resulting net $E_{LUC}$, the updated estimates each of BLUE (Hansis et al., 2015), OSCAR (Gasser et al., 2020), and H&C2023 (Houghton and Castanho, 2023), and the new estimates of LUCE (Qin et al. 2024).*"**

Line 627 what are hidden neurons - bootstrapping?

**The term "Hidden Neurons" refers to artificial neurons in a neural network layer referred to as "hidden layer" in the literature. Hidden layer simply refers to a network layer that is neither input nor output layer (thus hidden). For more clarity we rephrased the sentence to: "*To avoid overfitting, the neural network training was done using a Monte Carlo approach, with a variable number of artificial neurons (varying between 2-5) and 20% of the randomly selected training data were withheld for independent internal testing*"**

Line 654 summer is not always boreal, also this sentence has something wrong grammatically

**Corrected and slightly rephrased to: "*Like for the ocean forecast, the land $CO_2$ sink forecast for the year 2024 is based on (a) the historical (Lan et al., 2024) and our 2024 estimate of atmospheric $CO_2$ concentration, (b) the historical and our 2024 estimate of global fossil fuel emissions, and (c) the boreal summer (June, July, August) Oceanic Niño Index (ONI) (NCEP, 2024).*"**

Line 659 - missing neurons?

**Similar to above we rephrased for clarity: "*To avoid overfitting, the neural network was trained with a variable number of artificial neurons (varying between 2-15), larger than for $S_{OCEAN}$ prediction due to the stronger land carbon interannual variability. As done for $S_{OCEAN}$, a Monte Carlo type pre-training selects the optimal number of artificial neurons based on 20% withheld input data, and in a second step, an ensemble of 10 forecasts is produced to provide the mean forecast plus uncertainty*"**

Line 666 and subsequent, shouldn't xCO2 be written XCO2?? not mole fraction of $CO_2$

**Done**

Line 1047 this statement seems self-evident something is always greater than nothing

**Sorry, we cannot see which statement in the manuscript this referee comment refers to.**

Line 1100 Orphan "("

**Done**

Line 1106 "does" should be "do"

**No, the subject of this sentence is singular (the effect of climate change)**

line 1176 strange formatting

**Corrected**

Line 1212 add "may" underestimate

**Done**

Line 1247 This sentence doesn't quite make sense. "As for the ocean…"

**Corrected**

Line 1348 should be "is derived from"

**Done**

Tables: 2 in $CO_2$ should be a subscript

**Done**